# The 79°N Glacier cavity modulates subglacial iron export to the NE Greenland Shelf

Stephan Krisch [1], Mark James Hopwood [1], Janin Schaffer [2], Ali Al-Hashem[1], Juan Höfer [3], Michiel M. Rutgers van der Loeff[2], Tim M. Conway [4], Brent A. Summers[4], Pablo Lodeiro [1,5], Indah Ardiningsih[6], Tim Steffens[1] & Eric Pieter Achterberg [1✉]

Approximately half of the freshwater discharged from the Greenland and Antarctic Ice Sheets enters the ocean subsurface as a result of basal ice melt, or runoff draining via the grounding line of a deep ice shelf or marine-terminating glacier. Around Antarctica and parts of northern Greenland, this freshwater then experiences prolonged residence times in large cavities beneath floating ice tongues. Due to the inaccessibility of these cavities, it is unclear how they moderate the freshwater associated supply of nutrients such as iron (Fe) to the ocean. Here, we show that subglacial dissolved Fe export from Nioghalvfjerdsbrae (the '79°N Glacier') is decoupled from particulate inputs including freshwater Fe supply, likely due to the prolonged ~162-day residence time of Atlantic water beneath Greenland's largest floating ice-tongue. Our findings indicate that the overturning rate and particle-dissolved phase exchanges in ice cavities exert a dominant control on subglacial nutrient supply to shelf regions.

[1] GEOMAR Helmholtz Centre for Ocean Research Kiel, Kiel, Germany. [2] Alfred Wegener Institute, Helmholtz Centre for Polar and Marine Research, Bremerhaven, Germany. [3] Escuela de Ciencias del Mar, Pontificia Universidad Católica de Valparaíso, Valparaíso, Chile. [4] College of Marine Science, University of South Florida, St Petersburg, FL, USA. [5] Department of Chemistry, University of Lleida – Agrotecnio-Cerca Centre, Lleida, Spain. [6] NIOZ Royal Netherlands Institute for Sea Research, Utrecht University, Den Burg, Texel, The Netherlands. ✉email: eachterberg@geomar.de

Glacial weathering processes enrich meltwater in iron (Fe) and silicic acid (Si(OH)$_4$) relative to other nutrients such as nitrate and phosphate[1,2]. The availability of Fe limits primary production across much of the high latitude oceans, including most of the Southern Ocean[3] and parts of the sub-polar North Pacific[4] and North Atlantic Oceans[5]. Whilst Si(OH)$_4$ concentrations are relatively high around the coastline of Antarctica, low Si(OH)$_4$ availability across most of the Arctic constrains diatom growth[6,7] and thus Si(OH)$_4$ availability directly influences the magnitude of the Arctic biological carbon pump[8]. The annual supply of Fe and Si(OH)$_4$ from glacial meltwater, icebergs and associated particles into the high latitude oceans is presently assumed to scale with solid and liquid discharge volume[9–11]. With accelerating mass loss from the Greenland and Antarctic Ice Sheets, the associated nutrient supply to the ocean is therefore projected to increase in magnitude under future climate scenarios[11,12] with potential implications for high latitude primary production[6,13,14].

Dissolved Fe (dFe) and Si(OH)$_4$ both exhibit non-conservative behavior across estuarine salinity gradients[15,16] which means that large differences can arise between the net and gross fluxes of these nutrients into the ocean at the glacier-ocean interface[17]. By far, the largest uncertainty in present estimates of annual dFe (e.g., ref. [18]) and Si(OH)$_4$ (e.g., contrasting refs. [1,19]) fluxes from Greenland to the ocean arises in how non-conservative processes are parametrized. Furthermore, there is little agreement between present model estimates concerning the magnitude of ice sheet-ocean fertilization, which partially arises because of a poor mechanistic understanding of processes occurring at the data-deficient ice-ocean interface[14,20]. Multiple spatially overlapping factors including scavenging[16,21], particle dissolution[19], benthic recycling of nutrients[22,23], entrainment of nutrient-rich deep marine waters[24,25] and biological drawdown[26] affect nutrient distributions close to calving ice faces.

Subglacial discharge plumes from marine-terminating glaciers may moderate fjord-scale biogeochemistry and primary production on timescales of hours-to-weeks after freshwater discharge enters the marine environment[25,26]. For many larger ice shelves however, freshwater first experiences prolonged residence times in subglacial cavities. For example, inflowing water masses are resident for approximately 2 years underneath the Ross Sea Ice Shelf[27], and ~162 days underneath Nioghalvfjerdsbrae (commonly referred to as the 79° North Glacier)[28]. These dark subsurface cavities provide case studies to understand changing nutrient availability by contrasting the inflowing and outflowing properties of water masses to derive the integrated change from all glacier associated processes in the absence of confounding effects of summertime phytoplankton blooms. Given the non-conservative behavior of both Fe and Si(OH)$_4$ across salinity gradients, and the multiple processes affecting their exchange between the particle and dissolved phases, we hypothesize that these prolonged residence times may decouple the outflowing chemical enrichment from the ice sheet-derived inputs.

Here we investigate constraints on lateral Fe and macro-nutrient supply into the Atlantic from the large cavity underneath the floating ice tongue of the Nioghalvfjerdsbrae glacier (hereafter the 79NG). During GEOTRACES cruise GN05, a comprehensive survey of Fe fractions (Table 1) alongside other trace elements and macronutrients was conducted up- and downstream of the 79NG following the main pathways of Atlantic water inflow and glacial meltwater outflow close to the peak of the melting season (August 2016). Due to favorable weather and ice conditions, oceanographic sampling was conducted immediately adjacent to the ice tongue, providing, to our knowledge, a unique dataset to investigate nutrient fluxes exiting a subglacial cavity where the water residence time is well constrained from prior surveys[29] as well as 5 moorings on the shelf which were deployed for the same year where we present nutrient distributions, 2016–2017[28].

## Results and discussion

**Study region.** Hydrographic and trace metal clean chemical profiles were conducted at 11 stations upstream and downstream of the 79NG (S1-11; Fig. 1). Three major water masses are present on the NE Greenland Shelf: Polar Surface Water (PSW, $\sigma_\Theta < 26.1$ kg/m$^3$), Atlantic Intermediate Water (AIW, $\sigma_\Theta > 27.73$ kg/m$^3$), and Atlantic Intermediate Water modified by mixing with glacial meltwaters (mAIW, $\sigma_\Theta = 27.00$–$27.73$ kg/m$^3$)[28]. The ocean circulation on the continental shelf is steered by the underlying shelf bathymetry. The East Greenland Current follows the shelf break forming the eastern limb of the anti-cyclonic shelf circulation[30,31]. On the continental shelf, AIW follows a C-shaped trench system (passing stations S8–S10; Fig. 1) towards the vicinity of the 79NG[32].

Warm (0.6–1.3 °C) and saline (S = 34.58–34.80) AIW found at depths >268 m (Supplementary Fig. 1a, b) induces basal melting along the floating 79NG ice-tongue that covers the entire length of Nioghalvfjerdsfjorden[29,33]. This inflow of heat presently occurs throughout the year and thus, as the basal melt of the ice tongue accounts for ~89% of freshwater exiting the cavity[28], there is less pronounced seasonality in mAIW outflow than would be expected in a system dominated by subglacial discharge which largely occurs over a narrow time period in summer[34]. Basal melt and subglacial discharge at 79NG combined result in a total cavity freshwater flux of 0.63 ± 0.21 mSv (19.9 ± 6.6 km$^3$/yr)[28], which is roughly equivalent to 2% of annual runoff from the Greenland Ice Sheet (2016 values)[35]. Mixing between inflowing AIW, subglacial discharge, and basal ice melt dictates the properties of mAIW which exits the 79NG cavity as a subsurface flow (Fig. 2). The mean residence time of water in the cavity for 2016–2017 was ~162 days[28]. The mAIW is observed exiting the cavity between 96–268 m depth at the glacial front (S1) (Fig. 3a). Oxygen stable isotope measurements ($\delta^{18}$O) and calculated meteoric freshwater

**Table 1 Definition of Fe fractions in seawater.**

| Fraction | Size | Description | Reference |
|---|---|---|---|
| Soluble Fe (sFe) | <0.02 μm | Filtrate | Fitzsimmons et al. (2015) |
| Dissolved Fe (dFe) | <0.2 μm | | |
| Colloidal Fe (cFe) | 0.02–0.2 μm | Determined as difference between dFe and sFe (cFe = dFe–sFe) | |
| Ligand-bound dFe (dFeL) | <0.2 μm | Dissolved Fe bound to organic molecules (samples for iron ligands are filtered; dFeL is part of the dFe pool) | Ardiningsih et al. (2020) |
| Labile particulate Fe (LpFe) | >0.2 μm | Reactive particulate fraction, including nanoparticulate ferrihydrite physically immobilized on particles and cellular Fe | Berger et al. (2008) |
| Total dissolvable Fe (TdFe) | Unfiltered Seawater | Dissolved and particulate Fe released after storage at pH 1.9 | Edwards and Sedwick (2001) |

fractions (Fig. 3b, c) corroborate the lateral transport calculations as per ref. [28], demonstrating that mAIW is subsequently carried with the anti-cyclonic shelf-circulation away from the fjord mouth following the Westwind Trough (via S3 and S5) towards the shelf break (S6) in a north-easterly direction[29,30].

From the shelf-break proceeding towards the 79NG terminus, hydrographical measurements show freshening of surface waters (Fig. 3), a deepening of the AIW core, and thickening of the mAIW and PSW layers driven by freshwater release from the

79NG and its neighboring glacier, Zachariæ Isstrøm. A comparison of mean water properties entering ($S = 34.81$ for AIW) and exiting the 79NG subglacial cavity ($S = 34.41$ for mAIW) suggests a meltwater to AIW ratio of 1:84 indicative of strong dilution of glacial freshwater close to source. This physical framework provides an interesting case study to quantify the net effects of subglacial processes on Atlantic water properties. During GN05 it was possible to sample AIW and mAIW immediately adjacent (<1 km distance) to the glacier terminus i.e. prior to any subsequent modification from dilution or uptake due to summertime primary production (Fig. 1), representing the only site where AIW inflow and mAIW outflow was simultaneously observed.

**Geographical nutrient distributions**. Note that due to the deployment of two sampling rosette systems (ultraclean CTD for contamination prone trace metal sampling and large CTD for general sampling), the dataset for macronutrients is larger than for trace metals. For consistency and to facilitate a single statistical analysis, we primarily discuss the distribution of all components at ultraclean CTD stations S1–11 (Fig. 1), but also include additional analysis (see Supplement) demonstrating that there are no significant changes to our interpretation when the two data sets are considered separately or in combination.

Similar to summertime macronutrient distributions elsewhere in Greenland's glacier fjords, concentrations of nitrate ($NO_3$, Fig. 4a) and phosphate ($PO_4$, Fig. 4b) in the 79NG coastal region were depleted in surface waters (~0.1 μM $NO_3$, ~0.3 μM $PO_4$) and increased with depth (up to 12 μM $NO_3$ and 0.9 μM $PO_4$)[24,26]. In contrast to the relatively homogenous distribution of $NO_3$ within mAIW and AIW, $Si(OH)_4$ concentrations were enriched sporadically at multiple depths close to the glacier front across all 3 water masses (range 2.3–41.3 μM for all stations >19°W), indicating a glacier-associated $Si(OH)_4$ source (Fig. 4c). Distinct maxima in $Si(OH)_4$ enrichment were observed in PSW (30.4 and 36.3 μM, <20 m at large CTD station 228), and once in mAIW (41.3 μM at 100 m at large CTD station 229). Elevated $Si(OH)_4$ concentrations (>8 μM) were however absent throughout the inner-shelf area (S2–S5, Fig. 4c).

Positive Fe anomalies were observed across the region (Fig. 5). A pronounced maximum in labile particulate Fe (LpFe) was found within mAIW (~100 nM, Fig. 4d) coinciding with

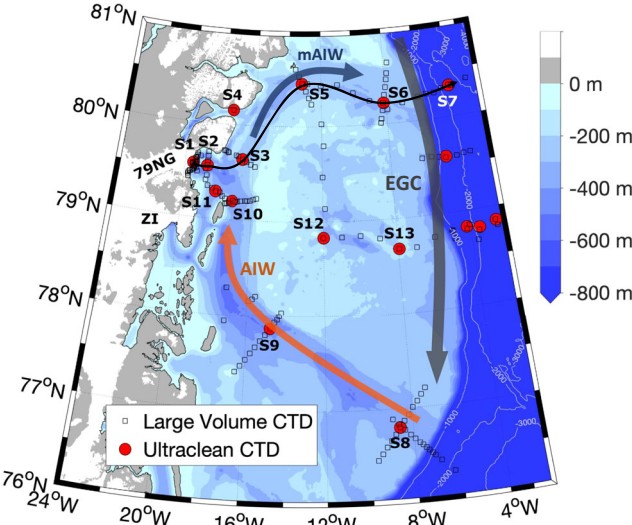

**Fig. 1 Topography and location of sampling stations on the NE Greenland Shelf.** Norske Trough (S8–10), Nioghalvfjerdsbrae (79NG) front (S1) and Dijmphna Sund side-exit (S4, a sill station), Nioghalvfjerdsfjorden Bay (S2), Westwind Trough (S3, S5), the shelf break (S6) and a station in Fram Strait (S7). One station (S11) was sampled in proximity to Zachariæ Isstrøm (ZI). Two further stations were sampled in the center (S12) and outer (S13) NE Greenland Shelf. Other stations from the same cruise, not discussed herein, are marked for completion (see ref. [44] for an overview). The black thin line from S1 to S7 highlights the section from which depth profiles are discussed in Figs. 3–5. The predominant circulation of Atlantic Intermediate Water (AIW), modified AIW (mAIW) and the East Greenland Current (EGC) is shown.

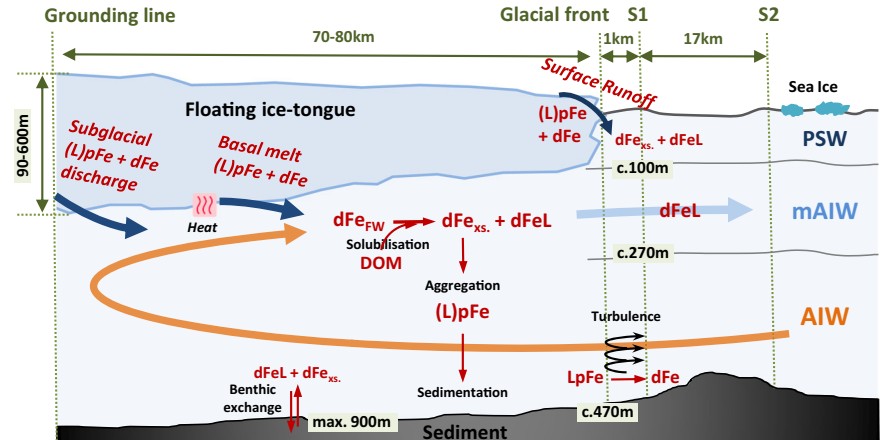

**Fig. 2 Concept figure illustrating the subglacial cavity underneath the Nioghalvfjerdsbrae (79NG) floating ice tongue and illustrated with key aspects of the Fe cycle.** The majority of glacial freshwater is released as basal melt into Atlantic Intermediate Water (AIW) forming modified AIW (mAIW). A further portion of glacial freshwater is discharged into Polar Surface Water (PSW). Only a fraction of subglacial freshwater dissolved Fe ($dFe_{FW}$) is stabilized through the aid of ligands (dFeL) derived from dissolved organic matter (DOM). The majority of freshwater-derived dFe, not stabilized ('excess dFe', $dFe_{xs}$) aggregates and forms (labile) particulate Fe (LpFe, pFe).

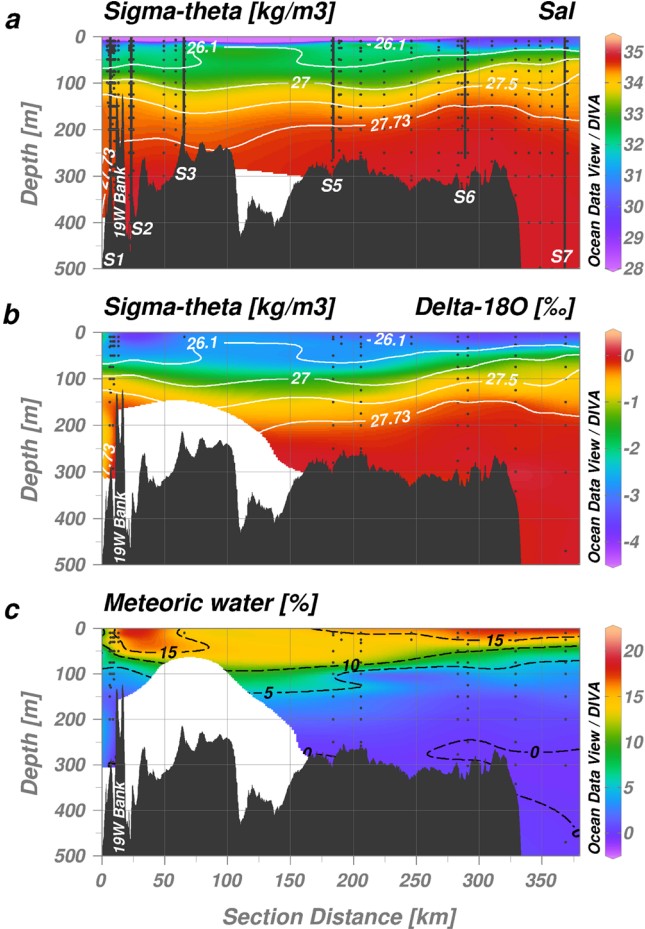

**Fig. 3 Freshwater distribution following the main Nioghalvfjerdsbrae (79NG) discharge into Nioghalvfjerdsfjorden Bay. a** Salinity. Dots (black) indicate the locations of discrete macronutrient measurements, vertical lines (bold black) indicate locations of trace metal profiles as per Fig. 1. **b** δ18O. Dots (black) indicate the locations of discrete oxygen isotope measurements; negative values suggest release of meteoric freshwater. **c** Meteoric water fractions calculated from δ18O and a freshwater reference δ18O value of −20‰[38]. Isopycnal surfaces (white contours) distinguish between Polar Surface Water ($\sigma_\Theta$ <26.1 kg/m³), Atlantic Intermediate Water (AIW, $\sigma_\Theta$ >27.73 kg/m³), and modified AIW (mAIW, $\sigma_\Theta$ = 27.00–27.73 kg/m³). Two thirds of subglacial cavity outflow is observed between $\sigma_\Theta$ = 27.5–27.73 kg/m³ corresponding to depths of 125–268 m at S1[28]. The 19°W Bank between S1 and S2 is bypassed both to its north (main AIW gateway) and south and thus does not impede the flow of AIW. Additional mAIW via Dijmphna Sund outflows between S3 and S5.

maximum light attenuation indicative of a particle-rich layer exiting the cavity (Fig. 4e, Supplementary Fig. 2). Concentrations of LpFe in mAIW and AIW greatly exceeded dFe in the vicinity of the glacier; dFe was equivalent to 7.6 ± 5.5% (mean ± standard deviation) of LpFe concentrations at S1 and S2. Yet similar to the localized nature of the Si(OH)₄ anomalies at the ice front, a clear LpFe maximum was no longer evident 20 km away from the glacier terminus at any depth (at S2). This sharp decline may have occurred through either a loss of particles from the water column (i.e., sedimentation) or from a decrease in the lability of particulate Fe following particle aging[36,37]. Across all stations LpFe constituted 9.8 ± 2.7% to TdFe indicating that the variability in particle Fe lability along the fjord (9.0 ± 2.4% at station S1 and S2) was not pronounced. Combined with the strong gradient in turbidity between S1 and S2, this suggests sedimentation, rather

than particle aging, was the dominant loss process for LpFe from the water column. More modest peaks in LpFe and light attenuation were evident at the sediment-bottom water interface near the glacier terminus (~40 nM at S1 and S2, Fig. 4d) possibly indicating re-suspension of labile particles. Elsewhere across the NE Greenland Shelf, LpFe generally exhibited low nanomolar concentrations 3.6 ± 2.8 nM (Supplementary Fig. 3).

In contrast to LpFe, the distribution of dFe across the shelf was not dominated by mAIW (Fig. 5). Dissolved Fe peaks were generally observed near the surface (Fig. 5b) and coincided with strong negative isotopic shifts in δ18O indicative of meteoric freshwater supply[38,39] (Fig. 3b, c). Modest dFe maxima were observed near the 79NG glacial terminus (S1, 2.3 nM at 25 m depth and 2.1 nM at 50 m depth), but also in the surface waters of Fram Strait (S7, 2.5 nM at 10 m depth) (Fig. 5a). In mAIW, mean dFe concentrations steadily decreased away from 1.3 nM at the glacier terminus (S1) to 0.6 nM beyond the shelf break (S7). Within AIW, dFe was rather uniformly distributed with a concentration of 0.72 ± 0.21 nM (mean for S1, S2 and S8–10). The shelf break station (S6) showed bottom water dFe enrichment, but no similar enrichment was observed throughout the inner shelf region (S1–S5, Fig. 5). Among other environmental factors, this may result from higher turbulence at the shelf break[40].

Surface dFe concentrations close to the 79NG (1.5 nM at 5 m, S1) and neighboring glacier termini (2.5 nM at 10 m, S11) were comparable to, or lower than, concentrations downstream of other Arctic glacier catchments at similar salinities (e.g., 4–7 nM, Kongsfjorden[21]; <3–6 nM, Godthåbsfjord[41]; ~4 nM, Copper River estuary[42]; ~4 nM, Inglefield Bredning[43]). Whilst a pronounced peak in surface dFe was observed beyond the NE Greenland shelf break in Fram Strait (S7, Fig. 5a), the discontinuity with other surface samples suggest that this peak did not arise from local freshwater sources. Note that whilst this only appears as a single elevated surface dFe datapoint in the plotted section (Fig. 5a), elevated dFe concentrations were consistently found in East Greenland Current surface waters across the broader region surveyed by the GN05 cruise (1.1 nM ± 0.7 nM, 0–10°W, $n = 10$)[44].

Long-lived radium isotopes (228Ra $t_{1/2}$ 5.75 years and 226Ra $t_{1/2}$ 1600 years), which are produced continuously from decay of Th in sediment[45], are useful tracers of subglacial influence. Similarly, noble gases provide two further chemical tracers of subglacial influence, due to enhanced Ne and He concentrations in subglacial discharge[46]. He and Ne data from the same expedition demonstrate a relatively uniform subglacial content throughout mAIW and show no He/Ne at any site beyond the Greenland shelf break (Huhn et al., personal communication), which further corroborates the lateral transport calculations based on water mass analysis[28] and highlights the disparity between peaks in dFe close to the glacier termini and over the shelf.

228Ra distributions in surface waters across the region largely reflect an Arctic derived signal from further north[47,48], but the Ra depth profiles obtained at S1 are insightful (Fig. 4f, g). 226Ra and 228Ra both show high activities in PSW (97 ± 8 dpm/m³ 226Ra and 60 ± 15 dpm/m³ 228Ra). In the AIW (300–450 m depth), 226Ra and 228Ra vary around 80 and 17 dpm/m³, respectively, as in other stations sampled over the shelf. At 150–250 m in the mAIW, 226Ra is unchanged while 228Ra is slightly enriched compared to shelf stations at the same depth. At 125 m, we observe a signal of highly elevated 226Ra (111 dpm/m³) and 228Ra (48 dpm/m³), indicative of a sedimentary input to cavity mAIW (with a 228Ra/226Ra ratio of 0.9). Subglacial runoff is estimated to be 0.07 mSv (2.2 km³/yr) or only 0.15% of the cavity overturning rate (calculated from ref. [28]). With the discharge-averaged concentrations found in subglacial runoff of the Leverett Glacier in western Greenland[49] this would, after mixing in the cavity,

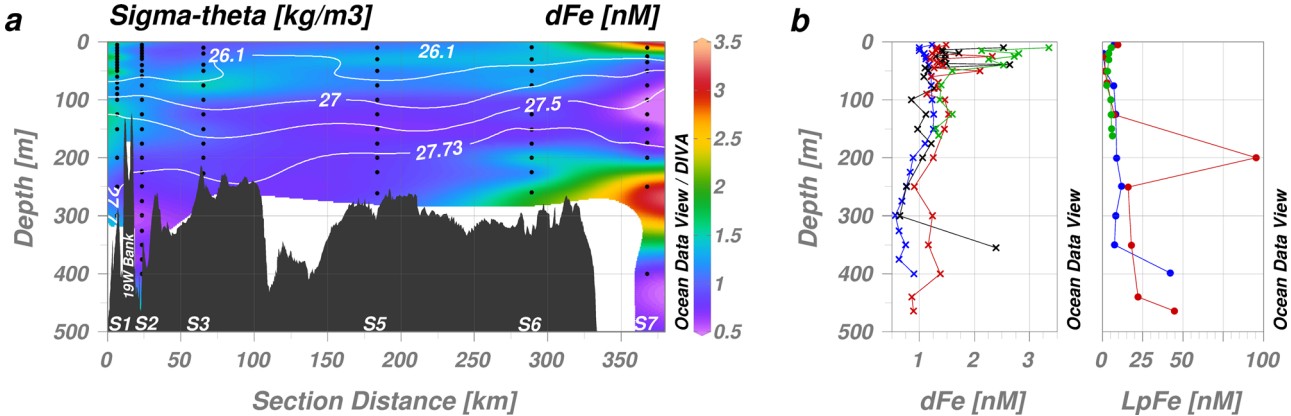

**Fig. 4 Macronutrient, labile particulate Fe (LpFe) and radium (Ra) distributions in Nioghalvfjerdsfjorden. a** Nitrate ($NO_3$), **b** phosphate ($PO_4$), **c** silicic acid ($Si(OH)_4$) and **d** LpFe at the inner-most stations close to the Nioghalvfjerdsbrae (79NG) terminus (S1–S3 as per Fig. 1). Sample depths used for interpolation are shown as black dots. Additional macronutrient stations from large CTD measurements (stations 213, 229, 231–233, 252, 254, section 'Data availability') are shown alongside S1–S3. Isohalines (white contours) distinguish between water masses: Polar Surface Water (<70 m), Atlantic Intermediate Water (AIW, > 270 m) and modified AIW (100–270 m). Station bottom depth is applied to define a basic bathymetry for clarity. **e** Depth profile of light attenuation (S1 in red, S2 in blue, S3 in black) evidences greatest turbidity between 190–250 m depth at S1 corresponding to the main subglacial cavity outflow. **f** Depth profile of $^{226}$Ra near the calving front (S1). **g** $^{228}$Ra near the calving front (S1).

**Fig. 5 Dissolved Fe (dFe) concentrations in Nioghalvfjerdsfjorden and Westwind Trough. a** Cross-section of the main flow path of modified Atlantic Intermediate Water ($\sigma_\Theta = 27.00–27.73$ kg/m$^3$) from Nioghalvfjerdsbrae (79NG, S1) to Fram Strait (S7) following Westwind Trough. The location of the presented section is shown in Fig. 1. Sample depths used for interpolation are shown as black dots. Isohalines (white contours) distinguish between water masses (as per Fig. 3). **b** Depth profiles for dFe (left) and labile particulate Fe (LpFe, right) at the 79NG terminus (S1, red), Nioghalvfjerdsfjorden Bay (S2, blue), Dijmphna Sund (S4, green) and Zachariæ Isstrøm (S11, black).

cause increases of only 0.03 dpm/m$^3$ $^{226}$Ra and 0.35 dpm/m$^3$ $^{228}$Ra in the cavity outflow, negligible compared to the observed signals at 125 m. We conclude that the radium is sedimentary in origin, either emerging from the seafloor or from sediments entrained within the ice-tongue and released into the water column by basal melting.

**Statistical data treatment.** A linear, positive correlation between salinity and macronutrient concentrations for all 3 macronutrients (R$^2$: 0.75 NO$_3$, 0.56 PO$_4$, 0.50 Si(OH)$_4$, Supplementary Fig. 4), indicates relatively conservative behavior and the dominance of AIW as a nutrient source to the region (Supplementary Tables 3 and 4). Whilst the intercepts for all macronutrients are negative, a linear regression between salinity and dFe for all samples collected on the shelf has a zero salinity intercept of 7.34 ± 0.70 nM (Supplementary Fig. 5).

We conducted a redundancy analysis (RDA) that considers chemical, physical and geographical factors that may explain the distribution of nutrients downstream of 79NG, with two additional factors added to the nutrient and oceanographic data for every sampling depth. Firstly, a lateral distance was calculated between each station and the glacier terminus along sections connecting adjacent stations (Fig. 1) e.g. 0.1 km at S1 to 260 km at S6. Secondly, a vertical distance from seafloor to the sampling depth at each station was computed. Additionally, we consider the distribution of other trace elements (Co and Mn) alongside dFe and LpFe to assist our interpretation. Like Fe, Co and Mn are both scavenged type elements; but with subtle differences in their oceanographic distributions. Dissolved Co (dCo) is less prone to scavenging than Fe, and dissolved Mn (dMn) generally shows increased dissolved concentrations in near-surface waters due to a more pronounced positive effect of photochemical cycling on its stability (e.g., refs. [50,51]).

The RDA explains 61% of the variance ($p < 0.001$) shown by all trace elements and macronutrients using four explanatory variables (Fig. 6). There are strong similarities in the factors that explain the distribution of Si(OH)$_4$, PO$_4$ and NO$_3$ with all macronutrients closely associated with increasing salinity and proximity to the seafloor (i.e. AIW inflow and remineralization processes). Dissolved trace elements (dFe, dCo and dMn) cluster with decreasing salinity and increasing proximity to surface waters. Labile particulate trace elements (LpFe, LpCo and LpMn) on the other hand are mainly explained by proximity to the terminus and high turbidity.

Through the use of General Additive Models (GAMs) on water mass properties downstream of 79NG, it is possible to investigate nutrient-distance relationships that are not necessarily linear[52]. GAMs for all nutrients produced good fits to equation 1, where [x] is the concentration of a nutrient, and $c$ is an intercept and s(Distance to terminus | Salinity) represents a term composed of the interaction between salinity and the distance from glacier terminus (R$^2$: 0.773 dFe, 0.878 Si(OH)$_4$, 0.965 NO$_3$ and 0.944 PO$_4$, respectively, with $p < 0.001$ in all cases) (Supplementary Table 5). As per RDA (Fig. 6), the three macronutrients are clearly similar in their behavior whereas dFe displays a distinct distribution which is roughly inverted compared to the macronutrients (Fig. 7). As per linear regression, the output of GAMs at glacier terminus and zero salinity for all macronutrients was negative (Supplementary Table 5), whereas predicted dFe concentration was 3.13 ± 0.96 nM (SE).

$$[x] = c + s(\text{Distance to terminus}|\text{Salinity}) \qquad (1)$$

Having considered the oceanographic context of inflowing AIW and outflowing mAIW (Figs. 1 and 2), one further statistical

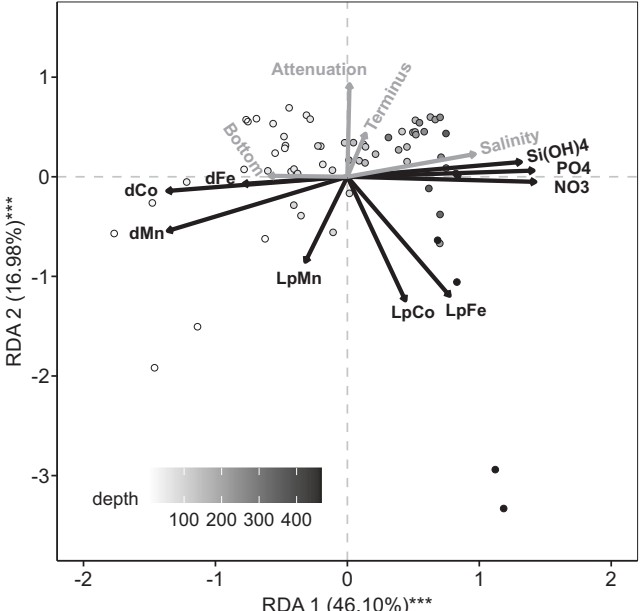

**Fig. 6 A redundancy analysis (RDA) illustrating trends in water column properties on the NE Greenland Shelf (stations S1 to S11).** The angles between arrows represent the positive (0°), negative (180°) or no relationship (90°) among the variables of (i) dissolved and labile particulate fractions of Fe (dFe, LpFe), Mn (dMn, LpMn) and Co (dCo, LpCo), (ii) the macronutrients silicic acid (Si(OH)$_4$), phosphate (PO$_4$) and nitrate (NO$_3$), (iii) the distance to the seafloor ('bottom', in m), (iv) the distance from the terminus along the cruise sections connecting adjacent stations ('terminus'), and (v) attenuation that is inversely proportional to turbidity (arbitrary units).

test is to determine whether or not significant changes occur in the concentrations between these two water masses (Supplementary Table 6). Irrespective of where mAIW and AIW are defined, there is always an enrichment in dFe contrasting mAIW with AIW ranging from 0.46 ± 0.27 to 0.67 ± 0.23 nM depending on the stations used to define each water mass (Supplementary Table 6). Conversely, there is no significant ($p < 0.05$) change in Si(OH)$_4$ concentrations contrasting the two water masses (maximum calculated change 0.38 ± 3.6 μM), suggesting that any Si(OH)$_4$ inputs from ice melt, subglacial discharge and the subsequent re-working of sediments are too small to be detected as a regionally significant nutrient flux. For NO$_3$, there is a slight decline in concentration from AIW to mAIW of up to −1.0 ± 1.4 μM, however its significance depends on the exact distance from the terminus where mAIW is defined. This suggests increasing NO$_3$ drawdown with time, and distance, from the termini. For PO$_4$, there is a significant ($p < 0.05$) decline of as much as −0.17 ± 0.12 μM contrasting mAIW with AIW which is evident irrespective of the stations considered. Importantly, this trend is already evident at S1 (0.1 km from the terminus) and thus must partially reflect a net PO$_4$ loss in the subglacial cavity or close to the glacier termini as has been reported in limited examples elsewhere[25,53].

The distribution of nutrients downstream of 79NG suggests that the only nutrient enriched significantly by glacier-associated processes is dFe. The negative zero salinity intercepts for Si(OH)$_4$, NO$_3$ and PO$_4$ calculated by GAMs and linear regressions alike must reflect a change of gradient in the macronutrient-salinity relationship at salinities <24 (i.e., in the subglacial cavity). This is not unexpected given that similar shifts occur at low salinities in other large glacier fjords with similar two-dimensional inflow and outflow dynamics (e.g., refs. [24,25]).

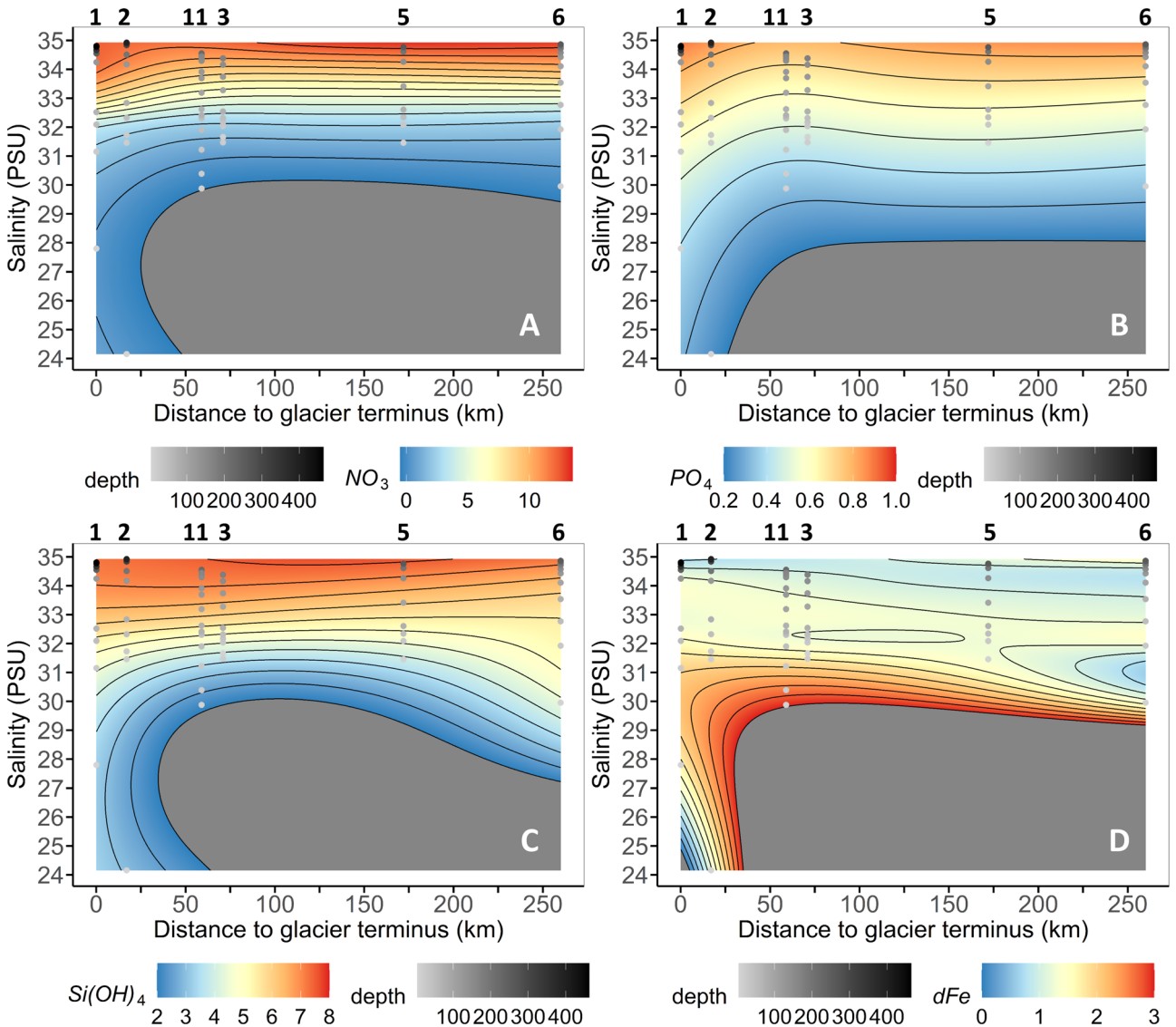

**Fig. 7 Predicted concentration of all nutrients for stations S1 to S11 using a General Additive Model (GAM) generated by the interaction between salinity and distance to the glacier terminus. A** Nitrate ($NO_3$), **B** phosphate ($PO_4$), **C** silicic acid ($Si(OH)_4$) and **D** dissolved Fe (dFe). Each contour plot shows the predicted nutrient concentration while the dots represents the conditions (salinity, distance to glacier terminus and depth) of the data used to generate the GAM. Station numbers are shown above each panel. Concentrations for $NO_3$, $PO_4$ and $Si(OH)_4$ are in µM, whereas dFe concentrations are in nM.

**Insights into lateral dFe export**. A GAM fit and a linear regression produced comparable dFe concentrations at zero salinity of $3.13 \pm 0.96$ and $7.34 \pm 0.70$ nM, respectively. The difference likely arises because the GAM fit considers the relationship between distance from the terminus and salinity, while the linear regression solely considered salinity. These values represent the dFe concentration remaining after significant loss of dFe has occurred across the salinity gradient. This is useful in a model context because estuarine losses are challenging to constrain and occur on a scale which is sub-grid even in high resolution models. Alternatively, when a removal factor is multiplied by freshwater concentration to estimate lateral fluxes, small changes in the estuarine removal factor, for example contrasting rough lower and upper estimates for estuarine dFe loss of 90% and 99%, produce very large differences in the net dFe flux[18]. A further implication of the difference between a GAM fit based on salinity and distance, and a regression for salinity alone, is that dFe loss is continuing downstream of the 79NG terminus in mAIW. This is

also evident when considering the station-by-station dFe concentration from S1 to S7 (Fig. 8).

Further insight into the nature of dFe sources close to 79NG can be gained from stable dFe isotopic composition ($\delta^{56}Fe$, Supplementary Table 9). Isotopically light $\delta^{56}Fe$ values were observed for the local dFe maxima in PSW at the Nioghalvfjerdsbrae glacial front ($-0.12 \pm 0.09$‰ at 25 m, S1) and likewise near Zachariæ Isstrøm ($-0.60 \pm 0.09$‰ at 10–19 m, and $-0.89 \pm 0.09$‰ at 39 m, S11). These light $\delta^{56}Fe$ values likely reflect recent meltwater discharge as dFe in glacial meltwater around Greenland is typically enriched with light isotopes potentially due to reductive dissolution and dissolved $Fe^{2+}$ in subglacial environments[18]. Observed freshwater dFe $\delta^{56}Fe$ values in Arctic glacier outflows include Russell Glacier (West Greenland, range $-0.64$ to $-2.12$‰), Glacier 'G' (East Greenland, range $-0.19$ to $-0.92$‰), Kongsfjorden (Svalbard, range $-0.11$ to $+0.09$‰) and the Copper River (Alaska, $-0.81$‰)[18,21,54].

In contrast to the isotopically light dFe signal in surface water near the Nioghalvfjerdsbrae glacial front, mAIW from 125-200 m

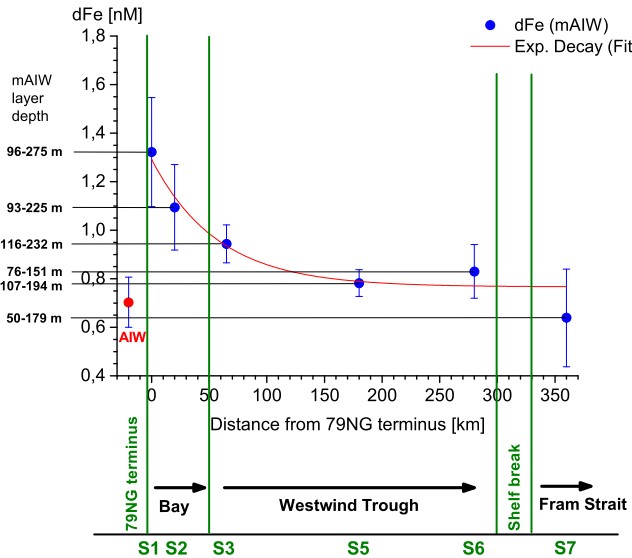

**Fig. 8 Dissolved Fe (dFe) concentrations in modified Atlantic Intermediate Water (mAIW) downstream of the Nioghalvfjerdsbrae (79NG) terminus.** Stations S1 to S6 follow the main outflow path via Nioghalvfjerdsfjorden Bay (S2) towards Fram Strait (S7). Whiskers show the standard deviation to the mean mAIW dFe concentration. Quasi-exponentially decreasing dFe concentrations (red curve) with distance from the 79NG terminus were observed. Station S2 functions as reference station for the dFe concentration of AIW prior to entering beneath the ice-tongue (red closed circle).

at the same station (+0.07 ± 0.09‰, S1) had heavier, near-crustal $\delta^{56}$Fe values (~+0.1‰)[55], equivalent to the isotopic composition of the West Spitsbergen Current (+0.15 ± 0.09‰). It is clear that mAIW is enriched in dFe compared to AIW and that this enrichment occurs either at the glacier terminus or underneath it (Fig. 8). We therefore suggest that any primary isotopically-light $\delta^{56}$Fe signature originating from dissolved $Fe^{2+}$ in subglacial discharge and basal meltwater is lost during extensive reworking and differential loss of dissolved, ligand-bound, particulate and sedimentary phases within the subglacial cavity, as well as exchange between the different phases[56–58]. Cavity reworking is consistent with Ra data showing a sedimentary rather than direct subglacial signal emerging from the cavity (Fig. 4).

Further evidence for notable transformations in the fractionation of Fe between glacial and saline environments can be found when specifically considering the colloidal fraction. Dissolved Fe concentrations herein are inclusive of colloidal Fe (Table 1) which is thought to be released from the surface of glacier-derived particles[59] and to constitute the majority (98-99%) of the dFe input from glaciers into the ocean[60]. Yet due to a paucity of cFe data in marine environments, it has not been possible previously to assess the cFe contribution to lateral export. The 79NG terminus (S1) exhibited cFe concentrations, calculated as the difference between measurements of sFe and dFe, of ~0.1 nM in the top 100 m excluding one outlier of 0.6 nM at 50 m which could be attributed to recent glacial freshwater input (Supplementary Fig. 6). Colloidal Fe concentrations of ~0.7 nM were observed at 100–400 m and an anomalously low concentration (<0.1 nM) near the bottom (460 m depth), possibly because of enhanced scavenging.

Within the inner-shelf region (S1, S2, S4) cFe constituted 3–54% (mean 27%) of dFe (Fig. 9). The sFe and cFe fractions are therefore of roughly comparable importance for Fe availability downstream of 79NG. This contrasts with freshwater Fe fractionation in glaciated catchments where cFe is 20–60 times

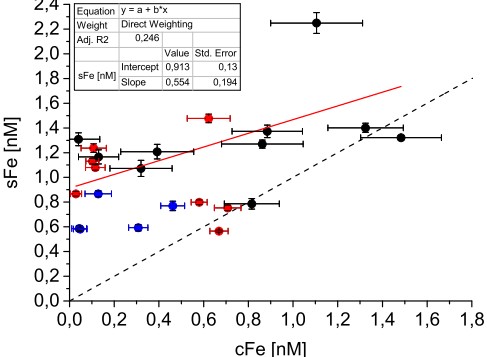

**Fig. 9 Fractions of dissolved Fe near the Nioghalvfjerdsbrae (79NG) glacier terminus.** Correlation of soluble Fe (sFe) and colloidal Fe (cFe) at S1 (red dots), S2 (blue) and S4 (black). Standard error (whiskers) of cFe obtained from error propagation and corresponds to the square root of the sum of squares. Linear fit (red line) combines S1, S2 and S4, and is calculated with direct weighting of standard errors. Dashed line depicts 1:1 ratio of sFe and cFe.

greater than sFe concentration[2]. Our findings are consistent with evidence from glacial meltwater input to Kongsfjorden (Svalbard) where colloidal/nanoparticulate Fe (0.02–0.4 μm) appears to preferentially precipitate during estuarine mixing[21]. The $\delta^{56}$Fe composition of mAIW and high fraction of sFe:cFe in mAIW compared to that expected in fresh glacial meltwater therefore all indicate pronounced re-working of Fe fractions within the ice cavity. By contrast, in surface waters near glacier termini, the concentration of cFe was higher and dFe was isotopically lighter.

**What controls dFe export?** Having quantified the enrichment of dFe in mAIW and attributed the increase to sedimentary processes occurring beneath the floating 79NG ice tongue, a critical question is how this dFe enrichment scales with environmental change. Would lateral dFe export be expected to increase with increasing discharge, turbidity or glacier terminus retreat—changes which are all anticipated as the ice tongue continues to disintegrate and regional warming continues?

As widely observed in both meltwater[59] and near-shore waters in glaciated regions[42,61], LpFe concentrations close to the 79NG terminus vastly exceeded dFe concentrations. Our hypothesis of substantial reworking of Fe phases in the glacier cavity supports the notion that LpFe is an important pool of labile Fe subject to dynamic exchange with dFe. Yet, LpFe and dFe were not closely correlated; increased dFe was notably not observed either at locations with high LpFe at the sediment-bottom water interface or in the mAIW outflow close to the 79NG terminus (Fig. 4d). The divergent behavior of LpFe and dFe is also evident from RDA (Fig. 6).

Considering the whole dataset, there was no clear correlation between dFe and either TdFe or LpFe (Fig. 6, Supplementary Fig. 7). For surface waters, this could simply reflect fast drawdown of dFe by biota while LpFe and TdFe remain in suspension. For example, downstream of the Pine Island Glacier (Antarctica) a faster decline in dFe than TdFe concentrations is attributed mainly to rapid biological uptake of dFe[62]. However, the absence of a correlation between dFe and either LpFe or TdFe in the 79NG dataset remains even if surface water masses are excluded. Dissolved Fe and total dissolvable Fe data from other, smaller Arctic glacier catchments including Godthåbsfjord[41], Kongsfjorden (Svalbard)[21], Bowdoin Fjord[43] and the Gulf of Alaska[16,61] corroborate our observations near the 79NG of intense localized particulate Fe enrichment and decoupling of dFe

and LpFe concentrations. This suggests that the distribution of dFe downstream of the 79NG is not unusual compared to other Arctic glacier fjords despite the presence of an ice tongue. The independence of dFe and particulate Fe distributions supports the hypothesis that factors other than particulate Fe availability control dFe distribution across the region. Across the NE Greenland shelf, dFe concentration was relatively constant (1.14 ± 0.48 nM, range 0.36–3.4 nM across S1–13) despite strong spatial gradients in LpFe (range 0.1–95 nM) and TdFe (range 2–1020 nM). Although there are inevitably biases in sampling of particles from water collectors depending on the apparatus used[63], the observed relationship between dissolved and particulate fractions of Fe is fundamentally different to other metals. For example, an increase in dMn at enhanced particle concentrations is evident at the 79NG marine terminus (Supplementary Figure 7).

The role of particle-bound LpFe as a large pool of labile Fe should also be considered alongside the role of ligands, small organic molecules that bind Fe, as critical agents for maintaining dFe in solution[64]. Organic ligands are able to maintain dFe concentrations in seawater >1 nM[65], an order of magnitude greater than inorganic dFe solubility under oxic conditions[66]. The ability of ligands to maintain dFe in solution depends on a combination of their concentration and binding strength, with the binding strength depending on many factors including pH, salinity, temperature and the nature of organic material present[67,68]. Particle surfaces also compete with ligands to bind dFe and thus at high particle loadings, increasing LpFe can coincide with decreasing dFe concentrations. This phenomenon is not unique to glacier-derived particles and has been observed following sediment resuspension and dust deposition in other marine environments (e.g. refs. [69,70]). In this context, LpFe may be better described as a buffer of dFe rather than a dFe source.

Analysis of dFe-binding ligands from the GN05 cruise suggests bacterial remineralisation and organic degradation have transformed the pool of ligands close to the 79NG terminus, resulting in elevated ligand concentrations but overall weaker ligands that are in total less efficient in stabilizing dFe[64]. The absence of ligand production in the euphotic zone in combination with degradation of organic material[65] underneath 79NG may explain why dFe-binding ligands, generally undersaturated on the shelf (45 ± 17% at S2, S8–10), were approaching saturation at the terminus (63 ± 13% at S1) with a ligand concentration of 2.3 ± 0.2 nM eq. Fe and a corresponding dFe concentration of 1.5 ± 0.3 nM[64]. It is well established that ligand properties and ligand concentration are a limiting factor for what fraction of glacier derived LpFe is dispersed as dFe[61,71] and so the concentration of ligands, which ranged up to 2.6 nM eq. Fe, places a cap on the dFe concentration that can be laterally exported. Yet the outcompeting of ligands by scavenging at the 79NG terminus implies further subtleties to how ligands moderate downstream dFe supply[64]. The optimum conditions for achieving ligand saturation and thus maximizing lateral transfer of ligand-bound dFe may be intermediate, rather than high turbidity and LpFe.

**Implications for lateral dFe transport**. Observations herein could be used to estimate the dFe flux out of the 79NG cavity in several ways. The 7.34 ± 0.70 or 3.13 ± 0.96 nM extrapolated freshwater concentrations from linear regression (Supplementary Table 3) and GAM (Supplementary Table 5) can be combined with the total cavity freshwater flux of 0.63 ± 0.21 mSv (19.9 ± 6.6 km³/yr)[28] to estimate net dFe outflow to the NE Greenland Shelf for comparison with existing literature. This approach produces dFe fluxes of 3.5 ± 2.2 (GAM) or 8.1 ± 3.5 Mg/yr (linear regression), and is inclusive of dFe loses occurring prior to S1, and so is therefore

more useful in an oceanographic context than fluxes derived from surface runoff, i.e., prior to estuarine loss. However, this approach also assumes that the lateral dFe flux scales with freshwater discharge. The distribution and role of Fe ligands[64], the $\delta^{56}$Fe and Ra signatures of mAIW, and the relative concentrations of sFe and cFe downstream of 79NG are all more consistent with cavity overturning dictating the lateral dFe flux. Inflowing AIW provides the heat responsible for basal ice melt of the floating ice tongue and the ligands required to maintain dFe in solution. The rate of AIW inflow also controls flushing of the subglacial cavity. Given the very limited contribution of freshwater input to cavity overturning (~1.4%)[28], lateral chemical fluxes scaled to freshwater discharge are likely very different from those scaled to overturning. To estimate fluxes from overturning, the dFe concentration in mAIW can be contrasted with the dFe concentration in AIW at any specific flux gate (Fig. 8) and combined with the cavity overturning flux of 46 ± 11 mSv. The largest (smallest) change in dFe concentration between mAIW and AIW attributable to processes occurring within the cavity is 0.67 ± 0.26 nM (0.46 ± 0.25 nM), corresponding to overturning driven lateral dFe flux estimates of 54.3 ± 24.5 (37.3 ± 21.8) Mg/yr.

Neither of these approaches is free from caveats. The extrapolation of regional data to calculate an integrated dFe zero salinity endmember assumes that dFe distribution is largely driven by one dominant freshwater source which is clearly incorrect as evidenced by high shelf concentrations in the ECG likely originating from the Transpolar Drift[72] (Fig. 5). The scaling of lateral flux to overturning volume assumes ligand properties in inflowing AIW are constant. The variation in ligand concentration across the region was modest (mean 2.1 ± 0.4 nM eq. Fe, range: 1.0–3.6 nM eq. Fe)[64]. Yet given that the concentration and binding strength of ligands along the AIW inflow from S8–10 during August 2016 appeared to be distinct from those in Fram Strait, and were likely affected by an earlier phytoplankton bloom on the shelf[64], there may be seasonal variation in ligand concentrations. Further refinement of these fluxes is not possible without better constraints on seasonality in the cycling of Fe and ligands. To the best of our knowledge, no such data is available from any geographical location close to ice shelves. Whilst logistically challenging, spring and summertime data from ice stations coupling trace metal, ligand and inert tracer (for example, Noble gases or $^{228}$Ra/$^{226}$Ra) would facilitate increased knowledge on the processes controlling lateral Fe transport with reduced uncertainties[73,74].

Whilst we cannot further reduce the uncertainty on lateral dFe transport at present, a comparison to findings from other ice shelf regions suggests that the overturning-driven flux, between 37.3 ± 21.8 and 54.3 ± 24.5 Mg/yr dFe, is more realistic than a flux derived from freshwater endmembers. It is also considerably larger than fluxes derived from freshwater discharge, inclusive of estuarine losses (3.5 ± 2.2 or 8.1 ± 3.5 Mg/yr, see above). Surveys near the Dotson Ice Shelf and the Pine Island Glacier similarly suggested benthic shelf derived dFe, rather than direct glacial meltwater input was the dominant source of dFe input into outflowing glacially modified waters[62,75]. This is corroborated by the most recent model studies of the Antarctic continental shelf suggesting that only 2–23% of dFe in glacially modified surface water originates directly from ice shelf meltwater[76] with the majority sourced from sediments and delivered to the upper water column via cavity overturning[77]. The rate of overturning is therefore likely the major factor controlling lateral fluxes.

Present estimates of Greenland Ice Sheet-to-ocean Fe fluxes are universally derived from measured freshwater concentrations and discharge volumes. A variety of approximations have been made to account for loss of dFe across the salinity gradient downstream of glacier outflows. If these are standardized at 10% of the

freshwater concentration, flux estimates for the Greenland Ice Sheet to date range from 0.07-4.91 Gg/yr[9,18,59,78]. The literature dFe concentrations for comparison to the GAM (3.13 ± 0.96 nM) and regression derived values (7.34 ± 0.70 nM) herein would range from 5 to 370 nM. It should be noted however that the extent to which dFe concentrations change along freshwater water courses and during mixing with saline waters is poorly constrained. The extent of dFe removal on timescales of hours to days may vary both between catchments and temporally with estimates ranging from 76% to 99% dFe loss[16,21,43]. Fluxes derived only from measured freshwater concentrations are thereby subject to uncertainty of at least an order of magnitude. Glacial dFe endmembers designed to be conservative in recent model studies for example range from 90 to 3000 nM[13,14], compared to 3–7 nM deduced herein at the point of outflow from an ice shelf cavity (Supplementary Tables 3–5) and do not directly reflect the notion supported by recent observations that most of the outflow from ice shelves is derived from sediment interaction rather than from freshwater outflow[75,77].

Whilst cavities created by floating ice tongues or shelves can only be found in northern Greenland and Antarctica, the basic two-dimensional description of glacier fjords as consisting of a deep saline inflow and a glacially modified near surface outflow is common to most large Greenland systems[79]. An open research question is therefore whether or not lateral export of trace metals such as dFe can be estimated from freshwater concentrations and discharge volume in these systems. For large marine-terminating glaciers where ice melt provides the largest fraction of freshwater to the cavity (~89% for 79NG)[28], estimates on freshwater nutrient inputs derived from surface runoff endmembers (e.g., refs. [9,59]) are likely overestimating glacial dFe shelf supply, also because ice melt contains comparatively sparse amounts of dFe, with concentrations presumably more akin to iceberg values[80]. An alternative hypothesis, which is supported by work in a range of catchments beyond 79NG, is that ligand dynamics and over-turning—which are interrelated—constrain lateral export[61,64,71]. In systems like 79NG, where shelf forcing rather than freshwater discharge controls the residence time of water masses within the glacier fjord[28,81], lateral dFe flux may be completely insensitive to short-term changes in freshwater and sedimentary dFe or LpFe supply (although is still ultimately dependent on a supply of labile Fe to maintain a large LpFe pool within the subglacial cavity) and far more sensitive to changes in ligand properties of AIW and cavity inflow rate.

**How representative is one process study at the 79NG to constrain nutrient export?** Freshwater discharge at glacier calving fronts is highly variable on both sub-daily and seasonal timescales[79,82,83]. Whilst herein we characterize mAIW and AIW water mass properties well enough to discuss the net changes induced by the 79NG system, only one ultraclean station was sampled at the 79NG glacier terminus (S1). This station (S1) was, however, located at the center of the sole cavity inflow depression (~480 m deep and ~2 km wide)[28,32] and thus represents the only pathway via which AIW interacts with the 79NG[28,29]. Furthermore, the seasonal variability in freshwater export from 79NG is less pronounced than for most of Greenland as the main fraction is derived from basal melt rather than subglacial runoff[28]. The highly dynamic nature of meltwater discharge and sediment re-suspension nevertheless imply that it is highly unlikely nutrient concentrations determined at S1 represent either mean or peak summertime values. Small changes in the location of this station in time or space along the ice front would likely have revealed high heterogeneity in turbidity and nutrient concentrations as was observed vertically at S1 and S2 (Figs. 3 and 5) for LpFe, dFe

and Si(OH)$_4$. However, the prolonged residence time of AIW within the 79NG ice cavity combined with the absence of phytoplankton blooms in this water mass as it exits the cavity in summer is advantageous for deriving the integrated modification of AIW as a result of all glacier-associated processes. Furthermore, the chemical signatures in mAIW denote a strong control of sedimentary processes in modulating cavity-exiting waters, suggesting limited influence of seasonal shifts in meltwater properties.

Whilst few dFe concentrations are available in similar localities worldwide, dFe concentrations reported herein are similar to those observed in outflow from the Dotson, Crosson and Getz Ice Shelf cavities (~0.7 nM)[62,75] and in the outflow from beneath the Pine Island Glacier (0.2–1.4 nM)[62]. Interestingly, dFe enrichment in subglacial discharge downstream to the Ross Ice Shelf was within uncertainty relative to shelf stations not affected by ice sheet discharge[84]. A comparison with the Pine Island Glacier system is particularly insightful as there, similarly to 79NG, ligand dynamics were also found to constrain lateral dFe transfer[71]. This is despite lower dFe concentrations in the corresponding shelf stations (<0.6 nM) and less pronounced changes in salinity (range 33.1–34.8 along a comparable 250 km long transect)[62].

We suggest a key reason for similarities in dFe concentrations between these diverse field sites is the prolonged residence time of water in ice cavities that allows thermodynamic equilibrium to be approached between dFe, LpFe and ligand concentrations. Higher dFe concentrations on the order of 10–100 nM are invariably associated with runoff corresponding to much more recent time periods since freshwater entered the marine environment[9,21,43] and thus could be described as under kinetic control. In a few case studies the timeframe of lateral dFe export has been quantified. Surface glacially modified water transports dFe from Heard Island (Southern Ocean) to 100 km offshore over a timescale of 1 month, with much of the ~80% decline in dFe over this spatiotemporal attributed to biological uptake[85]. In contrast, AIW/mAIW enriched in dFe would still be present underneath the floating ice-tongue on this timescale[28,29] with the removal of dFe independent of near-surface primary production.

**Future perspectives.** It is not yet clear how shrinking ice-tongues will affect shelf biogeochemistry, neither specifically for the 79NG nor more generally from ice-shelves elsewhere. Past observations[28,33,86] and models concerning the future of the 79NG[87] both project thinning of the ice-tongue in coming decades in response to increased ocean heat supply. Indeed, the ice tongue is significantly smaller at the time of writing than it was in August 2016 following the breakup of Spalte Glacier in July 2020 (which previously branched off from the 79NG). For systems such as the 79NG where the vast majority of nutrient export does not presently reach the euphotic zone, a disintegrating ice-tongue could create a more direct connection between overturning driven nutrient fluxes and primary production.

Based on our observations, we expect long-term disintegration of the ice tongue and shrinking of the association subglacial cavity to lead to higher dFe concentrations at stations close to the glacier terminus, potentially with a lighter $\delta^{56}$Fe signal and a higher $^{228}$Ra:$^{226}$Ra ratio. Similarly, disintegration may result in the more rapid downstream transport of (sub-)glacial modified waters, but will also likely affect interannual sea-ice dynamics[88] and thus the anticipated change to regional primary production and nutrient dynamics is more challenging to predict. Given that the 79NG ice tongue appears to be primed for further ice thinning and retreat, it however remains an interesting case study to investigate the marine biogeochemical responses to prolonged glacier retreat in coming decades.

## Methods

**Sampling and nutrient analyses**. Vertical, full depth profiles of salinity, temperature, pressure and light attenuation (turbidity) were conducted at high-resolution ($m^{-1}$) using a SEA-BIRD SBE 911 ultraclean CTD rosette (ucCTD) at all stations where trace elements were collected. For water column physical properties (salinity, temperature, pressure, turbidity, UV-light fluorescence) and macronutrient distributions ($NO_2$, $NO_3$, $PO_4$, and $Si(OH)_4$), the dataset from the ucCTD was combined with the largeCTD data (SEA-BIRD SBE 911plus) from the same campaign.

Trace metal samples were collected using the ucCTD rosette, equipped with 24 ×12 L GoFlo bottles following GEOTRACES sampling protocols[89]. LDPE bottles were pre-cleaned with sequential leaches in 2% detergent (mucasol), 1.2 M HCl and 1.2 M $HNO_3$ with ultra pure water rinses (>18.2 MΩ cm; Milli-Q, Millipore) between each stage. On deck, GoFlo bottles were then transferred into an overpressured clean room where water samples were filtered (Acropak 0.8/0.2 μm) into pre-cleaned LDPE (low density polyethylene) 125 mL bottles and acidified to pH 1.9 with 180 μL HCl (UpA, ROMIL) to determine dissolved trace metals. A further filtered (0.8/0.2 μm) sample was retained, passed through a 0.02 μm Anodisc peristaltic filtration unit within 4 h of sample collection, and then acidified to determine the sFe fraction (<0.02 μm)[90]. An unfiltered water sample was retained and acidified as above to determine total dissolvable trace metals after analysis >6 months later.

Particulate trace metal samples were collected onto pre-acid leached Polyethersulfone (PES) Membrane Filters (0.2 μm, Sartorius) through pressurization (0.2 atm) from each GoFlo bottle with filtered Nitrogen gas. 1.2–4.1 liters of seawater were passed through individual sample filters which were then sealed in clean petri-dishes, stored in a deep freezer immediately after collection at −20 °C and kept frozen until analysis[91]. The determination of the labile particulate trace metals (i.e. LpFe) was conducted by applying a weak acid leach (25% acetic acid, Optima grade, Fisher Scientific) with a mild reducing agent (0.02 M hydroxylamine hydrochloride, Sigma TM grade) and a short heating step (10 min, 90–95 °C) with a total leach time of 2 h[92].

Seawater for macronutrient analyses of $NO_2$, $NO_3$, $PO_4$ and $Si(OH)_4$ was also retained from each GoFlo rosette bottle. Note that while we refer to $Si(OH)_4$ and $PO_4$, as is conventional, it may also incorporate other species of dissolved silicon and phosphorus. Unfiltered surface macronutrient samples (upper 200 m) were stored at 5 °C and run within 18 h using the QUAATRO autoanalyzer[93] modified according to the methods provided by the manufacturer (Seal, Alliance). At all other depths, filtered samples (0.2 μm) were frozen and run at AWI using the same procedure.

Trace metal concentrations (i.e. dFe, sFe and TdFe) were measured via ICP-MS after pre-concentration exactly as per ref. [94]. Briefly, 15 mL sample aliquots were pre-concentrated using an automated SeaFAST system (SC-4 DX SeaFAST pico; ESI). All reagents for SeaFAST were prepared in MQ-Water (>18.2 MΩ/cm; Milli-Q, Millipore). Single-distilled sub-boiled $HNO_3$ (SpA grade, Romil) was used for sample elution. Ammonium acetate buffer (pH 8.5) was prepared from glacial acetic acid and ammonium hydroxide (both Optima, Fisher Scientific). The 10-fold pre-concentrated samples were analyzed by high-resolution inductively coupled plasma-mass spectrometry (HR-ICP-MS; Thermo Fisher Element XR). Calibration was via isotope dilution for Fe, and standard addition for Mn and Co. For labile particulate analyses, samples were measured via ICP-MS without pre-concentration and were quantified using external calibration prepared in the same sample matrix using multi-element standards (Inorganic Ventures). For details of quality control, see Supplementary Material.

Stable Fe isotopic composition ($\delta^{56}Fe$) was measured in seawater samples at the University of South Florida (USF) following a modification of ref. [95]. Filtered trace metal samples were combined from several depths (see Supplementary Table 9) to provide sufficient dFe for isotope analysis (10–20 ng). Fe was extracted from acidified seawater samples using a two-stage resin extraction method (Nobias PA-1 at pH 2, and AGMP-1), followed by analysis by Thermo Neptune Plus multi-collector ICP-MS with introduction via an ESI Apex Ω desolvator (with Ar but not $N_2$ added gas), Pt Jet and Al 'x-type' sample cones. The procedural blank using this method at USF has been established at $0.24 \pm 0.04$ ng Fe (1 SD, $n = 10$), similar to previous studies[95]. Instrumental mass bias was accounted for using an $^{57}Fe$-$^{58}Fe$ double spike, with $\delta^{56}Fe$ expressed in standard delta notation relative to the IRMM-014 standard. For this method we assess long-term instrumental precision using repeat analyses of the NIST-3126 Fe standard, for which we obtain a $\delta^{56}Fe$ value of $+0.36 \pm 0.04$‰ (2 SD, $n = 351$ during 24 sessions over 3 years). We apply this value as an estimate of 2σ uncertainty on $\delta^{56}Fe$ for all samples, except for samples where the larger 2 standard internal error is a more conservative estimate of uncertainty, as applies to samples in this study.

Stable oxygen isotopes ($\delta^{18}O$) were analyzed following the procedure described in ref. [96]. In short, seawater samples collected with the largeCTD were equilibrated for 6.5 h using carbon dioxide gas of known isotopic composition and platinum as catalyst. The equilibrated $CO_2$ gas was transferred into a Finnigan MAT Delta-S mass spectrometer equipped with two equilibration units, where the oxygen isotopic composition was analyzed a total of eight times. The oxygen isotopic composition is reported relative to a V-SMOW standard.

Natural radium isotopes ($^{226}Ra$, $^{228}Ra$) were measured via gamma spectrometry[97] following the procedure as per ref. [48]. Briefly, water column profiles were obtained using in-situ pumps including 0.8 μm Supor filtration and Ra

adsorption in $MnO_2$-coated cartridges. At 10, 50, and 350 m depth, two cartridges were used in series, allowing the calculation of Ra absorption efficiency. At the other depths only one cartridge was mounted, assuming the mean collection efficiency of all cartridge pairs (93 ± 8%). The cartridges were leached by Soxhlet extraction with 6 N HCl refluxing over 10 h, followed by Ra-coprecipitation with $BaSO_4$ as described in ref. [89].

## Data availability
The data that support the findings of this study are available from Pangaea oceanographic repository and the following links: PS100 (GN05) Physical Oceanography data can be obtained from: https://doi.pangaea.de/10.1594/PANGAEA.871030 (ucCTD), and https://doi.pangaea.de/10.1594/PANGAEA.871028 (largeCTD). PS100 (GN05) Macronutrient data can be obtained from: https://doi.pangaea.de/10.1594/PANGAEA.905347 (ucCTD), and https://doi.pangaea.de/10.1594/PANGAEA.879197 (largeCTD). PS100 (GN05) oxygen isotope data can be obtained from: https://doi.pangaea.de/10.1594/PANGAEA.927429 PS100 (GN05) ucCTD trace element data, and radium isotope data is provided in the Source Data File. Figure 1 was made by J.S. using MATLAB version 9.6.0 (R2019a, Natick, Massachusetts: The MathWorks Inc., 2019), Figs. 3, 4 and 5 were made by S.K. with Ocean Data View, version 5.3.0 (Schlitzer, R., Ocean Data View, https://odv.awi.de, 2020), Figs. 6 and 7 were plotted by J.H. using package ggplot2 for R (Wickham, H. 2009, Elegant Graphics for Data Analysis. Springer, New York, New York) and all other Figs. (8 and 9) were made by S.K. using Origin(Pro), version 9.1.0. (OriginLab Corporation, Northampton, MA, USA). Source data are provided with this paper.

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

## Acknowledgements

The authors thank Captain Schwarze and the crew of the RV Polarstern (GN05 cruise); chief scientist Torsten Kanzow (AWI); Nicola Herzberg and Jaw Chuen Yong (GEO-MAR) for assistance with sample collection; Takamasa Tsubouchi (AWI), Eike Köhn (GEOMAR) and Nat Wilson (Woods Hole Oceanographic Institution) for CTD handling; Gerd Rohardt (AWI) for CTD data processing; Christian Schlosser and Shao-Min Chen (GEOMAR) for assistance during trace element analyses; Martin Graeve and Kai-Uwe Ludwichowski (AWI) for macronutrient analysis; and Hanno Meyer (AWI) for the analyses of oxygen isotopes. S.K. was financed by GEOMAR and the German Research Foundation (DFG award number AC 217/1-1 to E.A.). M.H. received support from the DFG (award number HO 6321/1-1) and the GLACE project, organized by the Swiss Polar Institute and supported by the Swiss Polar Foundation. J.S. acknowledges support from the German Federal Ministry of Education and Research (BMBF) within the GROCE project (grant 03F0778A).

## Author contributions

S.K., M.H. and E.A. conceived the study. Sampling was conducted by S.K., J.S., MRvdL and P.L., trace metals were analyzed by S.K., A.H. and T.S., Ra-isotope analyses were performed by MRvdL., Fe-isotopes analyses were conducted by T.C. and B.S., and Fe-ligands were analyzed by I.A. Statistical analysis was conducted by J.H. and S.K. The manuscript was written by S.K. and M.H. with all authors contributing to its revision.

## Funding

## Competing interests

The authors declare no competing interests.
