## [Peer Review File · Nature Communications]

REVIEWER COMMENTS

Reviewer #1 (Remarks to the Author):

General Comments:

This was a really interesting study, with a novel set of samples that was well-written. The paper reported primarily iron, as well as macronutrient, data collected from a GEOTRACES cruise to the region encompassing a large tidewater glacier in NE Greenland. Specifically the authors were able to collect samples from the subglacial cavity out to beyond the shelf break. These samples are of a novel enough nature to be of interest to the community and the wider field I believe, though the conclusions themselves are not unexpected given previously published work on this topic by some of the authors on this submission. The conclusions regarding the Fe results and processes are more convincing than those for Si, because the authors only have dissolved Si results and thus must conjecture regarding particulate Si concentrations and processes. The paper also attempts to make the argument that the field should move away from detailed characterization of (glacial) freshwater towards instead using water mass enrichment in near-glacial areas for estimates of the impact of glacial meltwater on marine ecosystem productivity. While I agree, there is much value in this approach, to do so exclusively also is at determinant of understanding the dynamic on-ice processes – which are particularly susceptible to climate change. Also, as the authors note, this particular fjord system is novel for Greenland and more akin to systems in Antarctica and thus, the authors's argument to move away from characterising Greenland freshwater concentrations in favour of a mass enrichment approach is not as applicable/feasible. All that being said, this is a novel, well-conceived, well-executed, and well-written study that I believe will influence thinking in the field – and push us to consider full ice-to-ocean studies in the future where the freshwater concentrations can be directly compared to the water mass enrichment.

Specific Line Comments:

Abstract:

L22-23: What does was "equivalent to freshwater concentrations of" mean? Sentence is confusing as written – are those the concentrations measured here or freshwater concentrations being given as a comparison?

Introduction:

L36-38: Is this true? If it's annual supply of Fe and Si from icebergs it wouldn't make sense to scale with discharge volume?

L65: Can you describe these subglacial cavities a bit more? Is there a possibility for prolonged rock:water interaction during the half a year to 2 years that the water is stored there prior to release at the terminus? How similar are such cavities prior to ice shelves in Antarctica vs marine-terminating glaciers in Greenland?

L81-82: Are you speaking about Fe, Si, or both here?

L82-85: Move sentences about LpFe to be with previous sentences regarding LpFe, versus talking about LpFe, dFe, Si, and then back to LpFe.

L95: When you say "relatively well-constrained" are you referring to the 162-d estimate stated above in Line 66? It would be helpful to put in ()'s what you mean by relatively well-constrained given that you reference another paper in addition to the one referenced above in Line 66-67. Do they both come up with an estimate of 162-d? For what time of year is this estimate, and how would the storage time vary with seasonal glacier hydrological evolution?

Table 1: very useful to have this! But there seems to be some overlap between dFe and LFe based on size. Is LFe also the filtrate? Maybe provide more details so the reader can see how these are processed differently?

Figure 1: L129: S11 isn't marked on the map? Would also be good to label scale bar, and show an inset map of the broader region (i.e. showing all of Greenland and where this study site is specifically).

Methods:

L113: I understand that the cavity isn't the focus of this study, but it seems like residence time in the cavity and any prolonged water:rock interaction might be key to the crustal element fluxes measured downstream.. so a few more details so that the reader doesn't have to go and reference the other articles might be helpful.. e.g. mean 162-d residence over what portion of the meltseason, and how might this evolve seasonally? What is the physical nature of this cavity?

L122: How did you get this ratio?

Figure 2: Grey dots are VERY hard to see.. can you make bigger / clearer? Can you label PSW, mAIW, and AIW on the figure itself? Grey lines indicate, trace metal samples, but what depths were samples collected at? Can you summarize somewhere each of the depths were samples were collected? The resolution of the figure also appears to be poor and fuzzy. Can you improve upon this? It just makes it very hard to see the data.

Results & Discussion:

L143: Can you state in ()'s the enrichment range observed for Si and the depths that the Si was enriched at? It's very difficult to see on the figure. When you say "the glacier front" you are referring to 79NG vs its neighbouring glacier (mentioned on L120) correct? What is the grounding line of 79NG and the depth that the subglacial discharge comes out at? I don't believe I saw this mentioned in the methods? Are these elevated Si concentrations in line with this depth of release? In Fig3, it would be nice to see the turbidity data so the reader can confirm that the higher Si observed is in the plume and associated with a glacier source. And I assume you are referring to dSi here but perhaps good to clarify as you mention labile particulate Si in your introduction.

L148: The phrase "the dominance of AIW as a nutrient source to the region" seems inconsistent with the statements on L144 that the highest Si concentrations observed were from the glacier. The linear regression for the Si data in Fig 4 is also the weakest R2 (only 0.50). Which stations are being shown in Fig 4? It seems like on Fig 4 there is a clump of higher concentration higher S water (so marine source) but there are still some high(ish) concentration, lower S water .. which doesn't indicate AIW as the nutrient source.

Figure 4: Which stations are being shown here? Would be good state so we can see if it's just what is in Fig 3 or additional stations? Doesn't seem to match up with Fig. 3 - i.e in figure 3 for Si there are concentrations up to 20 uM which aren't reflected in Fig 4?

L150: The phrase "In contrast to macronutrient distributions, positive Fe anomalies were observed" isn't quite true for all your macronutrients - i.e. Si - as positive anomalies can also be seen for Si in Fig 3c.

L151: reference to Fig 3d - can you show turbidity on Fig 3d too so the reader can see for themselves the max LpFe coinciding with max turbidity vs putting that info in the supplemental?

L152: It seems as though the max in LpFe and TdFe is occurring when attenuation is lower according to Fig S2?

L177: Well, it seems like in Fig 5 at S1 , there are definitely higher concentrations down the water

column compared to the other stations so I'm not sure it's quite accurate to say that the distribution of dFe was less dominated by the near-glacier stations.

L178: state in ()'s again the depths of the mAIW outflow please.

L218: These extrapolated values aren't shown on Fig. S4? It's a bit unclear how from the linear regression shown of salinity and dFe the authors came up with an extrapolated mean FW end-member of 7nm.

L228-229: I agree there is no clear positive linear relationship but it would be nice to see the R2 values as you have shown / stated in the text for the other correlations. Also this is (I believe) the first mention of anomalous data from S4, which isn't shown in the other section plots .. why were these anomalous - and what were they? - and why were they excluded?

Figure 6: seems to come before Fig 7 but Fig 7 is referenced first? (L229)

L242: It would be useful to see the data from S4 somewhere ..

L257: It's unclear how the FW endmember value of 23 uM was calculated? Would this just be for dissolved Si?

L265-267: It's unclear to me if this statement is just based on the data presented here which is all dissolved Si? I don't see any data presented on labile particulate Si even though it's mentioned in the introduction. Without that data, it's difficult to compare the 23 uM endmember to the total Si export proposed for runoff or basal ice - since both of those estimates include particulate data (L264), which this study does not have, correct?

L284: I'm curious as to how the authors came up with the value of 1.3 nm for the "maximum cap" on the dFe.. was that the average value? Because certainly in Fig 7 (both A and B) there are many points with values above this 1.3 nm cap.

L285: The statement the "overall asymptoting dFe/LpFe relationship" refers to what figure specifically? Where is this relationship shown? It's difficult to assess the validity of this statement as written and with the data shown.

L336: "All glacier-associated processes" that occur in the cavity prior to release in the ocean.

L365: Is the "removal of dFe independent of near-surface primary productivity" because S1 is in the cavity which is a tunnel and thus in the dark? I'm still having a difficult time picturing this cavity so more descriptions of this earlier / a picture would be helpful, as the procurement of samples from this field site is definitely the novel aspect of this study. It would be nice to see PAR data to back-up this statement that S1 is independent of primary production activity. However, from Fig. S1D it seems like there is still primary production at S1?

L420-433: This argument assumes that the processes occurring in the cavity are the same as those that might be occurring in subglacial channels.. this may not be necessarily be true (e.g. flowing vs stagnant water for example). Thus, by only using the residence time of water in the cavity and concentration values from other studies conducted at different (non-floating ice-tongue systems) it seems (at least to me) unconvincing. And as the authors point out subsequently, this cavity they were able to sample - though novel - is not really representative of Greenland tidewater glaciers.

L444-445: But as the authors themselves point out above, glacial meltwater inputs vary on daily to seasonal timescales, as well as spatially depending on bedrock geology, upstream origin (e.g. supraglacial melt, basal ice melt) and processing of waters (e.g. subglacial flow paths, proglacial lake

storage).. thus, there is value in detailed characterization of glacial meltwater concentrations and processes in order to understand the mechanistic basis of this variability – especially because it is not always feasible to sample within a cavity directly at the calving ice front.

Reviewer #2 (Remarks to the Author):

The aim of this paper is to investigate and quantify the role of ice melt on nutrient concentrations in seawater off NE Greenland. Whilst this is only a temporal snapshot of one location, Nioghalvfjærdsbrae, where ice melt flows into the ocean via an subglacial cavity, the dataset is unique and adds to the overall picture of biogeochemical cycling in high-latitude settings. The data are of high quality and the manuscript is generally well-written and clearly structured: I enjoyed reading it. This subject will be of broad interest, and the manuscript should be suitable for publication in Nature Communications after some revisions. There are a few places that are a little opaque, leading to some apparent inconsistencies, which mean that some of the conclusions are not as strong as the authors suggest. As such, I have a few recommendations, clarifications and corrections, which I hope will be constructive.

In addition to the unique setting, there are some important key points that I've taken away from this paper. Firstly, dFe and dSi are being added to the ocean from glacial meltwater. Secondly, that there is a decoupling between direct meltwater inputs and nutrient release into the oceans due to residence time of water in the subglacial cavity. The influence of residence times is poorly constrained in general for fjord and estuary systems, and important to consider when attempting to quantify nutrient fluxes. Thirdly, a more subtle message is that there is a decoupling - as a result of dynamic interactions - between particulate and dissolved phases. Is meltwater a good tracer for behaviour of particulates, and how tightly coupled are particulates and their dissolved counterparts? What are the role of colloidal fractions and ligands? All of these are pertinent questions.

My main issues relate to how these points become a little lost towards the end of the manuscript, and need to be clarified so that the key findings of the study can be emphasised correctly and robustly.

How endmember values for the integrated freshwater input are calculated:

It appears that the authors use two different methods for this calculation for dFe and dSi: they use a linear regression between salinity and dFe to get at a dFe endmember (line 219), and use a dilution factor to calculate the dSi endmember. Apologies if I've missed it, but is there a reason for this? I'm guessing this is because the linear regression method results in a negative endmember value for dSi (as discussed below). (As an aside, I would be interested to know what endmember value would be obtained for dFe by using the dilution factor method used for dSi.)

There are some important issues that arise from these endmember calculations.

Iron:

The linear regression method, used for dFe, assumes a relationship between dFe and salinity that might not be expected given the complex array of processes occurring on the shelf (i.e. the decoupling between dFe and salinity due to phase transitions, sedimentation etc.). Given that the particulate phases are sinking (line 161) and that there is a fairly strong benthic input of dFe (line 185, Figure 5), this benthic supply of dFe (and in fact any dFe cycled in the water column) would not necessarily be

associated with low salinity waters. In fact, looking at Figure S4, there is a considerable amount of scatter that might well point towards such decoupling. I would at least like to see some statistics to back up the use of a linear model in this case (Figure S4, line 219). What is the uncertainty on this integrated freshwater endmember value? Without having the salinity values available I was unable to do the quick calculation, but I would suggest that there is a large range in the intercept value on the y axis of Figure S4. It is important to include those uncertainties in the manuscript.

This relates to a further complication with the discussion surrounding the lack of correlation between dFe and LpFe (line 234). The authors claim that this challenges “the widely used assumption in flux calculations that LpFe is a universal source of dFe and conversely may reflect the role of particles instead as a sink of dFe across the studied region”. However, I would suggest that all this observation could also just highlight the decoupling in space between dFe and LpFe as a result of the complexities associated with sinking and sedimentation. I would not expect a clear correlation between dFe and LpFe in samples taken from the same niskin bottle, partly because of all the challenges involved in obtaining reproducible particulate data from niskin bottles as a result of particle settling, and partly because of particle dynamics in the water column. The complexities of dealing with such a dynamic system are touched upon in the data (e.g. caption of Figure 7, and the exclusion of the datapoints from the region of strong surface mixing), but I feel is not fully appreciated.

Silicon:

Firstly, I think it's important to note that it's challenging to calculate a robust silicon budget without having amorphous silica (ASi) and biogenic silica (BSi) data. A lot of the conclusions are based on extrapolations from other – quite different - regions of Greenland. There are no data about how much of the dSi is taken up biologically. I'm surprised that these data are not available, given that this study was part of a GEOTRACES program (also see below). The fluorescence data are shown in the supplementary information, but it's not possible to see how much of this is diatom biomass. Were there any cell counts, HPLC or Chl a, b and c extractions from the bottle samples?

Secondly, I'm not clear how the freshwater endmember calculations were carried out. I've assumed that they're based on the 1:84 dilution mentioned on line 122. If this is the case then I agree that the difference between mAIW and AIW of 7.73 and 7.44 μM does give an endmember of approximately 24 μM . However, the authors state on line 256 that the 7.73 μM value for mAIW has an uncertainty of 3.7 mM, which means that there is a large uncertainty on their endmember composition value, which is not appreciated in the text. The lower estimated value would indicate that mAIW is diluted with respect to dSi, whereas the upper estimated value would result in an extreme endmember composition of approximately 335 μM (and this doesn't include any remaining particulates). Regardless, to make the strong statements (e.g. on line 263 onwards, and highlighted in the abstract) based on these calculations without a proper appreciation of the associated uncertainties is somewhat misleading.

In brief, the main conclusion from this section of the manuscript is that the meltwater from the subglacial cavity is a source of dSi to the mAIW (although I would question the magnitude of the input given in the manuscript without having the associated uncertainty stated). However, later on in the manuscript, I then feel that the authors contradict themselves, by saying the AIW is the major input of dSi because of the negative freshwater endmember from the linear regression method. I would have thought that this demonstrates that this simple linear regression model is not a robust way of judging the integrated freshwater input (for the same reasons as discussed for dFe, especially in the case for Si as there are no particulate data). Note also that there is no uncertainty given on this intercept value (-16 mM, line 403).

I'm also a little confused by the calculations that follow on from line 420 onwards. I assume that the calculation of a subglacial dSi concentration of 210 μM (line 422) assumes that subglacial discharge is 11% of the freshwater budget, which has an integrated concentration of 23 μM . This then would be

ignoring any input from the basal melt, and any amorphous silica remaining in solution, neither of which are mentioned as caveats. Lastly, there is also no appreciation of the impact on these calculations of the uncertainty on the integrated $23\mu\text{M}$ value – if this is towards the upper end of the estimate ($335\ \mu\text{M}$, see above) then the subglacial concentration would be instead over $3000\ \mu\text{M}$. I'm afraid I don't understand where the 100 and $14\ \mu\text{M}$ values come from on line 426, and I begin to lose the thread of the argument. Regardless, a final upper estimate of 28% dissolution is not unreasonable and does not necessarily contradict the findings in Hawkings et al., 2017. In this paper, Hawkings et al. do not suggest full dissolution over timescales shorter than 162 days; their experimental results (their Figure 5) would point towards 30-50% dissolution over this timespan (and these experiments were carried out using seawater with a salinity of 35.5, so dissolution might follow a different trajectory than fresher waters).

In summary, I find the endmember calculations for the integrated dFe and dSi a little opaque - I'm not really sure to what extent they question the validity (line 386) of previous estimates. I think the approaches themselves are fine, but the caveats and uncertainties involved are not discussed sufficiently. My recommendation would be to apply both methods to both elements systematically, and discuss the challenges involved. This should help to avoid the apparent contradictions and confusions.

General Minor comments:

Please refer to dSi rather than just Si throughout (including figures), to make it clear that you are only referring to the dissolved phase.

There are a number of datasets that I would add considerably to the paper that are not present. If they do exist - and I expect they will given that this study is based on a GEOTRACES cruise - then I would suggest adding them in. 1) I would include reference to the actual ligand data in Ardiningsih et al., 2020; 2) I would add any measurements of biological productivity or nutrient drawdown (discussed on line 229, but no values given other than fluorescence in the supplementary information) calculated from the cruise (and/or from satellite data); 3) I would include meltwater fraction calculations based on oxygen isotopes (also allowing quantification of any inputs from sea-ice melt/brine formation c.f Tonnard et al., 2020); 4) I would include any existing radiogenic isotope data to constrain process rates.

Minor comments and suggestions:

Could the authors please mark on Figure 1 the line of the section shown in Figure 2?

Line 186 – The authors should not necessarily capturing benthic inputs in niskin bottles fired at 2-5 m above the seafloor. Whilst the discharge from the subglacial cavity might represent a combination of pelagic and benthic processes (line 258) there will be benthic processes on the shelf impacting these nutrient cycles. Quantification of these processes would need measurements of porewater profiles, core incubations, benthic landers, and/or a radiogenic isotope data (e.g. U-series).

Line 191 - The "pronounced surface peak" (Figure 5a) at station S7 is interpreted as coming from the Transpolar drift (TPD) rather than a localised source. This is a reasonable hypothesis given that high dFe values have been found in the Central Arctic Ocean surface waters in regions of high meteoric water input (up to 5-6 nM, Charette et al., 2020). The authors could also add a comment about a new paper from the same group that is already cited (Ardiningsih et al., 2020), showing that TPD waters are enriched in Fe-binding ligands (explaining how dFe could remain in the surface waters so far downstream). However, there are a couple of caveats I would also add: 1) this peak is based on one bottle measurement; 2) without showing a transect of stations along the EGC path, it is not clear as to whether there is enrichment here due to meltwater inputs from Greenland or not. Also please note

that Tonnard et al., 2020 (cited on line 195), also interpret shelf waters off southern Greenland to be Fe fertilized from meteoric water inputs from Greenland:

(p.933) *"Surface waters at stations 53 and 61 were characterized by high MW [meltwater] fractions together with enrichments in PFe at station 53 and in both DFe and PFe at station 61... At station 61, the relative depletion of DFe at 30 m compared to 50 m may be due to phytoplankton uptake, as indicated by the high TChl a ... Hence, it seemed that meteoric water inputs from the Greenland Margin likely fertilized surface waters with DFe, enabling the phytoplankton bloom to subsist."*

Line 367 onwards - The authors should note that fluxes are very difficult to constrain in shelf environments without rate determining measurements, either from physical models or radiogenic isotopes. For the flux estimates that they have, it might be useful to convert fluxes to Tmol/year, so be consistent with other Si budget papers (e.g. Tréguer & De La Rocha, 2013).

Line 387 - I agree that this is not "informative of seasonal processes" - especially not considering critical processes including sea-ice formation/melt (see comment above on oxygen isotopes), mixing from internal waves, or storm-driven mixing. Perhaps add some of this detail.

References:

Ardiningsih, I., Krisch, S., Lodeiro, P., Reichart, G. J., Achterberg, E. P., Gledhill, M., ... & Gerringa, L. J. (2020). Natural Fe-binding organic ligands in Fram Strait and over the Northeast Greenland shelf. *Marine Chemistry*, 103815

Hawkings, J. R., Wadham, J. L., Benning, L. G., Hendry, K. R., Tranter, M., Tedstone, A., ... & Raiswell, R. (2017). Ice sheets as a missing source of silica to the polar oceans. *Nature communications*, 8(1), 1-10

Tonnard, M., Planquette, H., Bowie, A., Van Der Merwe, P., Gallinari, M., de Gésincourt, F. D., ... & Tréguer, P. (2020). Dissolved iron in the North Atlantic Ocean and Labrador Sea along the GEOVIDE section (GEOTRACES section GA01). *Biogeosciences*, 17(4), 917-943

Tréguer, P. J., & De La Rocha, C. L. (2013). The world ocean silica cycle. *Annual review of marine science*, 5, 477-501

Reviewer #3 (Remarks to the Author):

Krisch et al., present the results of a GEOTRACES transect and cruise from the 79NG Glacier in North Greenland into the Atlantic ocean focussing on the concentration and transformation of two important macronutrients, Fe and Si from within the subglacial cavity and subsequent export to the Atlantic ocean. The authors find enrichments of differing Fe phases do not occur in the same oceanic regions, and show for the first time that the distributions of the proportions of different iron phases within the filterable iron change with distance from the glacial terminus, and not uniform as previously thought in the literature to date, which makes this manuscript of high impact. The data are of excellent quality, especially regarding the difficulty of sampling and measuring soluble (sFe) to such low concentrations, and provide new data for colloidal and labile Fe (cFe and LpFe, respectively) concentrations which shift our current perspective on their proportional distributions within the water column and during (sub) glacial export. This 79NG glacier has the largest floating ice tongue in Greenland and data and results from this location are rare and highly valued. The manuscript is well written and set out, however I think the manuscript could greatly benefit from an annotated cartoon

of the subglacial ice cavity indicating the inferred Fe transformations, potentially based upon ref 32 fig 4 of the cavity, and the processes which determine Fe and Si availability and export.

The authors use a linear regression to calculate a freshwater endmember of Fe as 7nM, and subsequently use this to upscale the Fe flux from the Greenland ice sheet and other glaciers which discharge into the Atlantic (e.g. L367). There are no errors shown for this calculation or propagated errors, which should be shown as this value is pretty low compared to other insitu measured concentrations in other studies. The same linear regression is also applied to calculate a freshwater Si concentration but does not work, so the authors use another method. In Figure 6, the authors use a power law relationship to explain the removal of dFe, so I do not understand why this was not used in the calculation of the freshwater endmember, as the resultant Fe concentration will be much higher, and may then be more comparable to those actually measured in the literature (e.g. refs in L223). I also wonder what the Fe concentration would be if the authors used the same approach for Fe as they do for the final Si freshwater end member calculation.

I was hoping that the section starting L322 would convince me of the more global scope of the paper as the results are very intriguing, but in lines L323 to 339, L367 to 383 and L435 to 440) I find the reasons not to extrapolate more convincing than those for extrapolation of the data, and so I find that the impact of the paper is diminished. Local and regional extrapolations hold as they are well constrained within this paper, but not for all glaciers from Greenland or all those going into the Atlantic Ocean. There are also no references to the data used for upscaling or clarity as to what freshwater fluxes the data have been upscaled to. This section needs much more work and detail before it can be extrapolated to such an extent (more details in 'In-line comments').

I am also confused as to what phases were measured for Fe and what were not. Table 1 implies cFe was determined by the difference between dFe and sFe, whereas the methods in L481 suggest cFe was directly measured and sFe calculated. Please can the authors make this more explicit in the manuscript and methods as one of the conclusions rely on the result that the proportions of dFe and cFe greatly vary from the glacier front to the ocean, and it is vital to know if they were measured independently or not, to be sure of mass balance, and not a loss or gain of Fe to/from another phase.

I am also unsure how 'ligand-saturation' was determined and measured, L341, this is not in the methods and there is only a small mention that ligands can be organic or inorganic, nothing specific as to what they could be. This is very speculative yet a whole sections are dedicated to it (L268 to L321).

In-line comments:

L20: As written here, the abstract suggests that the residence time of 162 days has been measured in this study, it has not. Please reference.

L56 to L58: contradicts statement in previous paragraph about enhanced freshwater flux increasing nutrient input into the oceans. Suggests this will be diluting them.

L61: What downstream waters? As in the ocean proximal to the ice shelf? Or those closer to the exit of the ice-tongue? The open Ocean?

L73 to L75: This sentence is confusing as written as it suggests that LpFe dissolves to form dFe and cFe, but your definition in Table 1 is different to this, as cFe and sFe are part of dFe.

L82: I do not think that this sentence is true, a google scholar search will show that there are many studies which study Fe transformations at ice (glacier) seawater interfaces, for example...
Front. Mar. Sci., 14 June 2019; <https://doi.org/10.3389/fmars.2019.00332>
Biogeosciences, 15, 4973–4993, 2018; <https://doi.org/10.5194/bg-15-4973-2018>

L89: The reference to Table 1 after this sentence suggests that you have sampled all of these fractions, but I don't see LFe in the manuscript. Despite this the table is incredibly useful as it lays out precisely which phases are in which fractions and the references used.

L219: What is the error on this extrapolated value?

L221 to 223: Is it possible that Fe removal is not linear? It may be a power (like in Fig 6) or exponential removal which would, if fitted to data in Fig S4, increase your anticipated freshwater end member, and be more comparable to those actually measured in glacial fresh water (refs in L223), such as the exponential fit in Figure 6?

L238 to L239: What is meant by 'high particle loadings'? how does this change weather dFe is bound to organic-ligands or LpFe? Or are organic-ligands included in the 'increasing LpFe'? This then leads to my next comment in lines L239 to L244, where you suggest that they should be correlated, which then is the reverse of the observation in lines L232 to 233 (as written).

L257: 'Integrated freshwater endmember' so not a linear regression as for Fe in L221 to 223?

L258 to 268: Does this assumption also impact dFe and TdFe since active redox processes in the sediment may act to release or remove Fe to/from the water column and the sediment interface.

L275: Spatial, and seasonal?

L314: This is a tough claim to make, as definitions of the size fractions of colloidal Fe in publications really vary, from $< 0.45 \mu\text{m}$ and molecular weight $> 10 \text{ kD}$. You can certainly make the claim based on this paper, but not so much in comparison to some comparative literature. I think it is worth making that point here or referencing papers which use the same colloidal definitions as the one used here.

L336 to 339: I think I would argue the opposite as these signals can easily be monitored on a second to second time scale using many multi probes and tend to be much more logistically accessible for full seasonal and yearly studies. Or do you mean smaller ice tongues from fjord terminating glaciers? This sentence adds little to this manuscript.

L341 to 344: What about seasonal variations?

L368-369: Again, error bounds please for both the 7nM and 0.4Mg Fe yr. What references and values are used for upscaling here? Does this account for all ocean terminating glaciers, fjord terminating glaciers, all glaciers? What is the geographical extent of your upscaling and what freshwater fluxes are specifically used? Arguments to upscale need to be more explicit and convincing, especially how upscaling was achieved.

Additionally, not all these glaciers used in your model will have cavity residence times of 162 days - residence times are likely too dynamic between different glacial masses, but also annually, seasonally and depend on the thickness of ice, subglacial and supraglacial discharge flow, amount of sediment and distance between base and sea/fjord floor, local currents etc etc. e.g. L306 to 307 - Please expand on this calculation and explicitly state the assumptions.

L387 to 389: Again, the 79NG Glacier is notorious for being one of the fastest flowing glaciers, therefore is it representative of all glaciers flowing into the Arctic ocean? The calculations are also based on an extrapolation with extremely high error, so it would be good so see what the flux values are taking this into account.

L403 to 411: Again, problems using a linear extrapolation. What happens to freshwater dFe estimates if this modelling approach is used instead of a linear regression?

L468: What was the pressure used for filtration? Does this forcing impact the size fractions going through the filter?

Tables:

1: Very useful, please add on what was measured and what was calculated in this manuscript

Figures:

1: No S11 station on the figure

2: Could probably be joined with Figure 1.

3: Great, excellent figure.

4: I don't think this adds anything to the main text and could go in the supplementary material

5: Great figure, but for my personal preference I think Figure S2 where you almost see the covariation that is referred to section 'The role of colloidal Fe in glacier-to-ocean fluxes'.

6: Very useful, yet questions why you used a linear regression to find the fresh water composition of 7nM

7: put error bar key on the figure, so that there is a visual guide to actual variation. Some of the anomalous concentrations overlap with the accepted data, why were they deemed anomalous? The location?

8: Why are stations 1 and 2 put together as one data point? Please separate these as it looks like they are additive. Please add a 1:1 line so that a comparison to the distributions of cFe to sFe can easily be seen by the reader.

Supplementary material:

Table:

Station number missing in column A197 to 215

Row 3: what is std? the error? Standard error? Please indicate the error type

Response to reviewers

'The effect of ice cavities on ice sheet nutrient export: A case study at Nioghalvfjerdingsbrae, the 79°N Glacier' (Krisch et al.)

Reviewers are thanked for detailed and constructive comments on the text. We think the inputs by the reviewers have allowed the manuscript to improve considerably. To briefly summarise our major revisions before detailed comments:

A concept figure is added annotated with known features of the 79NG system, in addition to details of the moorings that AWI has deployed in the region to constrain water mass fluxes. It is evident in our prior text that the nature of the subglacial cavity and the details of how Atlantic Intermediate Water (AIW) and modified AIW (mAIW) fluxes in the region have previously been constrained was not well described.

We now introduce the data with both a geographical description of chemical distributions along transects upstream/downstream of the 79NG followed by a statistical description of the same data. In the supplement we explain the choice of definitions of water masses and concentrations in more depth, showing that our interpretations are not sensitive to changes in the definition of inflowing AIW or outflowing mAIW. This largely verifies our original interpretation of inflow/outflow dynamics, however please note that statistical treatment of silicic acid data shows no meaningful change in silicic acid concentrations across the region attributable to the 79NG and therefore we have slimmed much of our discussion concerning silicic acid distributions.

New Data: Several comments were made with respect to additional data that might be/is available from the cruise. The following parameters of interest have been analysed (other specific requests are not available): ^{18}O , Ne or He, ^{228}Ra and Fe stable isotopes. In brief:

- ^{18}O data is available, shows that there is a strong riverine signal across the EGC and verifies that high dFe (and ^{228}Ra) signals on the shelf are not locally sourced and instead come from further north. Similarly to Ne/He data, ^{18}O largely verifies the calculations of (Schaffer *et al.*, 2020), used to describe the physical circulation and freshwater content of mAIW.
- ^{228}Ra and ^{226}Ra data are available for a few stations at depth, including a profile at S1. This station shows, consistent with our trace metal data, that Ra in mAIW at 125 m is not of glacial origin, but originates from contact with sediments, where, based on previous studies, substantial dFe enrichment should have been observed.
- Isotopic data for dFe is most insightful as it verifies our original hypothesis that dFe exiting the cavity in mAIW is not a direct subglacial signal, but instead a signal from benthic exchange occurring over a sufficiently long period of time (months to years) that the isotopic signal of dFe emerging in mAIW is not glacial in origin. Similar to Ra and dFe speciation, the observed Fe isotopic signals at the glacier terminus imply that the associated cavity outflow flux cannot be scaled to freshwater discharge volume/concentration, but instead must be considered as a function of cavity overturning rate.
- Ne or He traces subglacial melting and discharge of mAIW, verifying that the physical calculations of (Schaffer *et al.*, 2017, 2020), are correct with respect to the depth distributions of mAIW and flux calculations out of the cavity across the shelf. The data is unfortunately not available for us prior to publication from the authors, but it does not show anything different to what can be obtained from ^{18}O or freshwater budgets, or from existing water mass analysis.

Reviewer comments are in black.

Responses to reviewers are in blue.

Reviewer 1

General Comments:

This was a really interesting study, with a novel set of samples that was well-written. The paper reported primarily iron, as well as macronutrient, data collected from a GEOTRACES cruise to the region encompassing a large tidewater glacier in NE Greenland. Specifically, the authors were able to collect samples from the subglacial cavity out to beyond the shelf break. These samples are of a novel enough nature to be of interest to the community and the wider field I believe, though the conclusions themselves are not unexpected given previously published work on this topic by some of the authors on this submission. The conclusions regarding the Fe results and processes are more convincing than those for Si, because the authors only have dissolved Si results and thus must conjecture regarding particulate Si concentrations and processes. The paper also attempts to make the argument that the field should move away from detailed characterization of (glacial) freshwater towards instead using water mass enrichment in near-glacial areas for estimates of the impact of glacial meltwater on marine ecosystem productivity. While I agree, there is much value in this approach, to do so exclusively also is at determinant of understanding the dynamic on-ice processes – which are particularly susceptible to climate change. Also, as the authors note, this particular fjord system is novel for Greenland and more akin to systems in Antarctica and thus, the authors' argument to move away from characterising Greenland freshwater concentrations in favour of a mass enrichment approach is not as applicable/feasible.

We made this comment on the preference for water mass enrichment, because uncertainty in how non-conservative components such as dFe and silicic acid behave across the salinity gradient accounts for practically all of the uncertainty on fluxes. Therefore, more freshwater data does not discernibly reduce this specific uncertainty, although we agree it is of course useful in other contexts. Note that freshwater discharge does not discernibly affect the turnover rate of this (or similar cavities), so there is no straightforward reason why outflow dynamics should scale with freshwater inflow (von Albedyll, 2018). Our new inclusion of Ra and Fe isotopic data further supports this hypothesis as it shows conclusively that the chemical signal of mAIW is sedimentary, not glacial meltwater, in origin and thus cannot be interpreted by considering the properties of runoff on annual timescales. A statistical approach shows that silicic acid behaves the same as PO₄ and NO₃, and there is no change in silicic acid availability discernibly associated with glacial processes or with particles (turbidity), although the reviewer is correct that we cannot really comment on why this is the case – our discussion of silicic acid dynamics is therefore slimmed.

The reason why 79NG is useful in this context is because we can clearly distinguish between processes that occur primarily as a result of mixing or inorganic effects, and processes that occur related to phytoplankton blooms in near-surface waters. This is not usually possible, we can make the same comment that a water mass enrichment approach would be insightful in any catchment, however there are more caveats in systems where glaciers feed directly into surface waters which host summertime phytoplankton blooms. As far as the peculiarities of the cavity are concerned, yes this does of course raise questions about whether the major processes driving change in the distribution of key elements at 79NG are the same as the major processes elsewhere, or whether there are fundamental differences. Statistics can provide some insight into this as there are two other case studies on the same coastline with similar AIW inflow dynamics, but no ice tongue and more “typical” glacier termini when we consider the 100s of large glaciers around Greenland (Cape *et al.*, 2019; Seifert *et al.*, 2019).

Both show similar distributions of macronutrients and the limited TdFe data at Sermilik is also comparable to that observed herein (Cape *et al.*, 2019). The underlying dataset for Scoresby Sund is not available, but for Sermilik it is and we see very similar features as per 79NG (e.g. the dominance of AIW inflow in silicic acid/PO₄/NO₃ distributions. The study quoted in fact adopts a similar rationale that only through an anomaly approach can the distributions be explained (Cape *et al.*, 2019).

All that being said, this is a novel, well-conceived, well-executed, and well-written study that I believe will influence thinking in the field – and push us to consider full ice-to-ocean studies in the future where the freshwater concentrations can be directly compared to the water mass enrichment.

This would indeed be nice, but as noted with different caveats. The concept of a ‘single’ endmember for either fresh or saline waters is not really applicable as the concentrations of most of these parameters are dynamic (e.g. see the dFe work in the Bayelva catchment which shows a prolific drop in dFe along both freshwater and saline systems (Zhang *et al.*, 2015)). The Ra and Fe isotopic data hopefully better demonstrates why freshwater concentrations are not necessarily a useful tool for understanding downstream enrichment/deficits, although of course it would be useful to have.

Specific Line Comments:

Abstract:

L22-23: What does was “equivalent to freshwater concentrations of” mean? Sentence is confusing as written – are those the concentrations measured here or freshwater concentrations being given as a comparison?

We have tried to explain better the concept on non-conservative losses in the supplement (page 7). In the text we simply derive the zero-salinity endmember rather than referring this in different ways (line 244-246 and 285-287).

Introduction:

L36-38: Is this true? If it’s annual supply of Fe and Si from icebergs it wouldn’t make sense to scale with discharge volume?

In terms of total mass (i.e. sediment), yes. In terms of dissolved non-conservative components, no. The stability of dFe in saline water is controlled entirely by physical factors

and organic Fe binding ligand dynamics. Thus, adding more inorganic Fe into a water mass does not increase dFe concentrations once thermodynamic equilibrium has been achieved unless you are adding more organic Fe binding ligands, changing the physical conditions (pH, O₂ etc) or if dFe is undersaturated to start with. Adding an increased surface area to an already saturated dFe solution would decrease dFe concentrations. This is well exemplified across a broad range of environments (Homoky *et al.*, 2012; Wuttig *et al.*, 2013; Ardiningsih *et al.*, 2020). For silicic acid, again there is not much specific literature on this, nor on how sediments scale with discharge, or even where it comes from: does the sediment content of ice stay constant as ice discharge decrease?

L65: Can you describe these subglacial cavities a bit more? Is there a possibility for prolonged rock:water interaction during the half a year to 2 years that the water is stored there prior to release at the terminus? How similar are such cavities prior to ice shelves in Antarctica vs marine-terminating glaciers in Greenland? Yes, we have added a concept figure as it is apparent that this was not described well (Figure 2). We cannot expand in much detail about what happens beneath the cavity, but we can make some deductions based on its size. It is well known that sediment release in subglacial discharge or ice melt is followed by settling out of this sediment close to glacier termini, so given the size of the cavity, yes it is practically certain that almost all sediment from the glacier accumulates underneath it (e.g. (Andresen *et al.*, 2012)). Hence why it is an interesting case study because the vast majority of glacier-derived sediment – water column interaction is occurring in the dark cavity, and by contrasting AIW inflow and mAIW outflow we can calculate the net effect of all this. Discharge around Greenland is generally higher and ice cavities smaller, thus the freshwater content of Greenlandic cavities and flux through such cavities will invariably be higher around Greenland compared with Antarctica.

L81-82: Are you speaking about Fe, Si, or both here?

We have clarified this (line 67-70):

“Given the non-conservative behavior of both Fe and Si(OH)₄ across salinity gradients, and the multiple processes affecting their exchange between the particle and dissolved phases, we hypothesize that these prolonged residence times may decouple the outflowing chemical enrichment from the ice sheet-derived inputs.”

L82-85: Move sentences about LpFe to be with previous sentences regarding LpFe, versus talking about LpFe, dFe, Si, and then back to LpFe.

Done.

L95: When you say “relatively well-constrained” are you referring to the 162-d estimate stated above in Line 66? It would be helpful to put in ()’s what you mean by relatively well-constrained given that you reference another paper in addition to the one referenced above in There are two studies we know of that have quantified cavity overturning timescales. (Wilson and Straneo, 2015) estimate a cavity exchange timescale between 110-320 days with the most probable value around ~215 days. We refer mainly to the (Schaffer *et al.*, 2020) work because it concerns the same year as the data referred to herein and was determined using 5 moorings across the shelf area. We can remove “relatively” as it would be, we think, correct to state it is the best constrained estimate for any comparable system. As this is relatively important for our flux discussions, we now mention these aspects in the introduction/methods. The new data requested in other comments (¹⁸O – also Noble gas work which we cannot publish) independently verify that the physical calculations of Schaffer *et al.*, are correct.

We have tried to summarize this in the introduction (line 61-80):

“For example, inflowing water masses are resident for approximately 2 years underneath the Ross Sea Ice Shelf (Reddy, Holland and Arrigo, 2010), and ~162 days underneath Nioghalvfjærdsbrae (commonly referred to as the 79° North Glacier) (Schaffer et al., 2020). [...] where the fresh water residence time is well constrained from prior surveys (Wilson and Straneo, 2015) as well as 5 moorings which were deployed for the same year where we present nutrient distributions, 2016-2017 (Schaffer et al., 2020).

Line 66-67. Do they both come up with an estimate of 162-d? For what time of year is this estimate, and how would the storage time vary with seasonal glacier hydrological evolution? The estimates of (Wilson and Straneo, 2015) and (Schaffer et al., 2020) concern different years. The 162 day estimate refers to the same year as we are concerned with and is more accurate as it is based on a year of extensive mooring observations. The (Schaffer et al., 2020) paper discusses this extensively. It is important to note that overturning in the cavity is not strongly affected by freshwater discharge i.e. seasonal glacier hydrological evolution does not affect the rate at which water circulates through the cavity, the flux is too small, the cavity is too big and the shelf forcing is too strong by comparison (von Albedyll, 2018). Please note that 90% of glacial freshwater discharge is driven by subglacial melting; submarine discharge e.g. to the glacier's grounding line contributes only 10% to subglacial freshwater discharge (Schaffer et al., 2020). Thus 'seasonal hydrological evolution' has practically no effect on cavity exiting water and shelf forcing essentially dictates the turnover rate. Given the extensive discussion of this in (Schaffer et al., 2020) we only make brief comments herein, but we add some text better describing this early in the text as it is critical for understanding the flux estimates.

We emphasize this later in the text (Line 503-508):

“In systems like 79NG, where shelf forcing rather than freshwater discharge controls the residence time of water masses within the glacier fjord (von Albedyll, 2018; Schaffer et al., 2020), lateral dFe flux may be completely insensitive to short-term changes in freshwater dFe or LpFe concentrations (although is still ultimately dependent on a supply of labile Fe to maintain a large LpFe pool within the subglacial cavity) and far more sensitive to changes in AIW ligand properties and inflow rate.”

Table 1: very useful to have this! But there seems to be some overlap between dFe and LFe based on size. Is LFe also the filtrate? Maybe provide more details so the reader can see how these are processed differently?

Please note we have now clarified that Fe-binding organic ligands in our study refer to the 0.2 µm-filtered dissolved Fe, hence are termed dFeL.

However, there is overlap between dFe and dFeL (formerly LFe). LFe can be filtered or unfiltered depending on the study, the organics which bind Fe can be present in either particles or filtered (<0.2 µm) components and it is thought that similar organic “L” material is found in the soluble, colloidal and possibly also particle phases (Gledhill and Buck, 2012), although this applies to marine environments where the colloids are likely less lithogenic in origin. An extra sentence is added (Table 1):

“Ligand-bound dFe (dFeL): Dissolved Fe bound to organic molecules (samples for iron ligands are filtered; dFeL is part of the dFe pool)”

Figure 1: L129: S11 isn't marked on the map? Would also be good to label scale bar, and show an inset map of the broader region (i.e. showing all of Greenland and where this study site is specifically).

S11 is added.

Methods:

L113: I understand that the cavity isn't the focus of this study, but it seems like residence time in the cavity and any prolonged water:rock interaction might be key to the crustal element fluxes measured downstream.. so a few more details so that the reader doesn't have to go and reference the other articles might be helpful.. e.g. mean 162-d residence over what portion of the meltseason, and how might this evolve seasonally? What is the physical nature of this cavity?

This is described extensively in much recent work by the Alfred-Wegener Institute (Schaffer *et al.*, 2020). As requested, we have added a concept figure to explain the cavity and a few sentences to summarise the excellent Schaffer paper on this (line 93-101):

*“Warm (0.6-1.3°C) and saline ($S = 34.58-34.80$) AIW found at depths >268 m (Fig. S1a/b) induces basal melting along the floating 79NG ice-tongue that covers the entire length of Nioghalyfjærdsfjorden (Wilson and Straneo, 2015; Mayer *et al.*, 2018). This inflow of heat presently occurs throughout the year and thus, as basal melt of the ice tongue accounts for ~89% of freshwater exiting the cavity (Schaffer *et al.*, 2020), there is less pronounced seasonality in mAIW outflow than would be expected in a system dominated by subglacial discharge which largely occurs over a narrow time period in summer (Mankoff *et al.*, 2020). Mixing between inflowing AIW, subglacial discharge, and basal ice melt dictates the properties of mAIW which exits the 79NG cavity as a subsurface flow (Fig. 2). The mean residence time of water in the cavity for 2016-2017 was ~162 days (Schaffer *et al.*, 2020).”*

Note as above freshwater discharge does not discernibly affect the cavity circulation so this is not a significant feature.

L122: How did you get this ratio?

It is derived from the salinity change between mAIW and AIW. See (Schaffer *et al.*, 2020) for more details.

Figure 2: Grey dots are VERY hard to see.. can you make bigger / clearer? Can you label PSW, mAIW, and AIW on the figure itself? Grey lines indicate, trace metal samples, but what depths were samples collected at? Can you summarize somewhere each of the depths were samples were collected? The resolution of the figure also appears to be poor and fuzzy. Can you improve upon this? It just makes it very hard to see the data.

High resolution images and a clearer description are provided for the revised manuscript.

Results & Discussion:

L143: Can you state in ()'s the enrichment range observed for Si and the depths that the Si was enriched at? It's very difficult to see on the figure. When you say “the glacier front” you are referring to 79NG vs its neighbouring glacier (mentioned on L120) correct?

The silicic acid enrichment is interesting because it's not significant, and an enrichment above inflow concentrations is only observed at 4 specific datapoints (now stated). The vast majority of silicic acid concentrations in the cavity outflow are equivalent to those in the

inflow which means that overall there is no significant glacier-derived enrichment. We are referring upon this from line 157:

'[...] Within mAIW and AIW, Si(OH)₄ concentrations were enriched sporadically at multiple depths close to the glacier front across all 3 water masses (range 2.3-41.3 μM for all stations >19°W), indicating a glacier-associated Si(OH)₄ source (Fig. 4c). Distinct maxima in Si(OH)₄ enrichment were observed in PSW (30.4 and 36.3 μM, <20 m at large CTD station 228), and once in mAIW (41.3 μM at 100 m at large CTD station 229).'

What is the grounding line of 79NG and the depth that the subglacial discharge comes out at? I don't believe I saw this mentioned in the methods? Are these elevated Si concentrations in line with this depth of release? In Fig3, it would be nice to see the turbidity data so the reader can confirm that the higher Si observed is in the plume and associated with a glacier source. And I assume you are referring to dSi here but perhaps good to clarify as you mention labile particulate Si in your introduction.

The grounding line of Nioghalvfjærdsfjorden is estimated to ~600 m (Mayer *et al.*, 2000), which is ~150 m deeper than the maximum depth at the cavity entrance (station S1). Note, there is no reason why an elevated signal would be seen at this depth, because freshwater released at this depth is buoyant. The depth at which water with subglacial discharge enrichment achieves neutral buoyancy in the water column will invariably be shallower than the grounding line depth – for deep grounding lines there is normally 100s of m difference between the grounding line and the neutral buoyancy depth (e.g. Carroll *et al.*, 2015). So, no the observed high silicic acid values do not correspond to the grounding line, but there is no reason why they should. Nor do they correspond well to the turbidity maximum in mAIW outflow. We refer to 'silicic acid' throughout, now clarified.

We have added turbidity data to the general distribution of macronutrients near the 79NG calving front (now Figure 4).

L148: The phrase “the dominance of AIW as a nutrient source to the region” seems inconsistent with the statements on L144 that the highest Si concentrations observed were from the glacier.

We consider that this is not inconsistent; please remember the vast majority of measured silicic acid concentrations are indistinguishable between AIW (with no glacier influence) and mAIW (with glacier influence). Furthermore, from a volume perspective, the freshwater outflow is very small compared to the saline inflow, thus the concentration of any nutrient in freshwater would have to be, on average, about 80x larger than inflowing AIW for freshwater to be roughly equivalent as a nutrient source. No nutrient distributions come close to this (not even dFe). The dominant drivers of nutrient dynamics should now be clear from the PCA and the GAM shown (following line 242). AIW dominates the distribution of NO_x/PO₄/silicic acid.

The linear regression for the Si data in Fig 4 is also the weakest R² (only 0.50). Which stations are being shown in Fig 4? It seems like on Fig 4 there is a clump of higher concentration higher S water (so marine source) but there are still some high(ish) concentration, lower S water .. which doesn't indicate AIW as the nutrient source.

The phrase is consistent with the gradient of the plot of salinity vs silicic acid (Figure S4). It is impossible for a freshwater source to dominate nutrient inputs to the region if the gradient of nutrients vs salinity is positive across the salinity gradient. As above, the dominant drivers of nutrient dynamics is clear from the PCA and GAM and neither freshwater or turbidity is a major driver of silicic acid dynamics; AIW is unambiguously the nutrient source in all cases. Silicic acid vs salinity plots always show less linearity over the shelf compared to NO_x and

PO₄ species due to the different remineralisation depths, so a poorer correlation does not necessarily have anything to do with the specific nature of freshwater sources or features unique to this catchment (both the GAM and PCA verify that it does not, line 242 onwards).

Figure 4: Which stations are being shown here? Would be good state so we can see if it's just what is in Fig 3 or additional stations? Doesn't seem to match up with Fig. 3 – i.e in figure 3 for Si there are concentrations up to 20 uM which aren't reflected in Fig 4?

All stations on the shelf are plotted (S1-6, S8-13, now Figure S4). Yes, there are 4 silicic acid concentrations which would not appear on this y-axis if all stations from the large CTD were plotted, but inclusion of the large CTD data does not change the gradient, fit, or interpretation of data (see Supplement). The main reason for separating the large/trace metal clean CTD is that statistics cannot be performed on the combined dataset without excluding the large CTD data that is not paired with trace metal data.

L150: The phrase “In contrast to macronutrient distributions, positive Fe anomalies were observed” isn't quite true for all your macronutrients – i.e. Si - as positive anomalies can also be seen for Si in Fig 3c.

The anomalies for silicic acid are not significant, they refer to only 4 datapoints (there are 254 datapoints in mAIW over the same scale showing no such change and so we think it would be cherry-picking data to construct an argument based on the 4 that do), whereas for LpFe, all LpFe values across the shelf are elevated. We think the above statement is therefore correct. Any statistical approach shows clear divergence between all macronutrients (including silicic acid) and all Fe species.

L151: reference to Fig 3d – can you show turbidity on Fig 3d too so the reader can see for themselves the max LpFe coinciding with max turbidity vs putting that info in the supplemental?

Yes, we add this to Figure for clarity (now figure 4).

L152: It seems as though the max in LpFe and TdFe is occurring when attenuation is lower according to Fig S2?

Correct for the innermost stations. Note that the attenuation values scale inversely with turbidity, i.e. the higher LpFe and TdFe at S1 is roughly scaling with turbidity as would be expected (this is also now made clear through the PCA, Figure 6).

L177: Well, it seems like in Fig 5 at S1, there are definitely higher concentrations down the water column compared to the other stations so I'm not sure it's quite accurate to say that the distribution of dFe was less dominated by the near-glacier stations.

Yes, but please look at the outer shelf. There is elevated dFe at S1, but the concentrations at S1 are lower than in the outer shelf. Please note as plotted this only shows elevated concentrations at S7, but such a signal is found throughout the EGC (see Krisch *et al.*, 2020). Using all the cruise data (which extends across Fram Strait and not just the stations shown in Fig. 1 close to the 79NG which are the focus of this study) shows that the elevated dFe at the shelf is higher and more spatially extensive than the locally elevated dFe close to 79NG.

L178: state in ()'s again the depths of the mAIW outflow please.

Stated (Line 102, and caption Figure 3).

L218: These extrapolated values aren't shown on Fig. S4? It's a bit unclear how from the linear regression shown of salinity and dFe the authors came up with an extrapolated mean FW end-member of 7nm.

Now shown (Figure S5), we also explain this fully in the supplement as it was not well explained in the original text (Supplement, line 125 onwards):

“For dFe, conversely, the intercept is positive (7.3 ± 0.7 nM for the region, or 5.0 ± 0.8 nM for stations S1-6) suggesting a freshwater concentration of ~5-7 nM. This is derived from saline data with $S > 24$, so the intercept is informative concerning the dFe remaining after non-conservative loss and not the freshwater concentration before this loss (which would invariably be much higher) (Schroth et al., 2014; Zhang et al., 2015; Kanna et al., 2020).”

L228-229: I agree there is no clear positive linear relationship but it would be nice to see the R2 values as you have shown / stated in the text for the other correlations. Also this is (I believe) the first mention of anomalous data from S4, which isn't shown in the other section plots .. why were these anomalous – and what were they? - and why were they excluded? Shown (Figure S7). No values are excluded, we just note that S4 has higher dFe compared to other 'inner' stations (S1-S4) which likely reflects its location in an area of intense mixing. S4 does not sit on the main inflow/outflow pathway so it is not shown in the main section proceeding away from the glacier, but the data is included in all stats. We now only use dFe/TdFe (dFe/LpFe) data from the two station (S1, S2) immediately adjacent to the 79NG terminus because we think that incorporation of other stations in greater distance are not representative to cavity outflow.

Figure 6: seems be come before Fig 7 but Fig 7 is referenced first? (L229)
Order now changed.

L242: It would be useful to see the data from S4 somewhere...
It was already shown in Fig. S5 (and still is in Fig. S6).

L257: It's unclear how the FW endmember value of 23 μ M was calculated? Would this just be for dissolved Si?

We explain these in the supplement (Page 6 onwards). This is for silicic acid (unfiltered). For clarity we refer to silicic acid - $\text{Si}(\text{OH})_4$ - throughout.

L265-267: It's unclear to me if this statement is just based on the data presented here which is all dissolved Si? I don't see any data presented on labile particulate Si even though it's mentioned in the introduction. Without that data, its difficult to compare the 23 μ M endmember to the total Si export proposed for runoff or basal ice - since both of those estimates include particulate data (L264), which this study does not have, correct?

The study does not have particulate Si data specific to reactive phases, although we do have HF-digested samples for a suite of 26 trace elements which show that the extent to which lithogenic derived elements are being laterally exported out of the cavity is minor. But we are not comparing our calculation to exported particulate data. We are comparing it to budgets that constrain lateral dissolved silicic acid export. (In the revised text, it is important to note that the silicic acid enrichment in mAIW is not significant, however it is defined (see Supplement), so the discussion of silicic acid is much shorter than originally). Given the length of the ice tongue, the vast majority of sediment deposition from freshwater is going to be deposited prior to freshwater exiting the cavity as mAIW. Thus, most transformation of any labile particulate Si into silicic acid should occur on this timescale (and then be evident in mAIW as silicic acid). Note the only places evidence of such Si dissolution has been found in the water column are within a few km of glacier outflows at low salinity (>15 , although

equally there are some case studies where no such phenomenon is evident on this scale), likely because this is where most glacier derived particles are being deposited. The Ra data is consistent with this hypothesis.

L284: I'm curious as to how the authors came up with the value of 1.3 nm for the "maximum cap" on the dFe.. was that the average value? Because certainly in Fig 7 (both A and B) there are many points with values above this 1.3 nm cap.

It was defined as the dFe concentration of the linear fit at 0 LpFe/TdFe. Retrospectively, this may have been misleading and did not consider the variability in dFe and dFe-binding ligand concentrations we observed. A better cap would be the ligand values from our colleagues (Ardiningsih *et al.*, 2020). We have edited this in the text (lines 426-429):

*'It is well established that ligand properties and ligand concentration are a limiting factor for what fraction of glacier derived LpFe is dispersed as dFe (Lippiatt, Lohan and Bruland, 2010; Thuróczy *et al.*, 2012) and so the concentration of ligands, which ranged up to 2.6 nM eq. Fe, places a cap on the dFe concentration that can be laterally exported.'*

L285: The statement the "overall asymptoting dFe/LpFe relationship" refers to what figure specifically? Where is this relationship shown? It's difficult to assess the validity of this statement as written and with the data shown.

Yes, this was perhaps misleading. This statement was referring to Figure 7 in the original text. This Figure is now found in the supplement (Figure S7); we have restructured our argument on the restrictions of cavity dFe export (paragraph following line 421).

L336: "All glacier-associated processes" that occur in the cavity prior to release in the ocean.

Yes, although depending in where and AIW properties are defined, this could also be construed as "all glacier-associated processes" that occur in the ice cavity, along the fjord or over the shelf. We explain this in the new supplement section (Table S6). It makes no difference where we define AIW properties (e.g. macronutrient concentration) in most cases (except with respect to the extent of NO₃ drawdown in outflowing mAIW), so it is likely the observed spatial scale is capturing all local processes that can be attributed to any sort of glacier associated effects. Given the Ra signal, and the Fe isotopic data, it is evident that the signal emerging from 79NG in mAIW is no longer glacial in origin i.e. we think the above comment is correct.

L365: Is the "removal of dFe independent of near-surface primary productivity" because S1 is in the cavity which is a tunnel and thus in the dark? I'm still having a difficult time picturing this cavity so more descriptions of this earlier / a picture would be helpful, as the procurement of samples from this field site is definitely the novel aspect of this study. It would be nice to see PAR data to back-up this statement that S1 is independent of primary production activity. However, from Fig. S1D it seems like there is still primary production at S1?

We hope the description is now much clearer, it is clear we needed a concept figure or cross section in the text to describe this. S1 is located in close proximity (≈ 100 m) to the ice tongue, so at this point any outflow in mAIW has just emerged from the cavity (which is indeed dark) and so it is a reasonable assumption that there was no significant phytoplankton activity up to this point in mAIW and that any biological activity does not discernibly change nutrient distributions between inflow and outflow (verified by comparing nutrient distributions in the Supplement). Note that this assumption even holds elsewhere in larger glacier fjords, because near-surface waters are being advected downstream, and inner-fjord waters are entrained from depth, so even the pronounced summertime blooms which have

been documented as a result of glacier induced mixing are not observed right at the glacier face. They only become evident some km/ a few days downstream of the terminus once ice melange is reduced and cell counts have had time to increase (e.g. see work by Meire in Godthabsfjord (Meire *et al.*, 2017)).

However, yes, there is phytoplankton activity downstream of S1, and there is phytoplankton in Polar Surface Water at S1, but these waters are not emerging from the dark in the same way that mAIW is. Please note that phytoplankton activity downstream of 79NG is only observed at depths <50 m; mAIW we observe at depths between 125-268 m.

L420-433: This argument assumes that the processes occurring in the cavity are the same as those that might be occurring in subglacial channels..

We are not sure what this comment refers to, but we hope that the concept figure better clarifies what the cavity is (Figure 2). The processes controlling nutrient distributions do appear to be very similar to other case studies on the same coastline near more “typical” marine-terminating glaciers e.g. the results herein for macronutrient distributions (no/very little Fe is presented elsewhere) are generally consistent with those from the other 2 case studies in East Greenland (Cape *et al.*, 2019; Seifert *et al.*, 2019).

...this may not be necessarily be true (e.g. flowing vs stagnant water for example). Thus, by only using the residence time of water in the cavity and concentration values from other studies conducted at different (non-floating ice-tongue systems) it seems (at least to me) unconvincing. And as the authors point out subsequently, this cavity they were able to sample – though novel – is not really representative of Greenland tidewater glaciers.

Please note this text is no longer included in the manuscript due to the statistical results. Note also, the concept of ‘stagnation’ definitely does not apply at 79NG as we know there is constant inflow/outflow and no sufficiently shallow sill to prevent this (Wilson and Straneo, 2015; Schaffer *et al.*, 2017, 2020). Stagnation could only occur in places where the circulation is restricted by a shallow sill, we cannot find any definitive published assessments of how often this is the case around Greenland, but it is certainly an unusual condition that a glacier fjord basin is stagnant because most large basins do not have shallow sills. The only published example we can think of where this term could apply is the inner basin in Kangerlussuaq in W Greenland where water in the inner basin does not appear to be flushed on annual timescales (Nielsen, Erbs-Hansen and Knudsen, 2010). There may be other cases, but this is definitely not the case for 79NG. The topography at 79NG would not permit stagnation and the annual survey and moorings across the region have defined the circulation. The cavity is not representative in the sense that there are only <10 systems in Greenland where a similar concept figure could be drawn, all in the north (e.g. Petermann and Ryder Glacier, (Wilson, Straneo and Heimbach, 2017), but the power of the study site is the lack of spatial overlap between mixing processes and processes associated with phytoplankton uptake/drawdown of nutrients. The dominant processes within the cavity controlling nutrient distributions do seem to be comparable to those observed elsewhere (see above).

L444-445: But as the authors themselves point out above, glacial meltwater inputs vary on daily to seasonal timescales, as well as spatially depending on bedrock geology, upstream origin (e.g. supraglacial melt, basal ice melt) and processing of waters (e.g. subglacial flow paths, proglacial lake storage).. thus, there is value in detailed characterization of glacial meltwater concentrations and processes in order to understand the mechanistic basis of this variability – especially because it is not always feasible to sample within a cavity directly at the calving ice front.

We agree with this comment in general, but note that the chemical signal emerging from the cavity in mAIW is not from meltwater in origin, (please see Ra data); rather it is an aged sedimentary signal that is several steps removal from direct freshwater input. Note also that almost all of the uncertainty in flux estimates of dFe and silicic acid occurs as a result of processes occurring across the salinity gradient and thus when trying to better constrain lateral fluxes (in the ocean), more freshwater data does not discernibly reduce the uncertainty. Contrasting (Meire *et al.*, 2016; Hawkings *et al.*, 2017), Si(OH)₄ estimates for example differ by a factor 10, with the difference entirely due to how salinity trends are interpreted and not at all due to changes in the freshwater silicic acid concentration used. Similarly, dFe fluxes are all constructed using the assumption that the fluxes scale with freshwater discharge/concentration, but there is sparse proof that this assumption holds because there is not much literature concerning how dFe removal changes across the salinity gradient specifically for glaciers. If this is not constant, which it does not appear to be in rivers which are better studied, relatively small changes in dFe losses over the salinity gradient have a larger impact on lateral fluxes than the same order of change in dFe concentration. We elaborate on these points in the section ‘Implications for lateral dFe transport’ (page 19-20).

Note also that the cavity circulation here is almost completely independent of subglacial discharge (von Albedyll, 2018) and thus not a major physical feature.

Reviewer 2

The aim of this paper is to investigate and quantify the role of ice melt on nutrient concentrations in seawater off NE Greenland. Whilst this is only a temporal snapshot of one location, Nioghalvfjærdsbrae, where ice melt flows into the ocean via an subglacial cavity, the dataset is unique and adds to the overall picture of biogeochemical cycling in high-latitude settings. The data are of high quality and the manuscript is generally well-written and clearly structured: I enjoyed reading it. This subject will be of broad interest, and the manuscript should be suitable for publication in Nature Communications after some revisions. There are a few places that are a little opaque, leading to some apparent inconsistencies, which mean that some of the conclusions are not as strong as the authors suggest. As such, I have a few recommendations, clarifications and corrections, which I hope will be constructive.

In addition to the unique setting, there are some important key points that I've taken away from this paper. Firstly, dFe and dSi are being added to the ocean from glacial meltwater. Secondly, that there is a decoupling between direct meltwater inputs and nutrient release into the oceans due to residence time of water in the subglacial cavity. The influence of residence times is poorly constrained in general for fjord and estuary systems, and important to consider when attempting to quantify nutrient fluxes. Thirdly, a more subtle message is that there is a decoupling - as a result of dynamic interactions - between particulate and dissolved phases. Is meltwater a good tracer for behaviour of particulates, and how tightly coupled are particulates and their dissolved counterparts? What are the role of colloidal fractions and ligands? All of these are pertinent questions.

My main issues relate to how these points become a little lost towards the end of the manuscript, and need to be clarified so that the key findings of the study can be emphasised correctly and robustly.

Thank you for the comments, the summary is largely correct but an important change in this revision manuscript is that a statistical approach suggests we should be more cautious about the role of the 79NG as a silicic acid source. There are some high silicic acid concentrations close to the terminus which are likely associated with glacier-derived silicic acid, but the enrichment over the region as a whole, or in mAIW as a whole, is not significant. The discussion of changes in silicic acid is therefore slimmed and we hope the shorter final section reads better.

How endmember values for the integrated freshwater input are calculated:

It appears that the authors use two different methods for this calculation for dFe and dSi: they use a linear regression between salinity and dFe to get at a dFe endmember (line 219), and use a dilution factor to calculate the dSi endmember. Apologies if I've missed it, but is there a reason for this? I'm guessing this is because the linear regression method results in a negative endmember value for dSi (as discussed below). (As an aside, I would be interested to know what endmember value would be obtained for dFe by using the dilution factor method used for dSi.)

A linear regression for dFe is the standard approach for estimating the endmember to use in calculations for non-conservative parameters. We now show all possible calculations in the supplement to avoid confusion about different definitions/derivations of zero-salinity concentrations and what they imply (page 6 onwards). Note the negative endmember arises with any method, likely as a result of change in gradient in the nutrient vs salinity trend at

low salinities. You can see such a ‘v-shaped’ trend in any nutrient on a comparable scale in the other glacier fjords where full depth nutrient concentrations are plotted around Greenland (see Figure 4 in (Hopwood *et al.*, 2020), the same trend is evident if PO₄ or NO_x concentrations are plotted from the same datasets), so this is not specific to 79NG. You also get a similar negative intercept via GAM fits to the same data. We have written an explanation about this in the supplement as it is not necessarily intuitive why this happens (section ‘Applying linear regression to 79NG’, page 8).

There are some important issues that arise from these endmember calculations.

Iron:

The linear regression method, used for dFe, assumes a relationship between dFe and salinity that might not be expected given the complex array of processes occurring on the shelf (i.e. the decoupling between dFe and salinity due to phase transitions, sedimentation etc.). Given that the particulate phases are sinking (line 161) and that there is a fairly strong benthic input of dFe (line 185, Figure 5), this benthic supply of dFe (and in fact any dFe cycled in the water column) would not necessarily be associated with low salinity waters. In fact, looking at Figure S4, there is a considerable amount of scatter that might well point towards such decoupling. I would at least like to see some statistics to back up the use of a linear model in this case (Figure S4, line 219). What is the uncertainty on this integrated freshwater endmember value? Without having the salinity values available I was unable to do the quick calculation, but I would suggest that there is a large range in the intercept value on the y axis of Figure S4. It is important to include those uncertainties in the manuscript.

We have added uncertainties on the values derived from linear regressions and other methods. We also add a GAM analysis which suggests that the interaction between salinity and distance from the terminus is the best factor that explains variance in dFe concentrations (see section ‘Statistical data treatment’, page 11-14). We suspect the scatter arises from the difference between areas where dFe is under kinetic rather than thermodynamic control (a large part of the scatter arises in surface waters at S4). Comments on sediment exchange are interesting in light of the new Fe isotopic and Ra values now discussed as they indicate a clear difference between the origin of dFe in subsurface mAIW and the elevated dFe in surface waters near the glacier terminus strongly suggesting that extensive sedimentary exchange is occurring beneath the ice cavity (following lines 221 and 333).

This relates to a further complication with the discussion surrounding the lack of correlation between dFe and LpFe (line 234). The authors claim that this challenges “the widely used assumption in flux calculations that LpFe is a universal source of dFe and conversely may reflect the role of particles instead as a sink of dFe across the studied region”. However, I would suggest that all this observation could also just highlight the decoupling in space between dFe and LpFe as a result of the complexities associated with sinking and sedimentation. I would not expect a clear correlation between dFe and LpFe in samples taken from the same niskin bottle, partly because of all the challenges involved in obtaining reproducible particulate data from niskin bottles as a result of particle settling, and partly because of particle dynamics in the water column. The complexities of dealing with such a dynamic system are touched upon in the data (e.g. caption of Figure 7, and the exclusion of the datapoints from the region of strong surface mixing), but I feel is not fully appreciated.

We consider that the comment on use of a niskin bottle is not particularly well supported by our data for the following 3 reasons (we assume the reviewer means our sampling bottles,

these are not niskin bottles but GoFlo bottles), nor are we sure it is well supported for the extensive GEOTRACES data available for other shelf areas.

Firstly, the lack of correlation between the total dissolvable and labile particulate fractions, and dissolved fraction, at 79NG arises mainly for Fe. If we look at some other trace elements (e.g. Mn or Pb), we see a much stronger correlation. A general method artefact would lead to poor correlations for all elements and their particle fractions:

Figure: Comparison of dissolved and total dissolvable Fe (top left), Mn (top right) and Pb (bottom) of PS100 station S1 immediately adjacent to the 79NG terminus. X- and y-axis values in [nM].

Second, generally in other shelf regions we do see a strong correlation for dFe and particle phases collected from the same apparatus using the exact same techniques. For example, if we look at dFe data from the next cruise conducted after PS100 (Meteor cruise M135 to the South Pacific) with the same equipment, same analysts, same sample handling and techniques, but at lower latitude in a shelf region with no glacier influence, we see perfect correlations between LpFe or TdFe and dFe (figure below). This is consistent with similar North Atlantic shelf work by other groups, for example the French GEOVIDE work and the British Shelf Sea Biogeochemistry programs, so there is not much evidence to suggest a critical widespread flaw in our apparatus.

It is a bit beyond the scope of a paper focused on only one case study region, but we suspect that a quantitative comparison to other GEOTRACES regions would also show that the absence of a correlation is not common elsewhere and may reflect the predominant lithogenic and rather inert nature of the particulate Fe phases in this area (for dFe vs TdFe and LpFe, this is certainly the weakest correlation we have found in our own case studies - across 5 shelf areas - which cover regions with similar or lower ranges in particle-Fe concentrations).

On a side note, a similar de-coupling is evident in work elsewhere with slightly different approaches, so this is not unique to our dataset ((Lippiatt, Lohan and Bruland, 2010; Hopwood *et al.*, 2016; Kanna *et al.*, 2020).

Figure: Dissolved and particulate Fe comparison for Meteor cruise M135 to the South Pacific (unpublished GEOTRACES data). LpFe and dFe definitions exactly as herein, total particulate refers to HF digested particles.

Finally, if it was the case that there was a spatial decoupling between particulate and dissolved Fe phases, but glacier derived particles were still a significant dFe source herein, then dFe would be higher across the region, especially in mAIW outflow. Consider that mAIW is the only outflowing water mass from the cavity. If there is any net export of dFe it therefore has to occur in mAIW. Even if particles were sinking through mAIW, being remineralized and forming dFe entirely in inflowing AIW beneath, the only way this dFe could then exit the cavity (or the inner fjord) would be by inward advection in AIW, followed by outward advection in mAIW.

It is also important to note that the lack of correlation between dFe and particle phases does not arise because of a lack of variation in TdFe or LpFe. Both, LpFe and TdFe, show large ranges compared to other cruises but there is limited variation in dFe.

Specifically with respect to processes occurring within our bottles after sample collection, work under more challenging conditions (in an oxygen minimum zone where there is the additional problem of O₂ seeping into the bottles and causing precipitation of dFe) we have previously demonstrated (Rapp *et al.*, 2019) that there is no significant shift in dFe, cFe or pFe concentrations over the first few hours after sample bottles are moved from a sampling rosette into the ship's clean container lab. We therefore have no evidence of substantial changes in Fe speciation between bottle closure and subsampling unless these changes are instantaneous and reproducible – which is why we are following GEOTRACES protocols to ensure a high degree of reproducibility.

We therefore politely disagree with the reviewer's comment, but would be interested in any data to verify that there is a general flaw in using GEOTRACES sampling systems that skews analysis towards finding poor correlations between dissolved and particle Fe phases.

Silicon:

Firstly, I think it's important to note that it's challenging to calculate a robust silicon budget without having amorphous silica (ASi) and biogenic silica (BSi) data. A lot of the conclusions are based on extrapolations from other – quite different - regions of Greenland. There are no data about how much of the dSi is taken up biologically. I'm surprised that these data are not available, given that this study was part of a GEOTRACES program (also see

below). The fluorescence data are shown in the supplementary information, but it's not possible to see how much of this is diatom biomass. Were there any cell counts, HPLC or Chl a, b and c extractions from the bottle samples?

We derived only silicic acid outflow, and not a budget. ASi or BSi data are not available (not required key parameters for a GEOTRACES cruise) and there are no cell counts – fluorescence data is now replotted to show there is no activity close to the glacier in mAIW. We hope the conceptual figure (Figure 2) better explains the role of the cavity, it should be more apparent that silicic acid outflow at S1 in mAIW has not been subject to recent silicic acid drawdown due to diatom uptake as there is no phytoplankton activity in the cavity. It should also be noted that the PCA and GAM (and similar analysis in other different Greenland fjords) demonstrate that AIW inflow is the dominant source of all macronutrients (see section 'Statistical data treatment' in the main manuscript, and supplementary material from page 7).

Secondly, I'm not clear how the freshwater endmember calculations were carried out. I've assumed that they're based on the 1:84 dilution mentioned on line 122. If this is the case then I agree that the difference between mAIW and AIW of 7.73 and 7.44 μM does give an endmember of approximately 24 μM . However, the authors state on line 256 that the 7.73 μM value for mAIW has an uncertainty of 3.7 mM, which means that there is a large uncertainty on their endmember composition value, which is not appreciated in the text.

The uncertainties are now shown in all values. We now note that the silicic acid enrichment in mAIW is not significant irrespective of how it is derived (see Supplement). There are 4 high silicic acid concentrations at specific points across the region, and these are what drive the high standard deviation on mean calculations. The majority of concentrations in mAIW are identical to AIW hence the difference in silicic acid concentration between the two water masses is not significant, irrespective of how or where it is defined (now shown explicitly in the supplement table S6).

The lower estimated value would indicate that mAIW is diluted with respect to dSi, whereas the upper estimated value would result in an extreme endmember composition of approximately 335 μM (and this doesn't include any remaining particulates). Regardless, to make the strong statements (e.g. on line 263 onwards, and highlighted in the abstract) based on these calculations without a proper appreciation of the associated uncertainties is somewhat misleading.

Statistics are now included and we show that there is no silicic acid enrichment across the region. This estimate does include any particulates which are acting as a source of silicic acid over the timescale of their residence in the cavity, and thus should include the vast majority of any Si dissolution (verified by Ra data, as the signal emerging from the cavity in mAIW is not glacial in origin but a shelf signal, line 225). Note that the key point is that there is no significant change in silicic acid, and therefore we can slim this discussion considerably. mAIW is neither diluted nor enhanced with respect to AIW in terms of its silicic acid content, the prospect of an extremely high silicic acid freshwater endmember could also be eliminated looking at the gradient of the silicic acid vs salinity plot as shown in both the original and new text (Figure S4).

In brief, the main conclusion from this section of the manuscript is that the meltwater from the subglacial cavity is a source of dSi to the mAIW (although I would question the magnitude of the input given in the manuscript without having the associated uncertainty stated). However, later on in the manuscript, I then feel that the authors contradict themselves, by saying the AIW is the major input of dSi because of the negative freshwater

endmember from the linear regression method. I would have thought that this demonstrates that this simple linear regression model is not a robust way of judging the integrated freshwater input (for the same reasons as discussed for dFe, especially in the case for Si as there are no particulate data). Note also that there is no uncertainty given on this intercept value (-16 mM, line 403).

A linear regression of nutrients with a positive gradient of all nutrients vs salinity shows unambiguously that the major source of nutrients to the shelf is saline waters (AIW) (Figure S4). This is also apparent from a PCA or a GAM analysis (Figure 6). It is not possible that a system with a major freshwater-associated nutrient source could produce such a distribution. Uncertainties are now given. The negative intercept likely arises from a gradient change in the relationship between nutrient concentrations and salinity in the ice cavity. This is not peculiar to 79NG, it applies to any other glacier catchment for which 2D data is available around Greenland: saline, nutrient rich water flows in at depth, is brought closer to the surface by a combination of processes akin to upwelling and is then drawdown by primary production. This produces a lob-sided 'v-shaped' nutrient distribution (see Figure 4 in (Hopwood *et al.*, 2020) for all available sections plotted for silicic acid, a similar trend applies for PO₄ and NO_x).

I'm also a little confused by the calculations that follow on from line 420 onwards. I assume that the calculation of a subglacial dSi concentration of 210 μM (line 422) assumes that subglacial discharge is 11% of the freshwater budget, which has an integrated concentration of 23 μM. This then would be ignoring any input from the basal melt, and any amorphous silica remaining in solution, neither of which are mentioned as caveats. Lastly, there is also no appreciation of the impact on these calculations of the uncertainty on the integrated 23μM value – if this is towards the upper end of the estimate (335 μM, see above) then the subglacial concentration would be instead over 3000 μM. I'm afraid I don't understand where the 100 and 14 μM values come from on line 426, and I begin to lose the thread of the argument.

The calculations above are no longer included in the text as all of our statistics show that the change in silicic acid attributable to glacier associated processes whether they occur along the fjord or in the cavity is within uncertainty and we think that therefore there is not much sense in discussing where any change comes from. Note the uncertainty propagation above is erroneously high (it was presumably derived assuming the dilution factor was a constant and not a well constrained physical parameter with a small uncertainty). We have added a new section in the supplement which details what each approach to quantifying changes in nutrient distributions assumes (from page 7).

Regardless, a final upper estimate of 28% dissolution is not unreasonable and does not necessarily contradict the findings in Hawkings *et al.*, 2017. In this paper, Hawkings *et al.* do not suggest full dissolution over timescales shorter than 162 days; their experimental results (their Figure 5) would point towards 30-50% dissolution over this timespan (and these experiments were carried out using seawater with a salinity of 35.5, so dissolution might follow a different trajectory than fresher waters).

With all respect, we think that this calculation (above) is unfortunately incorrect: it assumes that particles stay in suspension for 162 days and that an outflowing glacially modified water mass is only affected by the particles in it, i.e. particles previously entering the cavity have all been advected laterally out of the region. Particles certainly have a longer residence time in the cavity than water, a defining feature of fjords is their ability to trap particles. So the water flowing into and out of the cavity is affected not only by the particles being released at a certain time point, but also particles which have previously entered the cavity and remain

there (we know from surveys around Greenland there is a pronounced sediment deposition of glacier derived material in the first few km away from discharge outflow, and this may be a contributing feature to why non-conservative addition of silicic acid has only ever been observed at the lower end of the salinity gradient in these fjords). This is verified by Ra (line 225 onwards) and Fe data (paragraphs following line 333), which show that the mAIW outflow has a shelf signal; observed increases in Ra in mAIW cannot originate from freshwater and cannot be scaled accordingly.

If the above calculation was correct, it would not explain how we see pronounced non-conservative increases in silicic acid over timescales of hours within these low salinity zones, followed by more conservative behaviour at higher salinities.

In summary, I find the endmember calculations for the integrated dFe and dSi a little opaque - I'm not really sure to what extent they question the validity (line 386) of previous estimates. I think the approaches themselves are fine, but the caveats and uncertainties involved are not discussed sufficiently. My recommendation would be to apply both methods to both elements systematically, and discuss the challenges involved. This should help to avoid the apparent contradictions and confusions.

A supplement is added which we hope now does this, we show how different calculations can be performed to investigate changes in nutrient concentrations using all available methods.

With respect to prior work with dFe, the critical insight is that cavity overturning likely regulates the lateral flux, and cavity overturning is not related to freshwater outflow. Hence lateral flux does not scale with freshwater discharge (questioning a good deal of the fluxes derived to date). The isotopic and Ra data further verifies that dFe fluxes cannot be scaled with freshwater concentrations either, as the chemical signature exiting the cavity is not from a direct freshwater source.

With respect to silicic acid, the main insight is that mixing processes explain all of the change in Si(OH)_4 distributions and any chemical effects associated with the 79NG are minor. This questions the basis on which higher silicic acid inputs have been derived (interpreting trends as having solely arisen from freshwater supplied particles or silicic acid) and whether there is much Si(OH)_4 enrichment as a result of glacier outflow in this region at all.

General Minor comments:

Please refer to dSi rather than just Si throughout (including figures), to make it clear that you are only referring to the dissolved phase.

Strictly we are referring to 'silicic acid' in unfiltered samples. It is now written throughout.

There are a number of datasets that I would add considerably to the paper that are not present. If they do exist - and I expect they will given that this study is based on a GEOTRACES cruise - then I would suggest adding them in. 1) I would include reference to the actual ligand data in Ardiningsih et al., 2020; 2) I would add any measurements of biological productivity or nutrient drawdown (discussed on line 229, but no values given other than fluorescence in the supplementary information) calculated from the cruise (and/or from satellite data); 3) I would include meltwater fraction calculations based on oxygen isotopes (also allowing quantification of any inputs from sea-ice melt/brine formation c.f Tonnard et al., 2020); 4) I would include any existing radiogenic isotope data to constrain process rates.

Ligand data is now presented explicitly (paragraph line 421). We hope that the concept figure better explains why biological drawdown (by phytoplankton at least) is not relevant for discussing processes occurring within the cavity or at inner stations in subsurface mAIW (Figure S2). In the calculations within the supplement we assess the extent to which drawdown could be significant feature downstream of the glacier (page 7 onwards). There are unfortunately no direct estimates from the cruise so close to the glacier termini, only for off-shelf. Oxygen isotope data is added (line 102; Figure 3).

Please note, biological, He, Ne and Ra isotope parameters are not part of the GEOTRACES key parameters, and hence not determined on every GEOTRACES cruise.

With respect to radiogenic isotope data, however, we agree this would be interesting, but are not sure it would be as useful as the reviewer suggests given the available resolution. ^{228}Ra data is available, but the resolution is much higher in surface waters (whereas mAIW outflow occurs at depth) and shows, as we speculated, that an Arctic signal from further north rather than local runoff dominates Ra distributions. Long-lived Ra isotopes cannot distinguish between subglacial derived, and sedimentary derived origins, so the Ra does not tell us much we do not already know in the fjord. On the shelf, it does add weight to our hypothesis that the peaks in dFe are river derived from the Transpolar Drift, and not local (line 221). The ratio of long-lived isotopes at S1 in mAIW is however interesting, and verifies our interpretation of Fe data – the signal emerging in mAIW is from AIW and includes contributions from the shelf sediment, i.e. it cannot be interpreted as being driven by runoff, which is consistent with the new Fe isotopic data now also included.

We have tried to summarize this in lines 225-233 (Ra), and lines 343-350 (Fe-isotopes):

“[...]At 150-250 m in the mAIW, ^{226}Ra is unchanged while ^{228}Ra is slightly enriched compared to shelf stations at the same depth. At 125 m, we observe a signal of highly elevated ^{226}Ra ($111 \text{ dpm}\cdot\text{m}^{-3}$) and ^{228}Ra ($48 \text{ dpm}\cdot\text{m}^{-3}$), indicative of a sedimentary input to cavity mAIW (with a $^{228}\text{Ra}/^{226}\text{Ra}$ ratio of 0.9). Subglacial runoff is estimated to be 0.07 mSv or only 0.15% of the cavity overturning rate (Schaffer et al., 2020). With the discharge-averaged concentrations found in subglacial runoff of the Leverett Glacier in western Greenland (Linhoff, Charette and Wadham, 2020) this would, after mixing in the cavity, cause increases of only $0.03 \text{ dpm}\cdot\text{m}^{-3}$ ^{226}Ra and $0.52 \text{ dpm}\cdot\text{m}^{-3}$ ^{228}Ra in the cavity outflow, negligible compared to the observed signals at 125 m. We conclude that the radium is sedimentary in origin (either from the seafloor or released from the ice tongue by melt).”

“In contrast to the isotopically light dFe signal in surface water near the Nioghalyfjordsbrae glacial front, mAIW from 125-200 m at the same station ($+0.07 \pm 0.09\%$, S1) had heavier, near-crustal $\delta^{56}\text{Fe}$ values ($\sim+0.1\%$)(Beard et al., 2003; Poitrasson, 2006), equivalent to the isotopic composition of the West Spitsbergen Current ($+0.15 \pm 0.09\%$). It is clear that mAIW is enriched in dFe compared to AIW and that this enrichment occurs either at the glacier terminus or underneath it (Fig. 8). We therefore suggest that any primary isotopically-light $\delta^{56}\text{Fe}$ signature originating from dissolved Fe^{2+} in subglacial discharge and basal meltwater is lost during extensive reworking and exchange between the dissolved, ligand-bound, particulate and sedimentary phases within the subglacial cavity (Radic, Lacaan and Murray, 2011; Homoky et al., 2013; Conway and John, 2014; Fitzsimmons et al., 2017).”

He/Ne distributions further corroborate the above comments (^{18}O , He/Ne data and the physical calculations from Schaffer et al., are all in agreement with respect to the major comments we make concerning mAIW properties and downstream fate). Unfortunately, He/Ne data is not available to be presented in our study but ^{18}O data is now included.

Minor comments and suggestions:

Could the authors please mark on Figure 1 the line of the section shown in Figure 2?
Annotated.

Line 186 – The authors should not necessarily capturing benthic inputs in niskin bottles fired at 2-5 m above the seafloor. Whilst the discharge from the subglacial cavity might represent a combination of pelagic and benthic processes (line 258) there will be benthic processes on the shelf impacting these nutrient cycles. Quantification of these processes would need measurements of porewater profiles, core incubations, benthic landers, and/or a radiogenic isotope data (e.g. U-series).

This is correct, we do not quantify the rate at which benthic processes are changing nutrient concentrations in near-bottom water over the shelf, and it may well be that shelf sediments up- and downstream to 79NG are sources of dFe but have simply not been captured by our CTD casts. With respect to 79NG discharge, by contrasting the properties of AIW before it passes into the fjord, with mAIW flowing out of the cavity, we can quantify all changes that occurred between these two time points in terms of the net change in nutrient concentration in the water column which includes any benthic additions occurring along the fjord (see supplement for full calculations). This remains the case even if the bottom measurements along the fjord are too far from the seafloor to fully capture a benthic signal, as AIW outside the fjord and mAIW outflow are not subject to this limitation.

Line 191 - The “pronounced surface peak” (Figure 5a) at station S7 is interpreted as coming from the Transpolar drift (TPD) rather than a localised source. This is a reasonable hypothesis given that high dFe values have been found in the Central Arctic Ocean surface waters in regions of high meteoric water input (up to 5-6 nM, Charette et al., 2020). The authors could also add a comment about a new paper from the same group that is already cited (Ardiningsih et al., 2020), showing that TPD waters are enriched in Fe-binding ligands (explaining how dFe could remain in the surface waters so far downstream). However, there are a couple of caveats I would also add: 1) this peak is based on one bottle measurement

We agree that the TPD is enhanced in dFe and Fe binding ligands. Along our cruise transect, we collected a substantial number of samples in the EGC that is influenced by surface Arctic Ocean (i.e. TPD) outflow. Although we did not demonstrate this in the figures plotted and did not substantiate this point previously: the PS100 covered the Fram Strait region and thus there are many datapoints within the EGC not shown in the figures (which all focus on the 79NG region which is inshore of the EGC). The EGC is elevated in dFe compared to anywhere else in the whole section.

Surface EGC (<10 m at 0-8°W; 79-81°N): 1.7 ± 0.5 nM

79NG (S1): 1.3 ± 0.3 nM

Surface data outside EGC (<10 m): 1.0 ± 0.8 nM

Please see our recently published paper (Krisch *et al.*, 2020) for more information.

...[continued from above comment] 2) without showing a transect of stations along the EGC path, it is not clear as to whether there is enrichment here due to meltwater inputs from Greenland or not.

Figure: dFe at coastal stations along the coast of Greenland by latitude from GEOVIDE, PS100, D354 and MSM85.

There are 4 cruises with trace metal data in the EGC region, all in August/September, plotting them together, it is quite clear that there is a decrease in dFe moving from the northern most to the southernmost point along the coastline of E Greenland. This supports the hypothesis that dFe at 79° North is primarily sourced from the Arctic Ocean, and that this signal - from the Arctic Ocean - is advected southwards. It is beyond the scope of this text to evaluate data from multiple cruises much further north or south, but the available dFe data shown above suggests a decline in dFe as the influence of runoff from Greenland increases. This requires further analysis, but roughly corresponds to following a section where the meltwater content is increasing from 0% to roughly 50% of the meteoric water present. The available dFe data thus largely refutes the hypothesis that runoff from Greenland is a significant source of dFe on this scale (The above correlations and fits strengthen when Arctic dFe data from further north is also added).

Also please note that Tonnard et al., 2020 (cited on line 195), also interpret shelf waters off southern Greenland to be Fe fertilized from meteoric water inputs from Greenland: (p.933) *“Surface waters at stations 53 and 61 were characterized by high MW [meltwater] fractions together with enrichments in PFe at station 53 and in both DFe and PFe at station 61... At station 61, the relative depletion of DFe at 30 m compared to 50 m may be due to phytoplankton uptake, as indicated by the high TChl a ... Hence, it seemed that meteoric water inputs from the Greenland Margin likely fertilized surface waters with DFe, enabling the phytoplankton bloom to subsist.”*

Please note also the paragraph before this in the Tonnard paper. “MW” in the above paper does not stand for “(meltwater)” as added by the reviewer. It stands for “meteoric water” which includes other freshwater sources besides meltwater (e.g. riverine/surface discharge proceeding south along the East Greenland Current from the Arctic). Indeed, as noted herein (figure above), the surface signal along the coast is already present at 79NG, and gets weaker further south which somewhat questions the interpretation of it as a local signal coming from Greenland.

Line 367 onwards - The authors should note that fluxes are very difficult to constrain in shelf environments without rate determining measurements, either from physical models or

radiogenic isotopes. For the flux estimates that they have, it might be useful to convert fluxes to Tmol/year, so be consistent with other Si budget papers (e.g. Tréguer & De La Rocha, 2013).

Something we did not stress in the original paper is that the area is well constrained in this year by mooring coverage, which was used to derive the AIW and mAIW fluxes used herein (see (Schaffer *et al.*, 2017, 2020)). We explain this better now in the introduction (lines 79) and the study region section (following line 84), and propagate uncertainty from the fluxes and concentrations discussing further limitations and caveats better (i.e. page 21). Ra data is added which at least verifies an overturning rather than freshwater scaling is more appropriate for deriving lateral dFe transfer. For silicic acid, any lateral flux is within uncertainty according to our revised statistics so these comments are no longer in the text.

Line 387 – I agree that this is not “informative of seasonal processes” – especially not considering critical processes including sea-ice formation/melt (see comment above on oxygen isotopes), mixing from internal waves, or storm-driven mixing. Perhaps add some of this detail.

There are several excellent physical papers concerning the drivers of mixing in this environment, we tried to include a better summary of these in the paper sections.

Internal waves are indeed important; 53% of the variance in cavity circulation is attributed to barotropic tides, operating at a sub-daily range (von Albedyll, 2018). A further 29% of the variance is attributed to processes operating on a daily-to-monthly timescale, i.e. regional shifts such as wind-induced Ekman pumping. Only 14% of the variance in cavity circulation has been linked to intra-annual variability such as speed fluctuations. Yet, it is beyond the scope of the physical analyses to date to conclude on mixing processes related to this variability in current regimes. However, sea ice dynamics is now recognised to have profound impacts on downstream water column stability at 79NG (Syring *et al.*, 2020), and mixing during storm events will likely also play a role in PSW and mAIW advection.

Yet, because of the strong and persistent nature of mAIW cavity outflow evident by velocity measurements (Schaffer *et al.*, 2017, 2020) and He/Ne, and the buffering effect of cavity mAIW to changes in dFe content, we suspect little variation in dFe export efficiency related to seasonal change. We have tried to include these thoughts in the revision of the paper, e.g. following line 504:

*“In systems like 79NG, where shelf forcing rather than freshwater discharge controls the residence time of water masses within the glacier fjord (von Albedyll, 2018; Schaffer *et al.*, 2020), lateral dFe flux may be completely insensitive to short-term changes in freshwater dFe or LpFe concentrations (although is still ultimately dependent on a supply of labile Fe to maintain a large LpFe pool within the subglacial cavity) and far more sensitive to changes in AIW ligand properties and inflow rate.”*

Reviewer 3

Reviewer #3 (Remarks to the Author):

Krisch et al., present the results of a GEOTRACES transect and cruise from the 79NG Glacier in North Greenland into the Atlantic Ocean focussing on the concentration and transformation of two important macronutrients, Fe and Si from within the subglacial cavity and subsequent export to the Atlantic Ocean. The authors find enrichments of differing Fe phases do not occur in the same oceanic regions, and show for the first time that the distributions of the proportions of different iron phases within the filterable iron change with distance from the glacial terminus, and not uniform as previously thought in the literature to date, which makes this manuscript of high impact. The data are of excellent quality, especially regarding the difficulty of sampling and measuring soluble (sFe) to such low concentrations, and provide new data for colloidal and labile Fe (cFe and LpFe, respectively) concentrations which shift our current perspective on their proportional distributions within the water column and during (sub) glacial export. This 79NG glacier has the largest floating ice tongue in Greenland and data and results from this location are rare and highly valued. The manuscript is well written and set out, however I think the manuscript could greatly benefit from an annotated cartoon of the subglacial ice cavity indicating the inferred Fe transformations, potentially based upon ref 32 fig 4 of the cavity, and the processes which determine Fe and Si availability and export.

Now added (Figure 2). Yes, it is apparent from all reviewers that a concept figure is needed to explain what the cavity is.

The authors use a linear regression to calculate a freshwater endmember of Fe as 7nM, and subsequently use this to upscale the Fe flux from the Greenland ice sheet and other glaciers which discharge into the Atlantic (e.g. L367). There are no errors shown for this calculation or propagated errors, which should be shown as this value is pretty low compared to other insitu measured concentrations in other studies.

Now shown for all calculations.

The same linear regression is also applied to calculate a freshwater Si concentration but does not work, so the authors use another method. In Figure 6, the authors use a power law relationship to explain the removal of dFe, so I do not understand why this was not used in the calculation of the freshwater endmember, as the resultant Fe concentration will be much higher, and may then be more comparable to those actually measured in the literature (e.g. refs in L223). I also wonder what the Fe concentration would be if the authors used the same approach for Fe as they do for the final Si freshwater end member calculation.

We show all the calculations that could be done in the supplement with the associated uncertainties (following page 7). The actual freshwater concentration at zero salinity is of limited use for flux calculations, because we know that the removal factor for this dFe is highly variable. Because the removal factor is invariably also high whenever freshwater delivers dFe into the ocean, it makes a massive difference to flux calculations: e.g. the difference between best guess upper and lower bounds to removal for dFe across the salinity gradient glaciated catchments is about 77% to 99% (Schroth *et al.*, 2014; Zhang *et al.*, 2015; Hopwood *et al.*, 2016; Kanna *et al.*, 2020). When extrapolating from salinities >24 with a power law fit to derive such a number, the uncertainty would be high, and this value would not be useful in any flux calculation. We have tried to revise this in the manuscript (lines 481-496).

I was hoping that the section starting L322 would convince me of the more global scope of the paper as the results are very intriguing, but in lines L323 to 339, L367 to 383 and L435 to 440) I find the reasons not to extrapolate more convincing than those for extrapolation of the data, and so I find that the impact of the paper is diminished. Local and regional extrapolations hold as they are well constrained within this paper, but not for all glaciers from Greenland or all those going into the Atlantic Ocean. There are also no references to the data used for upscaling or clarity as to what freshwater fluxes the data have been upscaled to. This section needs much more work and detail before it can be extrapolated to such an extent (more details in 'In-line comments').

We now better explain different calculations that can be used to fit and explore the data in the supplement. The limitations we explain in the above lines are not unique to 79NG, and all of the available 'scaled' flux calculations in the literature are derived with similar issues that lead to large differences between different scaled estimates (e.g. a 10 fold difference between silicic acid fluxes from (Meire *et al.*, 2016; Hawkings *et al.*, 2017) despite similar freshwater endmembers, and larger differences for dFe due to different flux gates). Strictly, all of our calculations are derived only for the 79NG region and thus we are not upscaling, only presenting fluxes derived from this region. We better explain in the text the assumptions used in all flux calculations to date, and how water mass fluxes already reported for this region can be used to assess fluxes.

I am also confused as to what phases were measured for Fe and what were not. Table 1 implies cFe was determined by the difference between dFe and sFe, whereas the methods in L481 suggest cFe was directly measured and sFe calculated. Please can the authors make this more explicit in the manuscript and methods as one of the conclusions rely on the result that the proportions of dFe and cFe greatly vary from the glacier front to the ocean, and it is vital to know if they were measured independently or not, to be sure of mass balance, and not a loss or gain of Fe to/from another phase.

We have corrected this mistake in the method section, and also added a sentence in the corresponding paragraph that makes more clear that cFe was determined by the difference between dFe and sFe (Line 358-361):

'The 79NG terminus (S1) exhibited cFe concentrations, calculated as the difference between measurements of sFe and dFe, of ~0.1 nM in the top 100 m excluding one outlier of 0.6 nM at 50 m which could be attributed to recent glacial freshwater input (Fig. S6).'

I am also unsure how 'ligand-saturation' was determined and measured, L341, this is not in the methods and there is only a small mention that ligands can be organic or inorganic, nothing specific as to what they could be. This is very speculative yet a whole sections are dedicated to it (L268 to L321).

We have revised this section. Whilst much of the material is presented elsewhere, a summary is now included herein (paragraph line 421).

In-line comments:

L20: As written here, the abstract suggests that the residence time of 162 days has been measured in this study, it has not. Please reference.

This has already been referenced (Schaffer *et al.*, 2020).

L56 to L58: contradicts statement in previous paragraph about enhanced freshwater flux increasing nutrient input into the oceans. Suggests this will be diluting them.

With all respect, both statements are correct. A flux is a transfer of material from A to B irrespective of how the concentration at A compares to B. From a glacier's perspective, a glacier supplies a flux of all nutrients into the ocean (as it does salt) hence glaciologists can constrain fluxes of all nutrients into the ocean as a whole. That is not inconsistent or contradictory with meltwater diluting nutrient concentrations in surface waters (as it indeed dilutes salt concentrations), or otherwise lowering nutrient availability through subsequent reactions or increasing stratification - although there are a lot of muddled statements about this throughout the literature which may be mixing interpretations between geological and interannual timescales. We have explained this with the example of PO₄ in the supplement (line 76-82).

L61: What downstream waters? As in the ocean proximal to the ice shelf? Or those closer to the exit of the ice-tongue? The open Ocean?

Well, we do not know. The paper cited (Meire *et al.*, 2017), which we think is one of the few to explicitly test this, states that there is a measurable increase within fjords. A monitoring site at the mouth of the one of the largest fjords studied therein (Godthabsfjord, Southwest Greenland, Juul-Pedersen *et al.*, 2015) suggests that this effect is no longer evident at the fjord mouth where interannual primary production is more stable and less sensitive to increasing primary production and changes in fjord hydrodynamics. Looking at the coastal area around Greenland as a whole, there is no specific evidence that primary production is increasing faster than it is across the high latitude N Atlantic or Arctic in general, so maybe this puts an upper-bound on any glacier associated effects (that, if present, they are likely regional), but we are not sure we can comment further here. There are some studies suggesting changes in bloom timing as a result of changing meltwater dynamics on the Greenland shelf (Arrigo *et al.*, 2017), but nothing we are aware of that specifically shows changes integrated primary production changing in proportion to meltwater derived nutrients. Model studies certainly show that transport of nutrients out of fjords, or other effects of meltwater on productivity on the shelf is at least plausible under some circumstances (Oliver *et al.*, 2018, 2020), but we are not sure many studies have explicitly shown this.

L73 to L75: This sentence is confusing as written as it suggests that LpFe dissolves to form dFe and cFe, but your definition in Table 1 is different to this, as cFe and sFe are part of dFe. cFe and sFe are part of dFe. LpFe is widely suggested to act as a source of cFe to the water column (ie LpFe itself is particle bound, and not in the dFe phase, but it could release cFe phases into solution).

L82: I do not think that this sentence is true, a google scholar search will show that there are many studies which study Fe transformations at ice (glacier) seawater interfaces, for example...

Front. Mar. Sci., 14 June 2019; <https://doi.org/10.3389/fmars.2019.00332>
Biogeosciences, 15, 4973–4993, 2018; <https://doi.org/10.5194/bg-15-4973-2018>
JGR Ocean Volume 122, 8, 2017, 6371-6393; <https://doi.org/10.1002/2017JC013068>

Most studies were conducted some considerable distance from the glacier-seawater interface, none of these studies include low salinity waters, and the first two do not include include near-glacier stations, although these studies are insightful concerning the downstream plumes. We think these near-glacier processes are important because large transformations are happening on this scale, work in the Bayelva for example shows that even within the freshwater systems dFe can be unstable – the authors noting at 80% precipitation of dFe over

a scale over 4 km followed by additional losses over a few more km along the outflow path. Most marine studies are simply too far away from where these mixing processes are occurring to see them, although we note the recent (Kanna *et al.*, 2020) work is actually perhaps now unique in capturing dissolved and particulate dynamics in a plume close to source (within 4 km of the terminus).

The Marsay study cited should, however, have been mentioned considering the proximity to the Ross Sea Ice Shelf (lines 528-532):

*'Whilst few dFe concentrations are available in similar localities worldwide, dFe concentrations reported herein are reasonably similar to the ~0.7 nM in outflow from the Dotson, Crosson and Getz Ice Shelf cavities (Gerringa *et al.*, 2012; Sherrell *et al.*, 2015) and the 0.2-1.4 nM in outflow from beneath the Pine Island Glacier (Gerringa *et al.*, 2012). Interestingly, dFe enrichment in subglacial discharge downstream to the Ross Ice Shelf was within uncertainty relative to shelf stations not affected by ice sheet discharge (Marsay *et al.*, 2017).'*

L89: The reference to Table 1 after this sentence suggests that you have sampled all of these fractions, but I don't see LFe in the manuscript. Despite this the table is incredibly useful as it lays out precisely which phases are in which fractions and the references used.

LFe is included in a companion text, now included herein also for completion.

L219: What is the error on this extrapolated value?

Now added to all derived values.

L221 to 223: Is it possible that Fe removal is not linear? It may be a power (like in Fig 6) or exponential removal which would, if fitted to data in Fig S4, increase your anticipated freshwater end member, and be more comparable to those actually measured in glacial fresh water (refs in L223), such as the exponential fit in Figure 6?

It is certainly non-linear, there are a variety of models for how it occurs with not much clarity on how well they apply to glacier catchments (most of our theoretical understanding comes from river estuaries), but it is important to note that the removal factor over the salinity gradient is not well defined and varies within and between catchments. We suspect, each glacial system, unique in geometry, location and mixing processes, will potentially show a different Fe removal/solubilisation behaviour (Schroth *et al.*, 2014; Zhang *et al.*, 2015; Hopwood *et al.*, 2016; Kanna *et al.*, 2020). Thus, the freshwater concentration is not particularly useful for determining the lateral flux after removal has occurred. This is why a linear fit is used within high salinity data as it derives the dFe that is left after removal has occurred.

L238 to L239: What is meant by 'high particle loadings'? how does this change whether dFe is bound to organic-ligands or LpFe? Or are organic-ligands included in the 'increasing LpFe'? This then leads to my next comment in lines L239 to L244, where you suggest that they should be correlated, which then is the reverse of the observation in lines L232 to 233 (as written).

High surface areas of particles in suspension (glacier fjords are high deposition environments with the particle inputs into these fjords extremely high in any oceanographic context). A sentence is added for clarity. Theoretically this pushes the system towards saturation of Fe-ligand phases (Lippiatt, Lohan and Bruland, 2010). LpFe is thought to include the most labile mineral phases and cellular Fe. Generally, LpFe is correlated with dFe in shelf environments. Herein it may not be because of the high lithogenic particle load in the water column, which

has been demonstrated specifically in other high particle load environments – but contrasts with other shelf environments where particle inputs are less lithogenic and generally lower (see the figure in response to comments from reviewer 1).

L257: ‘Integrated freshwater endmember’ so not a linear regression as for Fe in L221 to 223? We have added a section in the supplement showing how these are derived and what they mean and tidied the terminology with respect the freshwater endmembers.

L258 to 268: Does this assumption also impact dFe and TdFe since active redox processes in the sediment may act to release or remove Fe to/from the water column and the sediment interface.

Yes, we cannot distinguish where dFe comes from in the mAIW outflow i.e. based only on the fact there is dFe enrichment, we could speculate it either came from benthic cycling within the cavity, subglacial discharge, or ice melt, or potentially even some sort of remineralisation of inflowing particles (or a combination of any of these). But we have now added isotopic data (lines 333-352), which is insightful mainly because it strongly suggests a different source for the dFe in surface waters near the glacier termini compared to subsurface mAIW. The surface signal is more consistent with runoff from the glacier, the subsurface mAIW signal is more consistent with Fe from a benthic source. Ra also supports this conclusion for mAIW (paragraph line 221).

L275: Spatial, and seasonal?

We comment on this, in light of the fact that ligands on the shelf are altered by bloom dynamics, there may be some seasonal shifts there (lines 460-468).

L314: This is a tough claim to make, as definitions of the size fractions of colloidal Fe in publications really vary, from $< 0.45 \mu\text{m}$ and molecular weight $> 10 \text{ kD}$. You can certainly make the claim based on this paper, but not so much in comparison to some comparative literature. I think it is worth making that point here or referencing papers which use the same colloidal definitions as the one used here.

We clarify based on this definition of colloids, However, if the filtration cut off was changed, the conclusion would still hold: LpFe + dFe puts a firm upper limit on the total cFe pool irrespective of what cut-offs are used, and both sFe and dFeL as measured herein make a considerable fraction of this pool compared. The ratio of cFe to sFe is indeed extremely high in meltwater (Raiswell *et al.*, 2018), but irrespective of how cFe is defined, we think it is reasonable to state that in this catchment there must have been a sharp decline in the relative importance of cFe from salinity 0 to 24.

L336 to 339: I think I would argue the opposite as these signals can easily be monitored on a second to second time scale using many multi probes and tend to be much more logistically accessible for full seasonal and yearly studies. Or do you mean smaller ice tongues from fjord terminating glaciers? This sentence adds little to this manuscript.

In theory, yes, but such an approach has not been published anywhere yet... The rate at which sensors are destroyed by iceberg movements in these inner-fjord regions are high, deployments further downstream are possible, but this would miss the mixing processes. For example, the disintegration of Spalte Glacier, an ice-tongue that previously branched off 79NG has likely destroyed American/German moorings deployed with sensors in summer 2020. Please note, it is also a question of where mixing occurs. The salinity gradient downstream of most glaciers is present in a plume in-front of the terminus, so even if a helicopter is used to attempt to collect water immediately in-front of the terminus (Mortensen *et al.*, 2020), the salinity is already high (20+), and a lot of transformations have already

occurred on a scale we cannot measure. Attempts to stick gliders, and surface deployed instruments in these plumes around Greenland have all failed to survey these regions of intense, often violent, mixing, to quote a famous PI “they just pop back out”. Sensors are much easier to deploy downstream after the mixing processes occur. There are therefore very few physical locations where you can deploy instruments to assess the effects of glacier outflow on downstream biogeochemistry before other factors start to affect biogeochemical cycling.

L341 to 344: What about seasonal variations?

At 79NG outflow and overturning occurs relatively steadily through the year (Schaffer *et al.*, 2020), so we would expect seasonal variation to be much lower than in other systems, this is very different from a catchment where freshwater release is dominated by subglacial discharge and occurs as a summer pulse (now clarified earlier in the text, new lines 502-508). Please also see last comment to Reviewer 2.

L368-369: Again, error bounds please for both the 7nM and 0.4Mg Fe yr. What references and values are used for upscaling here? Does this account for all ocean terminating glaciers, fjord terminating glaciers, all glaciers? What is the geographical extent of your upscaling and what freshwater fluxes are specifically used? Arguments to upscale need to be more explicit and convincing, especially how upscaling was achieved. This is not a pan-Greenlandic calculation. We have just changed the units on a flux derived from our data, which is now clarified in the revision. The calculation now shown include alternative flux derivations (line 468-473, 481-496).

Additionally, not all these glaciers used in your model will have cavity residence times of 162 days - residence times are likely too dynamic between different glacial masses, but also annually, seasonally and depend on the thickness of ice, subglacial and supraglacial discharge flow, amount of sediment and distance between base and sea/fjord floor, local currents etc etc. e.g. L306 to 307 - Please expand on this calculation and explicitly state the assumptions. As above, this is not a pan-Greenlandic calculation, we have not used multiple catchments, we have just changed the units on the flux derived in this catchment. Assumptions are now explicitly stated and the basis of the calculation hopefully better explained.

L387 to 389: Again, the 79NG Glacier is notorious for being one of the fastest flowing glaciers, therefore is it representative of all glaciers flowing into the Arctic ocean? The calculations are also based on an extrapolation with extremely high error, so it would be good so see what the flux values are taking this into account. Flux values now show uncertainties (and as above, are specific to 79NG, not Greenland or any kind of scaling-up).

L403 to 411: Again, problems using a linear extrapolation. What happens to freshwater dFe estimates if this modelling approach is used instead of a linear regression? We now show all possible methods for assessing changes in nutrient concentrations in the supplement (and a selection in the main text).

L468: What was the pressure used for filtration? Does this forcing impact the size fractions going through the filter? 0.2 atm. This is standard GEOTRACES protocol and therefore yes it may affect the effective cut-off size of the filters but it is standardised across cruises. 0.2 atm pressurized filtration has unlikely a major effect as the 0.8/0.2 flow-through filters used to collect all samples except

LpFe are quite robust and include a large surface-area prefilter precisely to minimise sample collection artefacts as water passes through the second filter). The alternative to low overpressure (<0.2 atm), or no-overpressure, would result in it taking substantially longer to filter a station which would allow aggregation to proceed in the bottles and cause filtration artefacts associated with all the particles in the samplers sinking to the bottom during filtration.

Tables:

1: Very useful, please add on what was measured and what was calculated in this manuscript
Added.

Figures:

1: No S11 station on the figure
Added

2: Could probably be joined with Figure 1.

We have added plots concerning oxygen isotopic composition and meteoric freshwater content, thus, we think this figure better stands by itself.

3: Great, excellent figure.

Thank you. We have now added turbidity and natural radium isotopes profiles of station S1.

4: I don't think this adds anything to the main text and could go in the supplementary material

Done.

5: Great figure, but for my personal preference I think Figure S2 where you almost see the covariation that is referred to section 'The role of colloidal Fe in glacier-to-ocean fluxes'.

Thank you. Turbidity data is now shown in Figure 4 alongside LpFe. We think Figure S2 is now therefore obsolete.

6: Very useful, yet questions why you used a linear regression to find the fresh water composition of 7nM

Linear regression is the 'normal' method across a salinity gradient. Note that the curve here does not necessary demonstrate that salinity is not best fitted linearly, it shows that distance matters – but salinity also changes along this distance (non-linearly). Nevertheless, we now include GAM which explains dFe distribution as a function of the interaction between distance and salinity (Figure 6 and 7), this slightly refines the classic linear regression of salinity and produces a similar endmember (Table S3 and S4).

7: put error bar key on the figure, so that there is a visual guide to actual variation. Some of the anomalous concentrations overlap with the accepted data, why were they deemed anomalous? The location?

They shouldn't be described as 'anomalous' as they aren't anomalies, we just noticed that the high surface glacier-outflow dFe concentrations all occur at this station (S4) which is subject to high turbulence that may explain this. We have now put this figure into the supplement (Figure S7) and added error bars as suggested.

8: Why are stations 1 and 2 put together as one data point? Please separate these as it looks like they are additive. Please add a 1:1 line so that a comparison to the distributions of cFe to sFe can easily be seen by the reader.

Done. We have now plotted sFe/cFe data from stations 1, 2 and 4 separately (Figure 9). The linear fit uses the combined data.

Supplementary material:

Table:

Station number missing in column A197 to 215

Unfortunately, we do not know what the reviewer is referring to. We have checked all tables to see for missing station numbers.

References

- von Albedyll, L. (2018) *Structure and variability of the circulation at tidal to intra-seasonal time scales near the 79 North Glacier*. University of Bremen & Alfred Wegener Institute Helmholtz Centre for Polar and Marine Research. Available at: [hdl:10013/epic.31fe6a83-74cc-4e81-a4b7-6d0e4cc62d19](https://hdl.handle.net/10013/epic.31fe6a83-74cc-4e81-a4b7-6d0e4cc62d19).
- Andresen, C. S. *et al.* (2012) ‘Rapid response of Helheim Glacier in Greenland to climate variability over the past century’, *Nature Geoscience*, 5(1), pp. 37–41. doi: 10.1038/ngeo1349.
- Ardiningsih, I. *et al.* (2020) ‘Natural Fe-binding organic ligands in Fram Strait and over the northeast Greenland shelf’, *Marine Chemistry*. Elsevier B.V., 224, p. 103815. doi: 10.1016/j.marchem.2020.103815.
- Arrigo, K. R. *et al.* (2017) ‘Melting glaciers stimulate large summer phytoplankton blooms in southwest Greenland waters’, *Geophysical Research Letters*, 44(12), pp. 6278–6285. doi: 10.1002/2017GL073583.
- Beard, B. L. *et al.* (2003) ‘Iron isotope constraints on Fe cycling and mass balance in oxygenated Earth oceans’, *Geology*, 31(7), pp. 629–632.
- Cape, M. R. *et al.* (2019) ‘Nutrient release to oceans from buoyancy-driven upwelling at Greenland tidewater glaciers’, *Nature Geoscience*. Springer US, 12(1), pp. 34–39. doi: 10.1038/s41561-018-0268-4.
- Carroll, D. *et al.* (2015) ‘Modeling Turbulent Subglacial Meltwater Plumes: Implications for Fjord-Scale Buoyancy-Driven Circulation’, *Journal of Physical Oceanography*. American Meteorological Society, 45(8), pp. 2169–2185. doi: 10.1175/JPO-D-15-0033.1.
- Conway, T. M. and John, S. G. (2014) ‘Quantification of dissolved iron sources to the North Atlantic Ocean’, *Nature*. Nature Publishing Group, 511(7508), pp. 212–215. doi: 10.1038/nature13482.
- Fitzsimmons, J. N. *et al.* (2017) ‘Iron persistence in a distal hydrothermal plume supported by dissolved-particulate exchange’, *Nature Geoscience*, 10(3), pp. 195–201. doi: 10.1038/ngeo2900.
- Gerringa, L. J. A. *et al.* (2012) ‘Iron from melting glaciers fuels the phytoplankton blooms in Amundsen Sea (Southern Ocean): Iron biogeochemistry’, *Deep-Sea Research Part II*, 71–76, pp. 16–31. doi: 10.1016/j.dsr2.2012.03.007.
- Gledhill, M. and Buck, K. N. (2012) ‘The organic complexation of iron in the marine environment: A review’, *Frontiers in Microbiology*, 3(FEB), pp. 1–17. doi: 10.3389/fmicb.2012.00069.
- Hawkings, J. R. *et al.* (2017) ‘Ice sheets as a missing source of silica to the polar oceans’, *Nature communications*, 8:14198. doi: 10.1038/ncomms14198.
- Homoky, W. B. *et al.* (2012) ‘Dissolved oxygen and suspended particles regulate the benthic flux of iron from continental margins’, *Marine Chemistry*. Elsevier B.V., 134–135, pp. 59–70. doi: 10.1016/j.marchem.2012.03.003.
- Homoky, W. B. *et al.* (2013) ‘Distinct iron isotopic signatures and supply from marine sediment dissolution’, *Nature Communications*. Nature Publishing Group, 4(2143), pp. 1–10. doi: 10.1038/ncomms3143.
- Hopwood, M. J. *et al.* (2016) ‘Seasonal changes in Fe along a glaciated Greenlandic fjord’,

Frontiers in Earth Science, 4(15). doi: 10.3389/feart.2016.00015.

Hopwood, M. J. *et al.* (2020) 'Review Article: How does glacier discharge affect marine biogeochemistry and primary production in the Arctic?', *The Cryosphere*, 14, pp. 1347–1383. doi: 10.5194/tc-2019-136.

Juul-Pedersen, T. *et al.* (2015) 'Seasonal and interannual phytoplankton production in a sub-Arctic tidewater outlet glacier fjord, SW Greenland', *Marine Ecology Progress Series*, 524(March), pp. 27–38. doi: 10.3354/meps11174.

Kanna, N. *et al.* (2020) 'Iron Supply by Subglacial Discharge Into a Fjord Near the Front of a Marine-Terminating Glacier in Northwestern Greenland', *Global Biogeochemical Cycles*, 34(10), p. e2020GB006567. doi: 10.1029/2020GB006567.

Krisch, S. *et al.* (2020) 'The influence of Arctic Fe and Atlantic fixed N on summertime primary production in Fram Strait, North Greenland Sea', *Scientific Reports*. Nature Publishing Group UK, 10(15230), pp. 1–13. doi: 10.1038/s41598-020-72100-9.

Linhoff, B. S., Charette, M. A. and Wadham, J. (2020) 'Rapid mineral surface weathering beneath the Greenland Ice Sheet shown by radium and uranium isotopes', *Chemical Geology*, 547, p. 119663. doi: 10.1016/j.chemgeo.2020.119663.

Lippiatt, S. M., Lohan, M. C. and Bruland, K. W. (2010) 'The distribution of reactive iron in northern Gulf of Alaska coastal waters', *Marine Chemistry*. Elsevier B.V., 121(1–4), pp. 187–199. doi: 10.1016/j.marchem.2010.04.007.

Mankoff, K. D. *et al.* (2020) 'Greenland liquid water discharge from 1958 through 2019', *Earth System Science Data*, 12(4), pp. 2811–2841. doi: 10.5194/essd-12-2811-2020.

Marsay, C. M. *et al.* (2017) 'Distributions, sources, and transformations of dissolved and particulate iron on the Ross Sea continental shelf during summer', *Journal of Geophysical Research: Oceans*, 122(8), pp. 6371–6393. doi: 10.1002/2017JC013068.

Mayer, C. *et al.* (2000) 'The subglacial cavity and implied dynamics under Nioghalvfjærdsfjorden glacier, NE-Greenland', *Geophysical Research Letters*, 27(15), pp. 2289–2292. doi: 10.1029/2000GL011514.

Mayer, C. *et al.* (2018) 'Large ice loss variability at Nioghalvfjærdsfjorden Glacier, Northeast-Greenland', *Nature Communications*. Springer US, 9(1), pp. 1–11. doi: 10.1038/s41467-018-05180-x.

Meire, L. *et al.* (2016) 'High export of dissolved silica from the Greenland Ice Sheet', *Geophysical Research Letters*, 43(17), pp. 9173–9182. doi: 10.1002/2016GL070191.

Meire, L. *et al.* (2017) 'Marine-terminating glaciers sustain high productivity in Greenland fjords', *Global Change Biology*, 23(12), pp. 5344–5357. doi: 10.1111/gcb.13801.

Mortensen, J. *et al.* (2020) 'Subglacial Discharge and Its Down-Fjord Transformation in West Greenland Fjords With an Ice Mélange', *Journal of Geophysical Research: Oceans*, 125(9), p. e2020JC016301. doi: <https://doi.org/10.1029/2020JC016301>.

Nielsen, M. H., Erbs-Hansen, D. R. and Knudsen, K. L. (2010) 'Water masses in Kangerlussuaq, a large fjord in West Greenland: the processes of formation and the associated foraminiferal fauna', *Polar Research*, 29(2), pp. 159–175. doi: 10.1111/j.1751-8369.2010.00147.x.

Oliver, H. *et al.* (2018) 'Exploring the Potential Impact of Greenland Meltwater on Stratification, Photosynthetically Active Radiation, and Primary Production in the Labrador Sea', *Journal of Geophysical Research: Oceans*, 123(4), pp. 2570–2591. doi:

10.1002/2018JC013802.

Oliver, H. *et al.* (2020) 'Meltwater-Enhanced Nutrient Export From Greenland's Glacial Fjords: A Sensitivity Analysis', *Journal of Geophysical Research: Oceans*, 125(7), p. e2020JC016185. doi: 10.1029/2020JC016185.

Poitrasson, F. (2006) 'On the iron isotope homogeneity level of the continental crust', *Chemical Geology*, 235(1–2), pp. 195–200. doi: 10.1016/j.chemgeo.2006.06.010.

Radic, A., Lacan, F. and Murray, J. W. (2011) 'Iron isotopes in the seawater of the equatorial Pacific Ocean: New constraints for the oceanic iron cycle', *Earth and Planetary Science Letters*. Elsevier B.V., 306(1–2), pp. 1–10. doi: 10.1016/j.epsl.2011.03.015.

Raiswell, R. *et al.* (2018) 'Iron in Glacial Systems: Speciation, Reactivity, Freezing Behavior, and Alteration During Transport', *Frontiers in Earth Science*, 6(222), pp. 1–17. doi: 10.3389/feart.2018.00222.

Rapp, I. *et al.* (2019) 'Controls on redox-sensitive trace metals in the Mauritanian oxygen minimum zone', *Biogeosciences*, 16(21), pp. 4157–4182. doi: 10.5194/bg-16-4157-2019.

Reddy, T. E., Holland, D. M. and Arrigo, K. R. (2010) 'Ross ice shelf cavity circulation, residence time, and melting: Results from a model of oceanic chlorofluorocarbons', *Continental Shelf Research*. Elsevier, 30(7), pp. 733–742. doi: 10.1016/j.csr.2010.01.007.

Schaffer, J. *et al.* (2017) 'Warm water pathways toward Nioghalvfjærdssjøen Glacier, Northeast Greenland', *Journal of Geophysical Research: Oceans*, 122, pp. 4004–4020. doi: 10.1002/2016JC012462.

Schaffer, J. *et al.* (2020) 'Bathymetry constrains ocean heat supply to Greenland's largest glacier tongue', *Nature Geoscience*. Springer US, 13(3), pp. 227–231. doi: 10.1038/s41561-019-0529-x.

Schroth, A. W. *et al.* (2014) 'Estuarine removal of glacial iron and implications for iron fluxes to the ocean', *Geophysical Research Letters*, 41, pp. 3951–3958. doi: 10.1002/2014GL060199. Received.

Seifert, M. *et al.* (2019) 'Influence of glacial meltwater on summer biogeochemical cycles in Scoresby Sund, East Greenland', *Frontiers in Marine Science*, 6(412). doi: 10.3389/fmars.2019.00412.

Sherrell, R. M. *et al.* (2015) 'Dynamics of dissolved iron and other bioactive trace metals (Mn, Ni, Cu, Zn) in the Amundsen Sea Polynya, Antarctica', *Elementa*, 3:71. doi: 10.12952/journal.elementa.000071.

Syring, N. *et al.* (2020) 'Holocene Interactions Between Glacier Retreat, Sea Ice Formation, and Atlantic Water Advection at the Inner Northeast Greenland Continental Shelf', *Paleoceanography and Paleoclimatology*, 35(11), p. e2020PA004019. doi: 10.1029/2020PA004019.

Thuróczy, C.-E. *et al.* (2012) 'Key role of organic complexation of iron in sustaining phytoplankton blooms in the Pine Island and Amundsen Polynyas (Southern Ocean)', *Deep-Sea Research II*, 76, pp. 49–60. doi: 10.1016/j.dsr2.2012.03.009.

Wilson, N. J. and Straneo, F. (2015) 'Water exchange between the continental shelf and the cavity beneath Nioghalvfjærdssjøen (79 North Glacier)', *Geophysical Research Letters*, 42(18), pp. 7648–7654. doi: 10.1002/2015GL064944.

Wilson, N., Straneo, F. and Heimbach, P. (2017) 'Satellite-derived submarine melt rates and mass balance (2011–2015) for Greenland's largest remaining ice tongues', *Cryosphere*, 11(6),

pp. 2773–2782. doi: 10.5194/tc-11-2773-2017.

Wuttig, K. *et al.* (2013) ‘Impacts of dust deposition on dissolved trace metal concentrations (Mn, Al and Fe) during a mesocosm experiment’, *Biogeosciences*, 10(4), pp. 2583–2600. doi: 10.5194/bg-10-2583-2013.

Zhang, R. *et al.* (2015) ‘Transport and reaction of iron and iron stable isotopes in glacial meltwaters on Svalbard near Kongsfjorden: From rivers to estuary to ocean’, *Earth and Planetary Science Letters*. Elsevier B.V., 424, pp. 201–211. doi: 10.1016/j.epsl.2015.05.031.

REVIEWERS' COMMENTS

Reviewer #1 (Remarks to the Author):

The authors have made a number of improvements that have helped to clarify, and in my opinion, strengthen the paper. The removal of the silica discussion and re-focus on iron specifically and the new data (Ra, Fe isotopes) and analyses (RDA, GAMS) provided have served to hone the message and clarify the importance of this contribution. In my opinion, the manuscript is providing novel, important insights into a question of interest to a broad community and of debate at the moment – i.e. that of the importance of iron fluxes associated with glacial systems. The dataset is very valuable as the authors note because it can clearly delineate processes occurring as a result of mixing or inorganic effects from biological processes. Thus, the authors are able to make significant contributions to our understanding of iron fluxes from glacial systems. Thus, in conclusion, I believe that the paper should be accepted to this journal for publication. Below I offer some brief comments on pitch and clarity for the authors' consideration.

One thing that the authors might consider is that of the expectation they set-up in the abstract and introduction of the paper. In the abstract (L16-21) and introduction (L40-45) the authors are speaking about runoff (glacial freshwater discharge) that represents mass balance loss on the surface that is drained to the bed and then discharged at the submarine terminus (subglacial discharge). This accounts for ~50% of mass loss from the Greenland ice sheet (<https://www.nature.com/articles/s41586-019-1855-2>) and is indeed an important flux to consider if this is a source of Fe and other nutrients to the ocean. But at 79N the authors note in the site description (L95-97) that subglacial discharge is not the main source of glacial freshwater to the cavity. Instead basal ice melt (that the authors argue is de-coupled from the seasonal summer melting cycle) is the main source of freshwater exiting the cavity. This is very interesting in itself as there are few studies (to my knowledge) of basal ice melt contributions to the marine ecosystem, and this flux could of course increase with climate change, but this is different from the subglacial discharge released at the grounding line that presently accounts for a ~50% of mass loss from Greenland. My point being from first reading the abstract and introduction I was prepared for a paper talking about nutrient export in glacial meltwater that is modified en route to the ocean due to long-term storage in large cavities – but this is not quite the case since the majority of the freshwater is being supplied by basal ice melt – which is a different end-member source, distinct from sub-glacially discharged annual mass balance losses. So instead of tying the significance of this study to increasing mass balance loss perhaps the authors might consider instead emphasizing the uniqueness of this study for providing insight into the impact of submarine basal ice melt to marine environments?

L95-97: I'm not sure if I missed it, but is there an estimate for the freshwater flux exiting the cavity? Just wondering how this flux compares to reported discharges for other Greenland glacial systems.

L233: In the sentence "We conclude that the radium is sedimentary in origin (either from the seafloor or released from the ice tongue by melt)" .. does "released from the ice tongue by melt" refer to basal melt?

L280: Maybe it would be useful to introduce what a general additive model is, why it was applied here, and how it's been applied previously to other studies for those not familiar with the technique? I understand that space might be limited and this is somewhat described in the Figure 7 caption, but just a suggestion.

L351: I'm a bit unclear what this "sedimentary" source is .. only because in L351-352 the authors say that the Ra data implicates as sedimentary vs direct subglacial signal, but in L232-233 the author's state that the radium is sedimentary in origin "either from the seafloor or released from the ice tongue by melt".. perhaps this is just a wording issue but how is it in L351-L352 that the authors are able to

now (compared to earlier in L232-233) be more definitive that the radium source is not basal ice melt at the tongue. Perhaps there is some further meaning intended by the phrase "rather than [a] direct subglacial signal" (L351) that can be made more clear?

L408: I have to admit that the buffer analogy is lost on me .. but perhaps this is just me!

L434-435: I see the cavity freshwater flux mentioned here.. might be good to move to study site description and offer a comparison to other Greenland outlet glaciers subglacial discharge? I think this would be useful, for example, in evaluating if the lack of $\text{Si}(\text{OH})_4$ enrichment (L305-306) is due to the fact that the main glacial source to this system is basal ice melt (vs submarine discharge) or is a question of flux.

L524: Again here it would be good to clarify what the authors mean by "sedimentary, rather than subglacial in origin".. see notes above. Also in L504-505 the authors imply that the supply of labile Fe is coming from the freshwater..

It seems like the source to the cavity is glacial in origin (e.g. L346-347 "It is clear that the mAIW is enriched in dFe compared to the AIW and that this enrichment occurs either at the glacier terminus or underneath it") but then there is subsequent modification in the cavity that results in the signal in the emerging mAIW being re-worked and thus "not glacial"? I think the confusion (for me) is the use of the phrase "in origin" vs a signal that is glacial in origin but then is modified in the cavity.

Reviewer #2 (Remarks to the Author):

Many thanks for the second opportunity to review this manuscript. With the addition of the extra datasets and the overhaul of the statistical approach and calculations, my main comments have been fully addressed. The minor comments have largely been dealt with as well during the revision process. The manuscript has been improved greatly, and in my view is suitable for publication in Nature Communications.

I like the revised approach to tackling the decoupling of the particulate Fe and dFe, especially with the addition of the new data and the redundancy analysis. In addition to the new Figure 6, I actually think that the plots shown in the response to reviewers, of dissolved vs. particulates for other metals are informative (especially for Mn) and straightforwardly highlight the 'different' behaviour of Fe in this environment. My only suggestion here would be to include these plots (at least for Mn) in the supplementary information.

With regards to the general challenges surrounding the collection of particulate material from bottle samples (Niskin or Go-Flo), I did not intend to suggest that the authors' methods were not appropriate, just that there are inherent challenges in sampling particulates from bottles (caused by settling, loss of large particles in washout of pre-filters etc. especially when only a portion of the bottle is filtered). I refer the authors to the paper by McDonnell et al., 2015. This paper discusses the different methods of collecting particulate samples, and associate challenges, specifically mentioning trace metal-clean particle collection from GO-Flo bottles (note that according to McDonnell et al., the "same principles to minimize artifacts from particle settling apply to particle collection from Niskin bottles") in the context of the GEOTRACES program.

McDonnell, A. M., Lam, P. J., Lamborg, C. H., Buesseler, K. O., Sanders, R., Riley, J. S., ... & Bishop, J. K. (2015). The oceanographic toolbox for the collection of sinking and suspended marine particles.

Progress in oceanography, 133, 17-31.

Reviewer #3 (Remarks to the Author):

Firstly, apologies to the authors for the late submission of my review. I reviewed this manuscript when it was originally submitted and I am pleased to see the authors have made substantive modifications and thorough responses to the reviews. There is much improvement, however with the addition of a significant amount of new data (referenced) to bolster their interpretations it took me longer to re-review.

Manuscript

My immediate response is that the incorporation of a summary figure, the incorporation of supportive isotopic measurements and a much more robust statistical analysis of the data has vastly improved the paper. And I believe with some minor modifications under editorial handling it will be ready for publication.

The section starting L146: geographical nutrient distributions I think is purely descriptive and in a journal such as NatComm the writing should be focussed on interpretations and discussion. I feel this section could be greatly shortened with detailed presentation of the profiles etc in the SI. Much of the data for the nutrients N and P are as expected in the ocean, so I think it would be better to report and discuss the relevant Si and Fe data, and also make those the focus in Figure 5.

The statistical treatment of the data is vastly improved and I am very please do see the linear models compared with the RDA analysis and the GAM. I very much like how the data is then presented in Figure 7.

I am happy with the data quality of the isotopic data and it adds much to their arguments. However to my knowledge Fe isotopes, especially in low concentrations, are notoriously hard to analyse. It would be good if the authors could provide some information on the quantity of Fe used in column chemistry and analysis, more details on how many samples were mixed to provide a data point, and a procedural blank.

There are currently over 100 references, I think this should be significantly reduced.

Finally, apologies for my idiocy again but just to make sure... am I right in thinking that both dFe and dFeL are the same Fe pool (both encompass $<0.2\mu\text{m}$), but that the dFeL is the ligand bound but still dissolved dFe but it is not colloidal Fe? i.e. colloidal Fe (cFe) is a different Fe pool from the dissolved ligand bound Fe dFeL but the dFe encompasses all of these (sFe, cFe and dFeL)?

In line comments

L32: I am still not sure what the authors mean by downstream environments, as in the Ocean? As phrased, this still suggests a river.

L228 to 229: Was the value of 0.07mSv calculated using the data in this paper?

L315 to 316: in this discussion of dFe, you should refer to Figure 2 maybe, how much of it is dFe(XS), dFeL etc?

L393 to 408: Very well written, succinct and clear.

L419 to 421: What is bacterial mineralisation source of ligands? Does this mean inorganic ligands formed by bacteria? How are these modified and what does it mean for the suspension of Fe? Does this change the distribution of cFe and dFeL? What makes these weaker ligands? I think a bit more explanation is needed here rather than just a reference.

L543 to 454: But remember you are comparing results to an Fe limited region in the southern ocean

to one which is not (79NG).

L595: missing a word in this sentence, 'while 'we' refer'

L613 to 614: What was the quantity of Fe put through chemistry? How large was the procedural blank?

Figure 1: Great

Figure 2: Fantastic, a great addition to the paper

Figure 3: Fine

Figure 4: The interesting data lies in panels C, D, E, F & G, panels A and B are fairly typical so these could be moved to the SI to make the other figures larger and stand out more.

Why is the Si(OH₄) peak (in panel C) at a higher depth ~100 m, compared to that of LpFe (panel D) ~ 200m?

Figure 5: Fine

Figure 6: Fine

Figure 7: Can you add on the station numbers to the plots?

Figure 8: great

Figure 9: Fine

Table 1: There is also an dFe(xs) term in Figure 2 which needs to be added in here.

Supplementary Information

I find the supplementary notes very useful, and they make some great points about discrepancies in the literature. A plot of the d⁵⁴Fe isotopes here would be nice too, to get a feel for the range relative to the other literature sited in the manuscript - point to crustal values and glacial run off values etc.

Reviewer #1 (Remarks to the Author):

The authors have made a number of improvements that have helped to clarify, and in my opinion, strengthen the paper. The removal of the silica discussion and re-focus on iron specifically and the new data (Ra, Fe isotopes) and analyses (RDA, GAMs) provided have served to hone the message and clarify the importance of this contribution. In my opinion, the manuscript is providing novel, important insights into a question of interest to a broad community and of debate at the moment – i.e. that of the importance of iron fluxes associated with glacial systems. The dataset is very valuable as the authors note because it can clearly delineate processes occurring as a result of mixing or inorganic effects from biological processes. Thus, the authors are able to make significant contributions to our understanding of iron fluxes from glacial systems. Thus, in conclusion, I believe that the paper should be accepted to this journal for publication. Below I offer some brief comments on pitch and clarity for the authors' consideration.

Thank you very much for these kind words. We appreciate that the manuscript also in the light of the reviewer's feedback can contribute to the current debate on glacial dFe fluxes.

One thing that the authors might consider is that of the expectation they set-up in the abstract and introduction of the paper. In the abstract (L16-21) and introduction (L40-45) the authors are speaking about runoff (glacial freshwater discharge) that represents mass balance loss on the surface that is drained to the bed and then discharged at the submarine terminus (subglacial discharge). This accounts for ~50% of mass loss from the Greenland ice sheet (<https://www.nature.com/articles/s41586-019-1855-2>) and is indeed an important flux to consider if this is a source of Fe and other nutrients to the ocean. But at 79N the authors note in the site description (L95-97) that subglacial discharge is not the main source of glacial freshwater to the cavity. Instead basal ice melt (that the authors argue is de-coupled from the seasonal summer melting cycle) is the main source of freshwater exiting the cavity. This is very interesting in itself as there are few studies (to my knowledge) of basal ice melt contributions to the marine ecosystem, and this flux could of course increase with climate change, but this is different from the subglacial discharge released at the grounding line that presently accounts for a ~50% of mass loss from Greenland. My point being from first reading the abstract and introduction I was prepared for a paper talking about nutrient export in glacial meltwater that is modified en route to the ocean due to long-term storage in large cavities – but this is not quite the case since the majority of the freshwater is being supplied by basal ice melt – which is a different end-member source, distinct from sub-glacially discharged annual mass balance losses.

So instead of tying the significance of this study to increasing mass balance loss perhaps the authors might consider instead emphasizing the uniqueness of this study for providing insight into the impact of submarine basal ice melt to marine environments?

It is correct that basal ice melt provides the larger fraction of freshwater volume into the cavity (ca 89% of subglacial freshwater flux, Schaffer *et al.*, 2020). However, considering reported high

dFe concentrations in glacial runoff (e.g. Bhatia *et al.*, 2013; Hawkings *et al.*, 2014, 2020), subglacial discharge would still provide the largest freshwater-derived nutrient inputs into the cavity because ice melt contains sparse concentrations of dFe/macronutrients with concentrations presumably more akin to iceberg values. The same conclusion can be made for most other chemical components we are aware of. While this may be a further reason why dFe in cavity exiting waters is low compared to dFe in waters from land-terminating glaciers, we are lacking ice samples from the 79NG floating ice tongue and, as noted in previous replies, the dynamics of nutrients in this system are very similar to those in the two other East Greenland case studies available (which are more ‘normal’ Greenlandic systems with runoff, rather than basal ice melt, forming the major freshwater outflow). We have added an extra sentence to the discussion (line 511):

*“For large marine-terminating glaciers where ice melt provides the largest fraction of freshwater to the cavity (□89% for 79NG)(Schaffer *et al.*, 2020), estimates on freshwater nutrient inputs derived from surface runoff endmembers (e.g. refs.(Bhatia *et al.*, 2013; Hawkings *et al.*, 2014, 2020)) are likely overestimating glacial dFe shelf supply, also because ice melt contains comparatively sparse amounts of dFe, with concentrations presumably more akin to iceberg values(Hopwood *et al.*, 2017). An alternative hypothesis, which is supported by work in a range of catchments beyond 79NG, is that ligand dynamics and overturning - which are interrelated - constrain lateral export(Lippiatt, Lohan and Bruland, 2010; Thuróczy *et al.*, 2012; Ardiningsih *et al.*, 2020).”*

L95-97: I’m not sure if I missed it, but is there an estimate for the freshwater flux exiting the cavity? Just wondering how this flux compares to reported discharges for other Greenland glacial systems.

Based on the most recent investigation of Schaffer *et al.* (2020), freshwater contributes ca. 1.4% to the cavity overturning circulation which is in line with our calculation of freshwater to AIW of 1:84 in cavity exiting mAIW (see lines 105-107 and 455-457, respectively).

L233: In the sentence “We conclude that the radium is sedimentary in origin (either from the seafloor or released from the ice tongue by melt)” does “released from the ice tongue by melt” refer to basal melt?

As the findings from Linhoff, Charette and Wadham (2020) suggest, glacial freshwater without contribution from sediments is very low in ^{228}Ra and ^{226}Ra . Thus, basal melt per-se is unlikely a significant contributor of ^{226}Ra and ^{228}Ra to cavity exiting mAIW. The calculations presented in the paragraph show that subglacial discharge, presumably enriched to a comparable extent found in subglacial runoff from the Leverett Glacier, is also unlikely a substantial source of ^{226}Ra and ^{228}Ra to cavity-exiting mAIW. Instead, enrichment of ^{226}Ra and ^{228}Ra observed at 125 m at

station S1 must originate from sedimentary sources inside the cavity, may this be from benthic release or from sediment material entrained within the ice-tongue, and released into the water column upon melting. We have modified the sentence accordingly (line 228):

“With the discharge-averaged concentrations found in subglacial runoff of the Leverett Glacier in western Greenland (Linhoff, Charette and Wadham, 2020) this would, after mixing in the cavity, cause increases of only $0.03 \text{ dpm}\cdot\text{m}^{-3}$ ^{226}Ra and $0.35 \text{ dpm}\cdot\text{m}^{-3}$ ^{228}Ra in the cavity outflow, negligible compared to the observed signals at 125 m. We conclude that the radium is sedimentary in origin, either emerging from the seafloor or from sediments entrained within the ice-tongue and released into the water column by basal melting.”

L280: Maybe it would be useful to introduce what a general additive model is, why it was applied here, and how it's been applied previously to other studies for those not familiar with the technique? I understand that space might be limited and this is somewhat described in the Figure 7 caption, but just a suggestion.

We have added an introductory sentence to clarify why we use general additive models in line 283:

“Through the use of General Additive Models (GAMs) on water mass properties downstream of 79NG, it is possible to investigate nutrient-distance relationships that are not necessarily linear (Hastie and Tibshirani, 1986).”

L351: I'm a bit unclear what this “sedimentary” source is .. only because in L351-352 the authors say that the Ra data implicates as sedimentary vs direct subglacial signal, but in L232-233 the author's state that the radium is sedimentary in origin “either from the seafloor or released from the ice tongue by melt”.. perhaps this is just a wording issue but how is it in L351-L352 that the authors are able to now (compared to earlier in L232-233) be more definitive that the radium source is not basal ice melt at the tongue. Perhaps there is some further meaning intended by the phrase “rather than [a] direct subglacial signal” (L351) that can be made more clear?

We have clarified our statement from line 233 (now line 232), and hope in this context, it is now clear that the Ra signal must originate from sediments (may they be from benthic sources or sediments entrained in the ice-tongue. Basal meltwater, without the addition of sediments, does not contribute to ^{226}Ra and ^{228}Ra inventories in mAIW.

L408: I have to admit that the buffer analogy is lost on me. but perhaps this is just me!

This is not an analogy. Buffering is the correct scientific (chemical) term when a concentration of a species in solution is held at a certain level although the concentration of its ‘parent species’ is changing, i.e. particles-dissolved exchange holds the concentration of dFe at a fixed concentration irrespective of changes to the particle load.

L434-435: I see the cavity freshwater flux mentioned here.. might be good to move to study site description and offer a comparison to other Greenland outlet glaciers subglacial discharge? I think this would be useful, for example, in evaluating if the lack of Si(OH)₄ enrichment (L305-306) is due to the fact that the main glacial source to this system is basal ice melt (vs submarine discharge) or is a question of flux.

As suggested, we have amended this sentence to provide some context for the size of the basal ice melt (line 91):

“Basal melt and subglacial discharge at 79NG combined result in a total cavity freshwater flux of 0.63 ± 0.21 mSv (19.9 ± 6.6 km³·yr⁻¹)(Schaffer et al., 2020), which is roughly equivalent to 2% of annual runoff from the Greenland Ice Sheet (2016 values)(Bamber et al., 2018).”

Yet, it is very hard to find other catchments to provide a meaningful specific value for the basal ice melt fraction, which is typically much smaller (as we indicate from line 478 and 542) but poorly constrained compared to runoff and generally not calculated in the same way. We could quote studies determining the ice melt occurring at vertical calving ice termini, but this is a very different context to a floating ice tongue and we are not sure it adds much to the discussion herein. As we point out in the discussion, Si dynamics at 79NG are not dissimilar from those in Sermilik or Scoresby Sund, so this issue is not specific to the basal ice melt.

L524: Again here it would be good to clarify what the authors mean by “sedimentary, rather than subglacial in origin”. see notes above.

We have clarified this statement in line 539:

“Furthermore, the chemical signatures in mAIW denote a strong control of sedimentary processes in modulating cavity-exiting waters, suggesting limited influence of seasonal shifts in meltwater properties.”

Also in L504-505 the authors imply that the supply of labile Fe is coming from the freshwater.

We have modified this statement (line 517):

“In systems like 79NG, where shelf forcing rather than freshwater discharge controls the residence time of water masses within the glacier fjord(von Albedyll, 2018; Schaffer et al.,

2020), lateral dFe flux may be completely insensitive to short-term changes in freshwater and sedimentary dFe or $LpFe$ supply (although is still ultimately dependent on a supply of labile Fe to maintain a large $LpFe$ pool within the subglacial cavity) and far more sensitive to changes in ligand properties of AIW and cavity inflow rate.”

It seems like the source to the cavity is glacial in origin (e.g. L346-347 “It is clear that the mAIW is enriched in dFe compared to the AIW and that this enrichment occurs either at the glacier terminus or underneath it”) but then there is subsequent modification in the cavity that results in the signal in the emerging mAIW being re-worked and thus “not glacial”? I think the confusion (for me) is the use of the phrase “in origin” vs a signal that is glacial in origin but then is modified in the cavity.

We have revised the statement (see comment above). With respect to our statement in line 346 (now line 351), we cannot be conclusive that all sediment in the cavity is derived from the glacier. It is highly likely the vast majority of sediment deposited in the inner-cavity is from the glacier, but particles are also transported into the cavity with the inflowing AIW entering the cavity. Inflow of AIW may for example deposit POC in the benthos and so it does not follow that a sedimentary signal emerging from the cavity has to be glacial in origin (although it likely is at least partially the case that a lot of the Fe released into solution was originally deposited in glacier-derived particles). See also the recent sedimentary Fe work by Laufer-Meiser *et al.* (2021), for example: Again a very different context, but if we super-imposed the same qualitative gradient in benthic- Fe cycling onto 79NG, it’s quite plausible that re-working of particles in the outer parts of the cavity, that weren’t glacial in origin, would have a disproportionately large effect on the transformation of Fe species within the cavity.

Reviewer #2 (Remarks to the Author):

Many thanks for the second opportunity to review this manuscript. With the addition of the extra datasets and the overhaul of the statistical approach and calculations, my main comments have been fully addressed. The minor comments have largely been dealt with as well during the revision process. The manuscript has been improved greatly, and in my view is suitable for publication in Nature Communications.

I like the revised approach to tackling the decoupling of the particulate Fe and dFe, especially with the addition of the new data and the redundancy analysis. In addition to the new Figure 6, I actually think that the plots shown in the response to reviewers, of dissolved vs. particulates for other metals are informative (especially for Mn) and straightforwardly highlight the 'different' behaviour of Fe in this environment. My only suggestion here would be to include these plots (at least for Mn) in the supplementary information.

Thank you very much for your kind words and your support. We have added a figure of dMn vs LpMn and TdMn (Supplementary Figure 7C+D).

With regards to the general challenges surrounding the collection of particulate material from bottle samples (Niskin or Go-Flo), I did not intend to suggest that the authors' methods were not appropriate, just that there are inherent challenges in sampling particulates from bottles (caused by settling, loss of large particles in washout of pre-filters etc. especially when only a portion of the bottle is filtered). I refer the authors to the paper by McDonnell et al., 2015. This paper discusses the different methods of collecting particulate samples, and associate challenges, specifically mentioning trace metal-clean particle collection from GO-Flo bottles (note that according to McDonnell et al., the "same principles to minimize artifacts from particle settling apply to particle collection from Niskin bottles") in the context of the GEOTRACES program.

McDonnell, A. M., Lam, P. J., Lamborg, C. H., Buesseler, K. O., Sanders, R., Riley, J. S., ... & Bishop, J. K. (2015). The oceanographic toolbox for the collection of sinking and suspended marine particles. *Progress in oceanography*, 133, 17-31.

Thank you for providing us with this interesting literature. We have modified our statement in line 412 accordingly.

"Although there are inevitably biases in sampling of particles from water collectors depending on the apparatus used(McDonnell et al., 2015), the observed relationship between dissolved and particulate fractions of Fe is fundamentally different to other metals. For example, an increase in dMn at enhanced particle concentrations is evident at the 79NG marine terminus (Supplementary Figure 7)."

Reviewer #3 (Remarks to the Author):

Firstly, apologies to the authors for the late submission of my review. I reviewed this manuscript when it was originally submitted and I am pleased to see the authors have made substantive modifications and thorough responses to the reviews. There is much improvement, however with the addition of a significant amount of new data (referenced) to bolster their interpretations it took me longer to re-review.

Thank you for having reviewed our paper. We appreciate the time and effort spent by the reviewers.

Manuscript

My immediate response is that the incorporation of a summary figure, the incorporation of supportive isotopic measurements and a much more robust statistical analysis of the data has vastly improved the paper. And I believe with some minor modifications under editorial handling it will be ready for publication.

The section starting L146: geographical nutrient distributions I think is purely descriptive and in a journal such as NatComm the writing should be focussed on interpretations and discussion. I feel this section could be greatly shortened with detailed presentation of the profiles etc in the SI. Much of the data for the nutrients N and P are as expected in the ocean, so I think it would be better to report and discuss the relevant Si and Fe data, and also make those the focus in Figure 5.

We agreed with this comment originally as no such plots were included in the 1st submission, but there were many comments from the other 2 reviewers that could only be robustly evaluated with such a statistical approach, particularly the suggestion that we were 'missing' a Si(OH₄) source – which is actually re-circulation of Atlantic water. The PCA/GAMs show that silicic acid only to an insignificant extent diverges from NO₃/PO₄, that PO₄ shows some local depletion relative to NO₃, and that Fe diverges from all 3 provides insight into processes which can't easily be 'seen', unless you are very familiar with what salinity-nutrient distributions look like in different cases, without these plots. This may be a case of who the reader is, as someone without a background in marine chemistry might find these plots much easier to visualize than salinity-nutrient distributions alone. In light of the comments from other reviewers, we do think that these will help readers, potentially from other disciplines.

The statistical treatment of the data is vastly improved and I am very please do see the linear models compared with the RDA analysis and the GAM. I very much like how the data is then presented in Figure 7.

I am happy with the data quality of the isotopic data and it adds much to their arguments. However, to my knowledge, Fe isotopes especially in low concentrations are notoriously hard to analyse. It would be good if the authors could provide some information on the quantity of Fe used in column chemistry and analysis, more details on how many samples were mixed to provide a data point, and a procedural blank.

This is an important point, and we have added some detail to the methods as requested. Iron isotope analysis was performed at the University of South Florida (USF) on pooled seawater samples that gave 10-20 ng of Fe per sample, and concentrations of 20-40 ng/mL for analysis (when dissolved in 0.5 mL). This number corresponds to at least 40x our established procedural blank at USF of 0.24 ± 0.04 (1SD, n=10) ng Fe per sample by the same analyst (see Summers, 2020) – equivalent to the 0.28 ng reported previously for this procedure by Conway *et al.* (2013). The quantity of Fe measured in this study (the equivalent of 1L of 0.5-2.3 nM dFe) is within the range routinely analysed at USF for seawater samples, where we can reliably and demonstrably measure as low as the equivalent of 0.08 nM in 1L of seawater (Sieber *et al.* (in revision); Conway *et al.*, 2013; Summers, 2020). We have modified the method accordingly (line 626):

*“Stable Fe isotopic composition ($\delta^{56}\text{Fe}$) was measured in seawater samples at the University of South Florida (USF) following a modification of ref.(Conway *et al.*, 2013). Filtered trace metal samples were combined from several depths (see Supplementary Table 9) to provide sufficient dFe for isotope analysis (10-20 ng). Fe was extracted from acidified seawater samples using a two-stage resin extraction method (Nobias PA-1 at pH 2, and AGMP-1), followed by analysis by Thermo Neptune Plus multi-collector ICP-MS with introduction via an ESI Apex Ω desolvator (with Ar but not N₂ added gas), Pt Jet and Al ‘x-type’ sample cones. The procedural blank using this method at USF has been established at 0.24 ± 0.04 ng Fe (1SD, n = 10), similar to previous studies(Conway *et al.*, 2013). Instrumental mass bias was accounted for using an ⁵⁷Fe-⁵⁸Fe double spike, with $\delta^{56}\text{Fe}$ expressed in standard delta notation relative to the IRMM-014 standard. For this method we assess long-term instrumental precision using repeat analyses of the NIST-3126 Fe standard, for which we obtain a $\delta^{56}\text{Fe}$ value of $+0.36 \pm 0.04\%$ (2SD, n = 351 during 24 sessions over 3 years). We apply this value as an estimate of 2 σ uncertainty on $\delta^{56}\text{Fe}$ for all samples, except for samples where the larger 2 standard internal error is a more conservative estimate of uncertainty, as applies to samples in this study.”*

Details on the mixture of samples can be found in the Supplementary Table 9.

There are currently over 100 references, I think this should be significantly reduced.

We are aware that this manuscript is stretching beyond the limit of 70 references as asked for by *Nature Communications*. It was closer to this limit when we submitted the first draft, but the extensive additions of new data have necessitated new references. Given the perhaps unusually large number of datasets which are discussed - NO₃, PO₄, Si(OH)₄, Fe, Mn, Co, Ra, ¹⁸O, Fe-

isotopes all require at least some introduction/discussion - we think it would be very difficult to present the manuscript with fewer references without substantial loss of information/context.

Finally, apologies for my idiocy again but just to make sure... am I right in thinking that both dFe and dFeL are the same Fe pool (both encompass $<0.2\mu\text{m}$), but that the dFeL is the ligand bound but still dissolved dFe but it is not colloidal Fe? i.e. colloidal Fe (cFe) is a different Fe pool from the dissolved ligand bound Fe dFeL but the dFe encompasses all of these (sFe, cFe and dFeL)?

Not at all idiocy, dFe and dFe-L are similar, but not exactly the same. By our definition (physical speciation), dissolved Fe is filtered $<0.2\mu\text{M}$ and comprises of the soluble Fe pool (filtered $<0.02\mu\text{M}$) and colloidal Fe (calculated from $\text{dFe} - \text{sFe}$, thus equivalent to the size range of $0.02\text{-}0.2\mu\text{M}$). By far the greatest fraction of dissolved Fe is complexed, thus stabilized in the water column by organic and inorganic ligands (chemical speciation). In general, $>99\%$ of oceanic dFe is stabilized by ligand material (Gledhill and Buck, 2012) which is what we refer to as dFeL. DFeL comprises of ligand-bound cFe and ligand-bound sFe. Some rather small fraction of dFe (typically $<1\%$) is free (i.e. uncomplexed) $\text{Fe}^{2+}/\text{Fe}^{3+}$ and, by our size cut-off, part of the sFe pool.

In line comments

L32: I am still not sure what the authors mean by downstream environments, as in the Ocean? As phrased, this still suggests a river.

We have modified the abstract and rephrased the corresponding line (now line 24):

“Our findings indicate that the overturning rate and particle-dissolved phase exchanges in ice cavities exerts a dominant control on subglacial nutrient supply to shelf regions.”

L228 to 229: Was the value of 0.07mSv calculated using the data in this paper?

Yes, we have clarified this (line 227):

“Subglacial runoff is estimated to be 0.07 mSv ($2.2\text{ km}^3\cdot\text{yr}^{-1}$) or only 0.15% of the cavity overturning rate (calculated from ref.(Schaffer et al., 2020)).”

L315 to 316: in this discussion of dFe, you should refer to Figure 2 maybe, how much of it is dFe(XS), dFeL etc?

This is hard to say, as we do not know the amount of dFe entering the cavity, either from subglacial discharge or basal melt. By measuring mAIW properties adjacent to the 79NG

terminus, we only know the combined effects of dFe supply to the cavity and thus cannot provide any information on the amount of dFe(xs) aggregating and sedimenting inside the cavity. The fraction of dFe complexed by ligand material (which is what we refer to as dFeL) remains similarly elusive. The Fe-binding ligand pool is extremely heterogenic (e.g. Gledhill and Buck, 2012), and the method applied by e.g. Ardiningsih *et al.* (2020) to samples on the NE Greenland Shelf can only determine the most prominent ligand classes. Clearly, there are other, also weaker ligands complexing and stabilizing dFe in mAIW too, but these remain undetermined (using any method, this is not a specific critique to the work by Ardiningsih *et al.*). We refer the reader to a recent publication by Ardiningsih *et al.* (2021) where specifics of ligand determination with focus on the NE Greenland Shelf is further discussed.

L393 to 408: Very well written, succinct and clear.

L419 to 421: What is bacterial mineralisation source of ligands? Does this mean inorganic ligands formed by bacteria? How are these modified and what does it mean for the suspension of Fe? Does this change the distribution of cFe and dFeL? What makes these weaker ligands? I think a bit more explanation is needed here rather than just a reference.

Bacterial remineralisation and degradation of organic material forms typically weaker (organic and inorganic) ligands, meaning their affinity to bind to dFe, thus stabilize dFe in solution, is decreased. It is beyond the scope of this study to go into greater detail on how ligands near 79NG are exactly transformed or what makes them different from ligands further offshore. However, it seems plausible that following the initiation of the spring bloom, organic material from surface waters on the shelf is exported into cavity-entering AIW, where, in the absence of new primary production, primarily degradation of ligand material is taking place. As for the nature of inorganic ligands, and the gross or organic ligands that are not determined by the ligand determination method (see comment above), we do not know how they shape dFe export from the cavity. For more information on this respect, we refer the reader to the review of Gledhill and Buck (2012) and reference therein.

We have modified the paragraph accordingly (line 428):

*“Analysis of dFe-binding ligands from the GN05 cruise suggests bacterial remineralisation and organic degradation have transformed the pool of ligands close to the 79NG terminus, resulting in elevated ligand concentrations but overall weaker ligands that are in sum less efficient in stabilizing dFe(Ardiningsih *et al.*, 2020). The absence of ligand production in the euphotic zone in combination with degradation of organic material(Gledhill and Buck, 2012) underneath 79NG may explain why dFe-binding ligands, generally undersaturated on the shelf ($45 \pm 17\%$ at S2, S8-10), were approaching saturation at the terminus ($63 \pm 13\%$ at S1) with a ligand concentration of 2.3 ± 0.2 nM eq. Fe and a corresponding dFe concentration of 1.5 ± 0.3 nM(Ardiningsih *et al.*, 2020).”*

L543 to 454: But remember you are comparing results to an Fe limited region in the Southern Ocean to one which is not (79NG).

Yes, all the more remarkable that we see some similarities at some SO ice shelves, which points towards strong buffering mechanisms.

L595: missing a word in this sentence, 'while 'we' refer'.

This is correct; thank you.

L613 to 614: What was the quantity of Fe put through chemistry? How large was the procedural blank?

Please see reply to earlier comment.

Figure 1: Great

Figure 2: Fantastic, a great addition to the paper

Figure 3: Fine

Figure 4: The interesting data lies in panels C, D, E, F & G, panels A and B are fairly typical so these could be moved to the SI to make the other figures larger and stand out more.

This was originally the case. However, in the light of questions from reviewers 1 and 2, we have added panels A and B to the figure in the main text.

Why is the Si(OH)₄ peak (in panel C) at a higher depth ~100 m, compared to that of LpFe (panel D) ~ 200m?

We have no specific answer to this yet there is no explicit reason why they should, other than approximately, especially at high resolution because particle-dissolved dynamics are different for the two nutrients. There are likely subtle differences in how Fe and Si are cycled within the cavity and thus when/where the peak concentrations occur. Concerning LpFe and Si(OH)₄ for example, the residence time of both species is very different in the ocean, with LpFe being subject to rapid sedimentation. Comparing dFe and Si(OH)₄, dFe is scavenged intensively whereas Si(OH)₄ is not. So at broad-scales we would roughly expect to see anomalies in Fe and Si(OH)₄ correlating with areas where there is runoff/discharge, but there are several reasons why the peaks in these different species may not co-occur. It is to some extent a circular answer, they are different because they are different, but the PCA makes it clear the drivers behind all Fe species and Si(OH)₄ are different.

Figure 5: Fine

Figure 6: Fine

Figure 7: Can you add on the station numbers to the plots?

We have adapted Figure 7 accordingly.

Figure 8: great

Figure 9: Fine

Table 1: There is also an $dFe(xs)$ term in Figure 2 which needs to be added in here.

We do not have any information on the magnitude of $dFe(xs)$ nor its species or chemistry, which is the reason why we have not dealt with $dFe(xs)$ in the text (or Table 1). Please see our comment on $dFe(xs)$ above.

Supplementary Information

I find the supplementary notes very useful, and they make some great points about discrepancies in the literature. A plot of the $d^{54}Fe$ isotopes here would be nice too, to get a feel for the range relative to the other literature sited in the manuscript - point to crustal values and glacial run off values etc.

There is little data on Fe isotopic composition of glacial systems in the Arctic. We refer the reviewer to the paper of Stevenson *et al.* (2017) who have compiled $\delta^{56}Fe$ measurements of glacial sites around Greenland. Besides our measurements near 79NG and Zachariæ Isstrøm, we are not aware of other Fe isotopic measurements of glacial sites in Greenland that could contribute to this earlier compilation. Excluding Greenland, there is only two glaciated catchments in the Arctic (i.e. Svalbard (Zhang *et al.*, 2015) and Alaska (Escoube *et al.*, 2015)) where $\delta^{56}Fe$ measurements have also been conducted (all of which were cited in our manuscript). Hence, we feel plotting Fe isotopic composition for glacial sites in the Arctic would largely reproduce Stevenson's work and we could not add much to this discussion.

Literature

von Albedyll, L. (2018) *Structure and variability of the circulation at tidal to intra-seasonal time scales near the 79 North Glacier*. University of Bremen & Alfred Wegener Institute Helmholtz Centre for Polar and Marine Research. Available at: [hdl:10013/epic.31fe6a83-74cc-4e81-a4b7-6d0e4cc62d19](https://hdl.handle.net/10013/epic.31fe6a83-74cc-4e81-a4b7-6d0e4cc62d19).

Ardiningsih, I. *et al.* (2020) 'Natural Fe-binding organic ligands in Fram Strait and over the northeast Greenland shelf', *Marine Chemistry*. Elsevier B.V, 224:103815. doi: 10.1016/j.marchem.2020.103815.

Ardiningsih, I. *et al.* (2021) 'Iron Speciation in Fram Strait and Over the Northeast Greenland Shelf: An Inter-Comparison Study of Voltammetric Methods', *Frontiers in Marine Science*, 7, p. 609379. doi: 10.3389/fmars.2020.609379.

Bamber, J. L. *et al.* (2018) 'Land Ice Freshwater Budget of the Arctic and North Atlantic Oceans: 1. Data, Methods, and Results', *Journal of Geophysical Research: Oceans*, 123(3), pp. 1827–1837. doi: 10.1002/2017JC013605.

Bhatia, M. P. *et al.* (2013) 'Greenland meltwater as a significant and potentially bioavailable source of iron to the ocean', *Nature Geoscience*. Nature Publishing Group, 6(4), pp. 274–278. doi: 10.1038/ngeo1746.

Conway, T. M. *et al.* (2013) 'A new method for precise determination of iron, zinc and cadmium stable isotope ratios in seawater by double-spike mass spectrometry', *Analytica Chimica Acta*. Elsevier B.V., 793, pp. 44–52. doi: 10.1016/j.aca.2013.07.025.

Escoube, R. *et al.* (2015) 'Iron isotope systematics in Arctic rivers', *Comptes Rendus - Geoscience*. Academie des sciences, 347(7–8), pp. 377–385. doi: 10.1016/j.crte.2015.04.005.

Gledhill, M. and Buck, K. N. (2012) 'The organic complexation of iron in the marine environment: A review', *Frontiers in Microbiology*, 3:69. doi: 10.3389/fmicb.2012.00069.

Hastie, T. and Tibshirani, R. (1986) 'Generalized additive models', *Statistical Science*, 1(3), pp. 297–310. doi: 10.1214/ss/1177013604.

Hawkings, J. R. *et al.* (2014) 'Ice sheets as a significant source of highly reactive nanoparticulate iron to the oceans', *Nature Communications*. Nature Publishing Group, 5:3929. doi: 10.1038/ncomms4929.

Hawkings, J. R. *et al.* (2020) 'Enhanced trace element mobilization by Earth's ice sheets', *Proceedings of the National Academy of Sciences of the United States of America*, 117(50), pp. 31648–31659. doi: 10.1073/pnas.2014378117.

Hopwood, M. J. *et al.* (2017) 'The heterogeneous nature of Fe delivery from melting icebergs', *Geochemical Perspectives Letters*, 3, pp. 200–209. doi: 10.7185/geochemlet.1723.

Laufer-Meiser, K. *et al.* (2021) 'Bioavailable iron produced through benthic cycling in glaciated Arctic fjords (Svalbard)', *Nature Communications*. Springer US, 12:1349. doi: 10.1038/s41467-021-21558-w.

Linhoff, B. S., Charette, M. A. and Wadham, J. (2020) 'Rapid mineral surface weathering

beneath the Greenland Ice Sheet shown by radium and uranium isotopes', *Chemical Geology*, 547:119663. doi: 10.1016/j.chemgeo.2020.119663.

Lippiatt, S. M., Lohan, M. C. and Bruland, K. W. (2010) 'The distribution of reactive iron in northern Gulf of Alaska coastal waters', *Marine Chemistry*. Elsevier B.V., 121(1–4), pp. 187–199. doi: 10.1016/j.marchem.2010.04.007.

McDonnell, A. M. P. *et al.* (2015) 'The oceanographic toolbox for the collection of sinking and suspended marine particles', *Progress in Oceanography*. Elsevier Ltd, 133, pp. 17–31. doi: 10.1016/j.pocean.2015.01.007.

Schaffer, J. *et al.* (2020) 'Bathymetry constrains ocean heat supply to Greenland's largest glacier tongue', *Nature Geoscience*. Springer US, 13(3), pp. 227–231. doi: 10.1038/s41561-019-0529-x.

Sieber, M. *et al.* (no date) 'Isotopic fingerprinting of biogeochemical processes and iron sources in the iron-limited surface Southern Ocean', *Earth and Planetary Science Letters (in revision)*.

Stevenson, E. I. *et al.* (2017) 'The iron isotopic composition of subglacial streams draining the Greenland ice sheet', *Geochimica et Cosmochimica Acta*. Elsevier Ltd, 213, pp. 237–254. doi: 10.1016/j.gca.2017.06.002.

Summers, B. A. (2020) *Investigating the Isotope Signatures of Dissolved Iron in the Southern Atlantic Ocean*. University of South Florida.

Thuróczy, C.-E. *et al.* (2012) 'Key role of organic complexation of iron in sustaining phytoplankton blooms in the Pine Island and Amundsen Polynyas (Southern Ocean)', *Deep-Sea Research II*, 76, pp. 49–60. doi: 10.1016/j.dsr2.2012.03.009.

Zhang, R. *et al.* (2015) 'Transport and reaction of iron and iron stable isotopes in glacial meltwaters on Svalbard near Kongsfjorden: From rivers to estuary to ocean', *Earth and Planetary Science Letters*. Elsevier B.V., 424, pp. 201–211. doi: 10.1016/j.epsl.2015.05.031.